# Theory of Chiral Electrodeposition by Chiral Micro-Nano-Vortices under a Vertical Magnetic Field -1: 2D Nucleation by Micro-Vortices

**Ryoichi Morimoto [1,\*], Miki Miura [2], Atsushi Sugiyama [3,4,5], Makoto Miura [6], Yoshinobu Oshikiri [7], Iwao Mogi [8], Yusuke Yamauchi [4,9], Satoshi Takagi [10] and Ryoichi Aogaki [11,\*]**

1. Saitama Industrial Technology Center, 3-12-18, Kamiaoki, Kawaguchi 333-0844, Japan
2. Polytechnic Center Kimitsu, 428, Sakata, Kimitsu 299-1142, Japan; miki3@mug.biglobe.ne.jp
3. Research Organization for Nano and Life Innovation, Waseda University, 513, Waseda Tsurumaki-cho, Shinjuku-ku, Tokyo 162-0041, Japan; a.sugiyama@yoshinodenka.com
4. JST-ERATO Yamauchi Materials Space-Tectonics Project and International Center for Materials Nano-Architectonics (WPI-MANA), National Institute for Materials Science, 1-1, Namiki, Tsukuba 305-0044, Japan; y.yamauchi@uq.edu.au
5. R&D Division, Yoshino Denka Kogyo, Inc., 1-2, Asahi, Yoshikawa 342-0008, Japan
6. Architectural Construction Systems Technology, Tohoku Polytechnic College, 26, Tsukidate Hagisawa Dobashi, Kurihara 987-2223, Japan; miura.makoto@tohoku-pc.ac.jp
7. Department of Architectural and Environmental Engineering, Yamagata College of Industry and Technology, 2-2-1, Matsuei, Yamagata 990-2473, Japan; oshikiri@yamagata-cit.ac.jp
8. Institute for Materials Research, Tohoku University, 2-1-1, Katahira, Aoba-ku, Sendai 980-8577, Japan; iwao.mogi.d4@tohoku.ac.jp
9. School of Chemical Engineering and Australian Institute for Bioengineering and Nanotechnology (AIBN), The University of Queensland, Brisbane, QLD 4072, Australia
10. Graduate School of Symbiotic Systems Science and Technology, Fukushima University, 1, Kanayagawa, Fukushima 960-1296, Japan; s1871001@ipc.fukushima-u.ac.jp
11. Department of Product Design, Polytechnic University, 2-20-12-1304, Ryogoku, Sumida-ku, Tokyo 130-0026, Japan
\* Correspondence: morimotoryoichi@mbn.nifty.com (R.M.); ryoaochan@aol.com (R.A.)

**Abstract:** Remarkable chiral activity is donated to a copper deposit surface by magneto-electrodeposition, whose exact mechanism has been clarified by the three-generation model. In copper deposition under a vertical magnetic field, a macroscopic tornado-like rotation called the vertical magnetohydrodynamic (MHD) flow (VMHDF) emerges on a disk electrode, inducing the precessional motions of various chiral microscopic MHD vortices: First, chiral two-dimensional (2D) nuclei develop on an electrode by micro-MHD vortices. Then, chiral three-dimensional (3D) nuclei grow on a chiral 2D nucleus by chiral nano-MHD vortices. Finally, chiral screw dislocations are created on a chiral 3D nucleus by chiral ultra-micro MHD vortices. These three processes constitute nesting boxes, leading to a limiting enantiomeric excess (*ee*) ratio of 0.125. This means that almost all chiral activity of copper electrodes made by this method cannot exceed 0.125. It also became obvious that chirality inversion by chloride additive arises from the change from unstable to stable nucleation by the specific adsorption of it.

**Keywords:** chirality; chiral electrodeposition; magnetic field; nucleation; micro-MHD vortex; nano-MHD vortex

## 1. Introduction

In recent years, it has been found that ionic vacancies are produced in solution phases as byproducts of electrode reactions [1,2]. Ionic vacancies are charged particles created to keep the conservations of linear momentum and electricity during electron transfers in electrode reactions. The initially created embryo vacancies are similar to ions isolated in free space, energetically unstable in solution phases. In accordance with the Debye–Hückel theory, ions in solution phases are stabilized by the solvation, surrounded by ionic clouds.

At the same time, from the ionic clouds, the solvation energies are liberated, producing entropies around the ions. This is the reason why the activities of ions are less than 1.0. In the same way, the embryo vacancies are also stabilized by the solvation, surrounded by ionic clouds. However, since embryo vacancies are composed of minute free spaces, the liberated solvation energies are not dispersed as heat but used for the dynamic works to enlarge their free-space cores, and stored in the cores. As a result, it is concluded that in the solvation, ionic vacancies do not produce entropies. As shown in Figure 1a,b, a solvated ionic vacancy is a charged free space of the order of 0.1 nm, surrounded by polarized solvent molecules and an ionic cloud with opposite charges. Though collided by surrounding solvent molecules in a collision time of $10^{-10}$ s, a solvated ionic vacancy keeps an intrinsic lifetime of 1 s [3,4], which is, compared with the collision time, extraordinarily long. This result strongly suggests that an ionic vacancy behaves as an iso-entropic particle without entropy production during transfer. Plainly, an ionic vacancy plays a role of an atomic scale lubricant, so that a vacancy layer formed on the electrode provides a free surface without friction, and the viscosity of the layer drastically decreases to zero. Such features have been validated by various experiments [3–6].

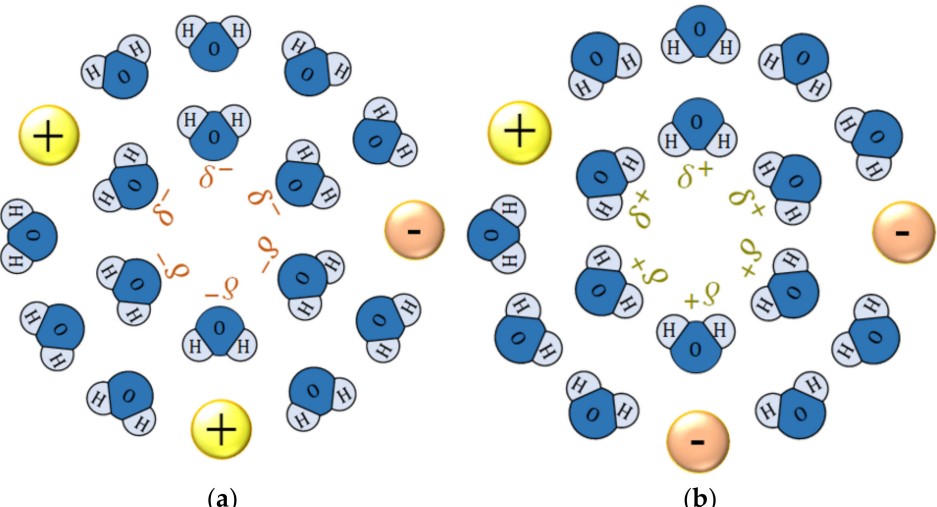

(a)                                                                                  (b)

**Figure 1.** Solvated ionic vacancies. (**a**) Negative ionic vacancy. (**b**) Positive ionic vacancy. H, proton; O, oxygen atom; $\delta^+$ and $\delta^-$, partial polarized charges of water molecules surrounding the free spaces; $\oplus$, cation; $\ominus$, anion.

In an electrode reaction under a vertical magnetic field, as shown in Figure 2, a macroscopic tornado-like rotation called the vertical magnetohydrodynamic (MHD) flow (VMHDF) emerges over a disk electrode with a fringe (fringed vertical MHD electrode (fringed-VMHDE)). In the preceding papers [5,6], the processes of the MHD rotation and the resulting mass transfer have been clarified. The electrode surface is covered with ionic vacancies, providing a free surface without friction. A fringed-VMHDF is divided by an upper rotational layer and a lower radial flow layer. The rotation of the upper layer is driven by the Lorentz force, and the radial flow arises from the pressure difference on the fringe of the electrode. By removing the fringe, we will find only a rotating piston-like flow without the radial flow layer.

Mogi and co-workers have been experimentally clarifying the chirality-emerging processes of copper electrodeposition under VMHDFs [7–18]. Chiral deposit films of copper were fabricated by chiral microscopic vortices called micro- and nano-MHD flows, formed on and in vacancy layers, which have chiral activities for enantiomeric reactions of amino acids. By changing the direction of the magnetic field and electrochemical conditions, various modes of chirality emergence are possible. The most important point of this process is that microscopic chiral vortices create chiral screw dislocations with chiral activities. Such chirality of the vortices is caused by the precession from the VMHDF [8,19]. The

rotational direction of the VMHDF is determined by the direction of the vertical magnetic field [5,6]; upward and downward magnetic fields provide the anticlockwise (ACW) and clockwise (CW) rotations, respectively. From these experimental results, it is concluded that the rotation of a VMHDF induces chiral precessions of the micro- and nano-MHD flows, which in turn produce chiral screw dislocations with chiral activities. From the hydrodynamic point of view, such microscopic vortices are only permitted in the case of a drastic decrease of viscosity or zero viscosity, i.e., the viscosity of the ordinary solution is too high for them to rotate. However, fortunately, the ionic vacancies mentioned above assist the vortex rotations with zero viscosity.

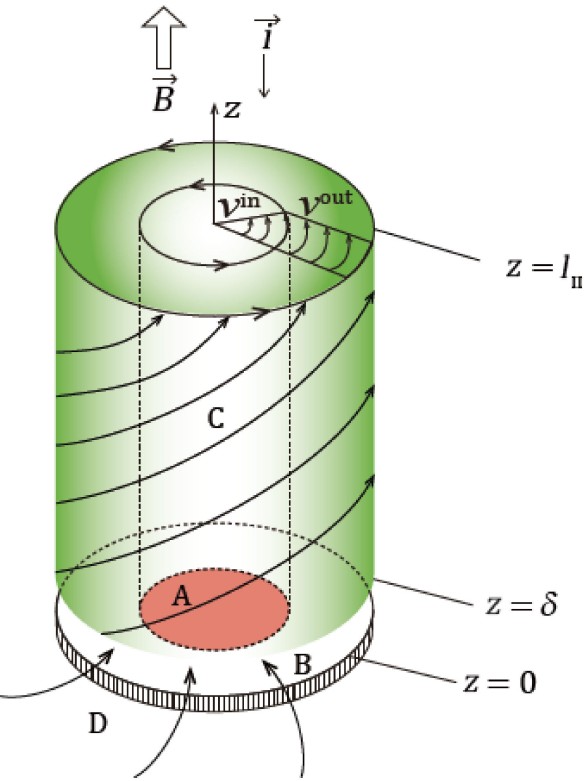

**Figure 2.** Schematic of a VMHDF on a fringed-VMHDE [6]. A, copper disk electrode; B, doughnut-shaped fringe of VMHDE; C, rotational-boundary-layer-flow; D, radial-boundary-layer-flow. $\delta$, the thickness of the radial boundary layer (~0.1 mm); $l_{II}$, the height of the rotational boundary layer (~several mm); $v^{in}$, inner tangential velocity on the electrode; $v^{out}$, outer tangential velocity on the fringe; $\vec{B}$, magnetic flux density; $\vec{i}$, current density; $z$, $z$ -axis. Reproduced with permission from Takagi, S.; Asada, T.; Oshikiri, Y.; Miura, M.; Morimoto, R.; Sugiyama, A.; Mogi, I.; Aogaki, R., *Journal of Electroanalytical Chemistry*; published by Elsevier B.V., 2022.

However, several important problems are still open to us; the scale of the length of a VMHDF (~1 mm) is $10^7$ times as large as that of a screw dislocation (~0.1 nm). The first question is—how are such chiral screw dislocations created by the rotation of the VMHDF despite extremely different scales of length? (Q1). Generally, nucleation in electrodeposition is classified into 2D nucleation of the order of 0.1 mm, 3D nucleation of the order of 0.1 μm, and screw dislocations of the order of 0.1 nm. Therefore, the emergence of the chiral activity would be composed of the three generations of chiral nuclei, i.e., chiral 2D nucleus, chiral 3D nucleus, and chiral screw dislocation. A chiral screw dislocation is created on a chiral 3D nucleus, which in turn grows on a chiral 2D nucleus developing under a VMHDF. These three processes form a nesting-boxes structure. As will be clarified later, based on a simple evidence, the fact that the chiral activity arises from the three generations is validated from both theoretical and experimental aspects. As for 2D and 3D nuclei, the nucleation

processes under parallel magnetic fields have been established [20–23], so in the present papers, we should examine how chiral 2D and 3D nuclei emerge under VMHDF rotations.

The chiral activity of the electrode is estimated by the enantiomeric excess (*ee*) ratio introduced by Mogi [9–18]. A characteristic fact derived from the previous experimental results is that the obtained *ee* ratios are distributed around 0.1. Does such a not-so-high ratio mean the low efficiency of this method? To tell the truth, it is important evidence for the three-generation model.

Furthermore, to receive the precessional motions from VMHDF rotations, vortices must revolve around a vertical axis with the VMHDF, whereas to create chiral nuclei fixed on the electrode, the vortices must keep their positions constant without any transfer. The second question is—how should such incompatible situations concerning the vortices be solved? That is, how do the fixed vortices without revolution receive the precessions from the VMHDF? (Q2).

Since the microscopic vortices are activated from a stationary state in the lower layer, to conserve their total angular momentums, the evolution probabilities of the vortices with ACW rotations must be equal to that of the vortices with CW rotations. Individual vortices have ACW or CW rotation, and adjoining vortices form a pair of vortices with opposite rotations. Even if one of a pair of the vortices receives the precession, due to the continuity of vortex motion, the opposite rotation of the other vortex is also enhanced. If the pair were composed of equivalent vortices with opposite rotations, we could not discriminate the selectivity of the precession, and would always observe achiral activities. To overcome such a contradiction, i.e., for either vortex to receive the precession, we must have two different types of vortices. If both kinds of vortices had similar properties, chirality breakdown would easily occur. The third question is—how are the two kinds of vortices self-organized? How is the precession selectively donated to either of them? (Q3).

From the above Mogi reports [9–18], the phenomenon that the chiral activity changes with the direction of an applied magnetic field is called "odd chirality". Namely, the copper films deposited under antiparallel (upward) and parallel (downward) magnetic fields provide D- (CW) and L- (ACW) chiral activities, respectively, which are opposed to the rotational chirality of the VMHDF mentioned above [18]. The fourth question is—why are the chiral activities of the electrode not consistent with the rotational chirality of the VMHDF? (Q4).

Then, Mogi also reported that by adding chloride additives, D-chiral activity changes to L-chiral activity, showing L-activity in both magnetic-field directions (the breakdown of odd chirality) [10,13]. The final and fifth question is—what is the mechanism of such a chirality change? (Q5).

To examine the microscopic processes mentioned above, it is necessary to precisely analyze the vertical MHD flow based on hydrodynamic and MHD theories. Fortunately, in magnetoelectrochemistry, we have already obtained various useful means for the analysis of electrochemical reactions under a magnetic field. Over five decades, many researchers have been struggling to develop magnetoelectrochemistry [24–32]. In electrode reactions under magnetic fields, two kinds of forces, i.e., Lorentz force and gradient field force (Kelvin force) appear. The Lorentz force often overwhelms the gradient field force, yielding a macroscopic convection called MHD flow. In accordance with Fahidy [27–29], MHD flow decreases the thickness of the diffusion layer, promoting mass transfer in the electrode reaction (MHD effect). As will be mentioned later, such a magnetic field effect on micro-electrodes was, as shown below, theoretically analyzed by Olivier [30–32]. Regarding the MHD flow in a channel electrode, called the MHD electrode, the diffusion current equations of the boundary layer flow and viscous flow were proposed by Aogaki [33–35].

Mutschke and co-workers examined electrodepositions in cuboid cells under magnetic fields accompanied by 3D convections affected by a gravitational field, which were numerically simulated in various cases [36,37]. Another important aspect of the MHD effect can influence the phase composition of composite metals (Olivier, Alemany, Daltin, Chopart, Hinds, Coey, Zabiński) [38–46].

The heterogeneous magnetic field yields magneto-convection by the gradient field force in a paramagnetic solution, enhancing the mass transfer process [47–49]. For the MHD effect under magnetic gradient fields, it has been clarified that a superimposed Lorentz force provides more complicated effects to deposit the pattern and composition (Tschulik, Uhlemann, Mutschke, Dunne, Coey) [50–54].

For the analysis of the electrochemical reaction using a micro-disk-electrode under a magnetic field, we can refer to some important achievements as follows: By using a micro-disk-electrode, Olivier established electrochemical impedance spectroscopy in a magnetic field [30,55,56], and at the same time found that the steady-state currents measured under parallel magnetic fields are proportional to $B^{1/3}C^{*4/3}$, where $B$ is the magnetic flux density and $C^*$ is the concentration of the electroactive species. White and coworkers [57–59] performed the investigation of magneto-electrochemical effects at ultra-micro-disk electrodes. The magnetic field effects on the limiting (steady-state) current were studied by using cyclic voltammetry in non-aqueous systems containing organic reactants acting as electroactive species. They concluded that the magnetic field effect was attributed to the convective flow caused by the viscous drag of the electrolyte ions accelerated by the magnetic force.

Recently, Mutschke and coworkers studied the electrodeposition of copper on a conically shaped diamagnetic electrode under the influence of a vertical magnetic field [60]. Using magnet arrays of small cylindrical magnets, Dunne and Coey studied deposit patterns of cathodic electrodeposition reflecting the non-uniform magnetic field [51,61].

The MHD electrode proposed by Aogaki was composed of a rectangular channel with two open edges, and a rectangular cathode and anode pair were face-to-face embedded on the inner walls. This type of electrode has been used for the measurement of the excess heat production by the pair annihilation of ionic vacancies with opposite signs created in cathodic and anodic reactions [62,63]. As for MHDE, the instability theory of nonequilibrium fluctuations in copper electrodeposition under a uniform parallel magnetic field has been first established, and examined for various deposition modes, especially concerning the effect of specific adsorption of ions [22]. Moreover, for measuring the lifetime of ionic vacancy, we have developed a new type of MHDE called cyclotron MHD electrode (CMHDE) [4,64], which is composed of a pair of partly shielded concentric cylindrical electrodes operated under a magnetic field. Ionic vacancies created in an electrode reaction circulate with an electrolyte solution by the Lorentz force.

Based on these various preceding attempts, in Part 1 of the present papers, we first formulate the theoretical equations of the microscopic vortex motions and mass transfer process under a vertical magnetic field. Then, by using the equations, characteristic morphological patterns called micro-mystery circles formed by 2D nucleation under a vertical magnetic field are calculated, and at the same time, the questions mentioned above are solved. The effect of chloride additive on the chirality is also examined. In Part 2, with the theoretical equations obtained in Part 1, the chiral 3D nucleation on a 2D nucleus will be treated.

## 2. Theory

In accordance with a vertical MHD flow (VMHDF) examined elsewhere [6], we introduce a simple model applicable to the three generations under assumptions of continuous fluid: On the electrode surface, as shown in Figure 3a, two types of solution layers are formed; the upper thick layer rotates around a $z$-axis, and in view of the pinning effect of the downward vortices, the thin lower layer is assumed stationary. The electrode surface is covered with ionic vacancies produced by electrode reactions, which are iso-entropic, making the surface free without friction in 2D nucleation. In the case of 3D nucleation as well as screw dislocation, due to smaller sizes than the thickness of the vacancy layer, the solution viscosity around vortices is assumed zero, if possible. The electrode surface or a flat surface of a nucleus is taken as an $x - y$ plane, and the $z$-axis is defined at the center in the upper direction so that the electrode phase is defined by $z \leq 0$, whereas the area of

$z > 0$ corresponds to the solution phase. The downward electrolytic current density in a metal deposition is thus defined as negative, so according to Mogi's definition, let us call upward (positive) and downward (negative) magnetic fields as antiparallel and parallel magnetic fields, respectively.

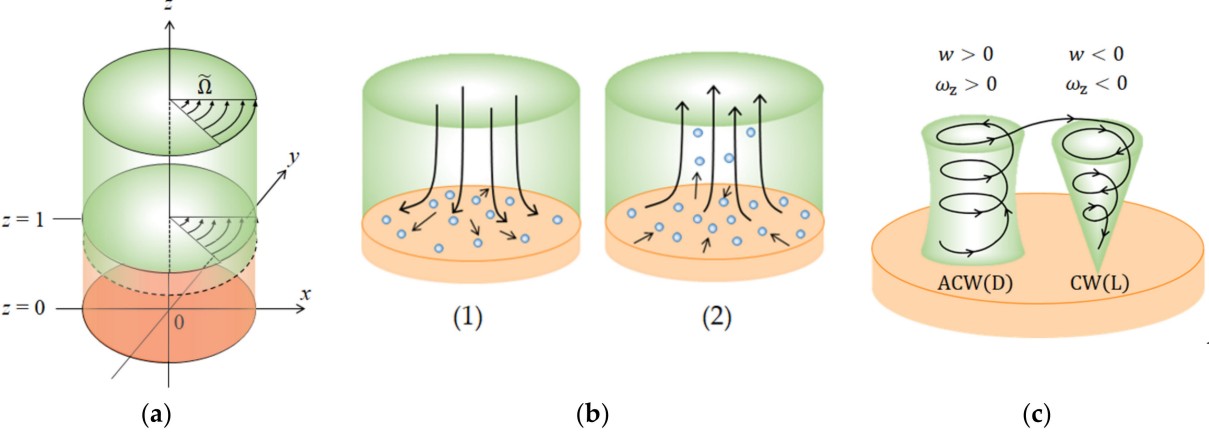

**Figure 3.** Chiral vortex formation under a rotating fluid layer. (**a**) Two layers model of the chiral nucleation. (**b**) Two kinds of vortices. (1) Downward vortex; (2) upward vortex. (**c**) Continuity of vortex motion and fluid flow. $x$, $y$, and $z$ stand for the non-dimensional coordinates normalized by the average size of vortices.

Adopting the notation of a right-handed system, we can define anticlockwise (ACW) and clockwise (CW) rotations in a bird's eye view as positive and negative, respectively. In the lower layer, microscopic vortices are first activated by the vertical magnetic field. As will be discussed precisely, they are composed of numerous pairs of vortices with upward and downward flows, respectively. As shown in Figure 3b, a downward vortex blows away ionic vacancies by the downward flow at the bottom, locally exposing rigid surfaces with friction. It works as a kind of pin to fix the vortex at a given point. In view of a nesting-boxes structure, the locally exposed surfaces correspond to the bottoms of the smaller-level vortices with downward flows. Since the positions are kept constant, at the bottoms of the vortices, chiral nuclei develop with time (1). An upward vortex pumps up ionic vacancies with the upward flow from the electrode surface so that its bottom is covered with ionic vacancies, forming a free surface without friction. The bottom of an upward vortex, differently from a downward vortex, rotates on the free surface, providing a flat surface without chiral nuclei. Though such a self-rotation, due to the pinning effect of the downward vortices, the upward vortex also does not move with the VMHDF, keeping the position constant (2). That is, downward vortices work as pins to stop the lower layer vortices to revolve with the upper layer, whereas the upward vortices supply free surfaces covered with ionic vacancies. Such different types of surfaces provide the different growth rates of vortices determining which vortices receive the precessions. At the same time, Figure 3c shows an important fact that, from the continuity of the vortex motion, a pair of adjoining upward and downward vortices must rotate in opposite directions. Then, as will be shown in Figure 4a, the individual vortex motions in the lower layer are transferred to the upper layer, where the newly induced vortices rotate with the upper layer, receiving the precessions. The precessional motions of the vortices in the upper layer are donated to the vortices in the lower layer.

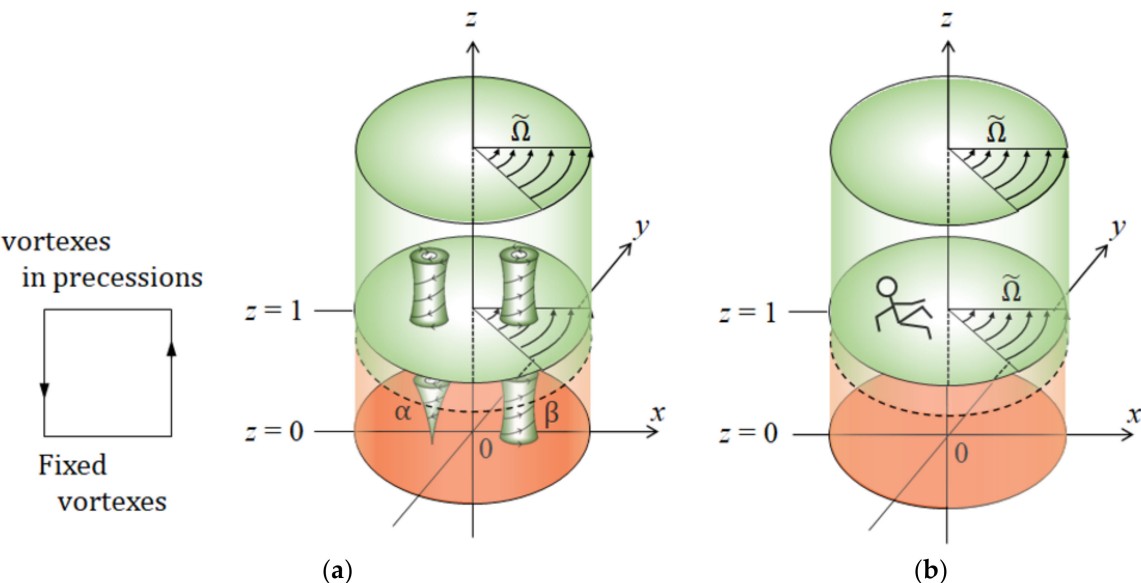

**Figure 4.** Two-layer model of chiral nucleation. (**a**) A feedback cycle between the vortices in the lower and upper layers. (**b**) An observer on a frame of reference rotating with the upper layer. *x, y,* and *z* stand for the non-dimensional coordinates normalized by the average size of vortices.

### 2.1. Vortex Motions in the Stationary Lower Layer

First, we consider explicitly the inertial frame with a static magnetic field. Because the sizes of fluctuations are much smaller than the belonging area of the electrode, a Cartesian coordinate system $(x, y, z)$ is taken for the special area, i.e., the electrode surface for the 2D nucleation or the surface of a 2D nucleus for the 3D nucleation.

Then, we consider an incompressible fluid at a uniform temperature, so the basic equations are given in the following (Appendix B) [65]. The momentum equation is in tensor notation,

$$\frac{\partial u_i}{\partial t} + u_j \frac{\partial u_i}{\partial x_j} - \frac{B_j}{\rho \mu_0} \frac{\partial B_i}{\partial x_j} = \nu \nabla^2 u_i - \frac{\partial}{\partial x_i}\left( \frac{P}{\rho} + \frac{\left|\vec{B}\right|^2}{2\mu_0 \rho} \right) \tag{B7}$$

where $u_i$. is the velocity component of vortices (i = 1, 2, 3), and the Cartesian coordinate $(x, y, z)$ is expressed by $(x_1, x_2, x_3)$. $\nu$ and $\rho$ are the kinematic viscosity and the density, respectively. $P$ is the pressure, and $\mu_0$ is the magnetic permeability. $\vec{B}$ is the magnetic flux density, and $B_i$ is the i-component of $\vec{B}$.

In view of an incompressible fluid, the continuity equation is obtained.

$$\frac{\partial u_i}{\partial x_i} = 0 \tag{B8}$$

In an electrolytic solution, the electricity is carried by diffusion as well as conductivity of ionic species, so that the current density will be given by

$$\vec{J} = \sigma^*\left( \vec{E} + \vec{u} \times \vec{B} \right) - F \sum_i z_i D_i \nabla C_i \tag{B9}$$

where $\vec{E}$ is the electric field, and $\sigma^*$ is the electrical conductivity defined by

$$\sigma^* = F^2 \sum_i z_i^2 \lambda_i^* C_i \tag{B10}$$

where $z_i$ is the charge number, including the sign, $\lambda_i^*$ is the mobility, $F$ is the Faraday's constant, $C_i$ is the concentration of ionic species i, and $D_i$ is the diffusion coefficient. The equation of magnetic flux density is simply written in the tensor notation as follows.

$$\frac{\partial B_i}{\partial t} + \frac{\partial}{\partial x_j}\left(u_j B_i - u_i B_j\right) = \eta \nabla^2 \vec{B} \tag{B16}$$

where $\eta$ is the resistivity defined by

$$\eta \equiv \frac{1}{\sigma^* \mu_0} \tag{B14}$$

and $\nabla^2$ implies $\partial^2/\partial x_1^2 + \partial^2/\partial x_2^2 + \partial^2/\partial x_3^2$.

As the reaction proceeds, the magnetic flux density first fluctuates, expressed by (Appendix C)

$$\vec{B} = \vec{B}^* + \vec{b} \tag{C1}$$

where $\vec{B}^*$ is the external magnetic flux density in the absence of the reaction and $\vec{b}$ is the fluctuation by the reaction. The fluctuation of the Lorentz force is written as

$$f_{Li} = \frac{\partial}{\partial x_i}\left(\frac{\vec{b} \cdot \vec{B}^*}{\mu_0}\right) + B_j^* \frac{\partial}{\partial x_j}\left(\frac{b_i}{\mu_0}\right) \tag{C2}$$

The concentration of the metallic ion is expressed by

$$C_m = C_m^* + c_m \tag{C6}$$

where $C_m^*$ and $c_m$ are the concentration in the absence of fluctuation and the concentration fluctuation, respectively.

The mass transfer equation of the fluctuation is written as

$$\frac{\partial c_m}{\partial t} + w L_m = D_m \nabla^2 c_m \tag{C7}$$

where $w$ is the z-component of the velocity, $u_3$. $L_m$ is the average concentration gradient in the diffusion layer.

$$L_m \equiv \frac{\theta_\infty^*}{\langle \delta_c \rangle} \tag{C8}$$

where $\theta_\infty^*$ implies the concentration difference between the bulk and the surface and $\langle \delta_c \rangle$ is the average thickness of a diffusion layer.

The i-component of the current density fluctuation is in tensor notation.

$$j_i = \frac{1}{\mu_0} \varepsilon_{ijk} \frac{\partial}{\partial x_j} b_k \tag{C9}$$

where $\varepsilon_{ijk}$ denotes the transposition of the tensor. Then, the i-component of the vorticity $\omega_i$ is given by

$$\omega_i = \varepsilon_{ijk} \frac{\partial}{\partial x_j} u_k \tag{C10}$$

Then, we shall restrict our discussion of this problem to the case where magnetic flux density is imposed vertically to the electrode.

$$\vec{B}^* = (0,\ 0,\ B_0) \tag{C19}$$

where $B_0$ is the vertical magnetic flux density with the sign. Therefore, we obtain the fluctuation equations.

$$\frac{\partial b_z}{\partial t} = \eta \nabla^2 b_z + B_0 \frac{\partial w}{\partial z} \tag{C20a}$$

$$\frac{\partial j_z}{\partial t} = \eta \nabla^2 j_z + \frac{B_0}{\mu_0} \frac{\partial \omega_z}{\partial z} \tag{C20b}$$

$$\frac{\partial \omega_z}{\partial t} = \nu \nabla^2 \omega_z + \frac{B_0}{\rho} \frac{\partial j_z}{\partial z} \tag{C20c}$$

$$\frac{\partial}{\partial t} \nabla^2 w = \nu \nabla^4 w + \frac{B_0}{\rho \mu_0} \frac{\partial}{\partial z} \nabla^2 b_z \tag{C20d}$$

### 2.2. Amplitude Equations of the Fluctuations in the Lower Layer

For the fluctuations including vortex motions, we assume the following 2D plane waves (Appendix D).

$$w = W^0(z,t) \exp\left[i(k_x x + k_y y)\right] \tag{D1a}$$

$$\omega_z = \Omega^0(z,t) \exp\left[i(k_x x + k_y y)\right] \tag{D1b}$$

$$b_z = K^0(z,t) \exp\left[i(k_x x + k_y y)\right] \tag{D1c}$$

$$j_z = J^0(z,t) \exp\left[i(k_x x + k_y y)\right] \tag{D1d}$$

$$c_m = \Theta^0(z,t) \exp\left[i(k_x x + k_y y)\right] \tag{D1e}$$

where $W^0(z,t)$, $\Omega^0(z,t)$, $K^0(z,t)$, $J^0(z,t)$, and $\Theta^0(z,t)$ are the amplitudes of the fluctuations, and $k_x$ and $k_y$ are the wavenumbers in the $x$- and $y$-directions, respectively.

Substituting Equations (D1a)–(D1e) into Equations (C7) and (C20a)–(C20e), we obtain the amplitude equations. Since the fluctuations are at quasi-steady states, neglecting the time-differential terms, we finally have

$$\left(D^2 - k^2\right) K^0 = -\left(\frac{B_0}{\eta}\right) D W^0 \tag{D3a}$$

$$\left(D^2 - k^2\right) J^0 = -\left(\frac{B_0}{\mu_0 \eta}\right) D \Omega^0 \tag{D3b}$$

$$\left(D^2 - k^2\right) \Omega^0 = -\left(\frac{B_0}{\rho \nu}\right) D J^0 \tag{D3c}$$

$$\left(D^2 - k^2\right)^2 W^0 = -\left(\frac{B_0}{\mu_0 \rho \nu}\right) D \left(D^2 - k^2\right) K^0 \tag{D3d}$$

$$\left(D^2 - k^2\right) \Theta^0 = \left(\frac{L_m}{D_m}\right) W^0 \tag{D3e}$$

where $D \equiv d/dz$ and $k \equiv \left(k_x^2 + k_y^2\right)^{1/2}$.

Substituting Equation (D3b) into Equation (D3c), and using Equation (B14) in Appendix B, we have

$$\left\{\left(D^2 - k^2\right)^2 - QD^2\right\} \Omega^0 = 0 \tag{D4a}$$

Then, the substitution of Equation (D3a) into Equation (D3d) leads to

$$\left\{ \left( D^2 - k^2 \right)^2 - Q D^2 \right\} W^0 = 0 \tag{D4b}$$

where the magneto-induction coefficient $Q$ is defined by

$$Q \equiv \frac{\sigma^* B_0^2}{\rho \nu} \tag{D4c}$$

Here, we introduce a representative length $d$. Then, let $a = kd$ be the wavenumber in a non-dimensional unit. We shall, however, let $x$, $y$, and $z$ stand for the non-dimensional coordinates normalized by $d$, so that the following parameter $Q$ and operator D are changed as follows.

$$Q^* \equiv \frac{\sigma^* B_0^2 d^2}{\rho \nu} \left( = Q d^2 \right) \tag{D5a}$$

$$D \equiv \frac{d}{dz} (= Dd) \tag{D5b}$$

where the coordinate $z$ is in the new unit of length $d$. Resultantly, Equations (D4a) and (D4b) are rewritten as

$$\left\{ \left( D^2 - a^2 \right)^2 - Q^* D^2 \right\} \Omega^0 = 0 \tag{D6a}$$

$$\left\{ \left( D^2 - a^2 \right)^2 - Q^* D^2 \right\} W^0 = 0 \tag{D6b}$$

As shown in Equations (D6a) and (D6b), $\Omega^0$ and $W^0$ are independent of each other. This means that the $z$-component of the vorticity does not interact with the $z$-component of velocity as they are.

### 2.3. Vortex Motions Induced in the Rotating Upper Layer

The upper layer is a reservoir of the vortices activated in the lower layer. In the lower layer, the activated vortices, due to the pinning effect of the downward vortices, keep their positions constant. On the contrary, in the upper layer, due to the rotation of the upper layer, the vortices induced by the vortices in the lower layer change their positions, revolving with the upper layer. At the same time, they start precessional motions. Then, through the upper boundary between the upper and lower layers, the motions conferred by the upper-layer rotation are transferred to the lower-layer vortices. All these processes, as shown in Figure 4a, form a positive feedback cycle.

Due to the low electric conductivity of an electrolytic solution, the electromagnetic induction by the upper layer vortices is neglected, so for simplicity, we only think of the effects of Coriolis force and centrifugal force. As shown in Figure 4b, let us consider an incompressible fluid of the upper layer rotating with an angular velocity $\vec{\Omega}$. In a frame of reference rotating with the same angular velocity, an observer at rest recognizes two kinds of acceleration (Appendix E) [65], i.e.,

$$\vec{F_R} = 2\vec{\Omega} \times \vec{U} - \frac{1}{2} \nabla \left( \left| \vec{\Omega} \times \vec{r} \right|^2 \right) \tag{E1}$$

where $\vec{\Omega}$ denotes the vector of the angular velocity, $\vec{U}$ is the vector of the velocity, and $\vec{r}$ is the position vector.

The term $2\,\overset{\rightarrow}{\Omega} \times \overset{\rightarrow}{U}$ represents the Coriolis acceleration and the term $-(1/2)\nabla\left(\left|\overset{\rightarrow}{\Omega} \times \overset{\rightarrow}{r}\right|^2\right)$ is the centrifugal force. The velocity is expressed by the main flow component of the rotation $U_i^*$ and the activated vortex flow $u_i$, i.e.,

$$U_i = U_i^* + u_i \tag{E4a}$$

However, since the observer is rotating with the upper layer, it follows that

$$U_i^* = 0 \tag{E4b}$$

The acceleration in Equation (E1) is fluctuated, expressed in the tensor notation.

$$f_{Ri} = 2\varepsilon_{ijk}U_j\Omega_k - \frac{1}{2}\frac{\partial}{\partial x_i}\left(\left|\overset{\rightarrow}{\Omega} \times \overset{\rightarrow}{r}\right|^2\right) \tag{E5}$$

where the first and second terms on the right-hand side of Equation (E5) denote the contributions of the Coriolis and centrifugal forces, respectively, where the second term is equal to zero without fluctuation.

Considering that a vector of rotation is an axial vector with $z$-axis, we can write down the following notation,

$$\overset{\rightarrow}{\Omega} \equiv \left(0, 0, \widetilde{\Omega}\right) \tag{E14}$$

where $\widetilde{\Omega}$ is the angular velocity of the upper layer. Therefore, we obtain the equations of the $z$-components of the vorticity $\omega_z$ and velocity $w$ for the vortices.

$$\frac{\partial \omega_z}{\partial t} = \nu\nabla^2\omega_z + 2\widetilde{\Omega}\frac{\partial u_z}{\partial z} \tag{E15}$$

and

$$\frac{\partial}{\partial t}\nabla^2 w = \nu\nabla^4 w - 2\widetilde{\Omega}\frac{\partial \omega_z}{\partial z} \tag{E16}$$

Substituting Equations (D1a) and (D1b) in Appendix D into Equations (E15) and (E16), and considering that the fluctuations are in a quasi-steady state, we disregard the time differential terms. Then, let $a = kd$ be the wavenumber in the non-dimensional. We shall, however, let $x$, $y$, and $z$ stand the coordinates in the new unit of length $d$. As a result, Equations (E15) and (E16) are changed to

$$\left(D^2 - a^2\right)\Omega^0 = -T^*DW^0 \tag{E20a}$$

and

$$\left(D^2 - a^2\right)^2 W^0 = d^2 T^* D\Omega^0 \tag{E20b}$$

where D is defined by the new coordinate $z$ as $d/dz$, and the rotation coefficient $T^*$ is expressed by

$$T^* \equiv \frac{2\widetilde{\Omega}d}{\nu} \tag{E20c}$$

At the boundary between the upper and lower layers, the lower-layer vortices will receive the precessional motions from the upper-layer vortices shown in Equations (E20a) and (E20b).

### 2.4. Boundary Conditions

#### 2.4.1. Hydrodynamic Conditions

The fluid in the lower layer is confined between the electrode and the upper layer. For convenience, the positions of the lower and upper boundaries are defined as 0 and 1 by the

scale of length $d$, which is equalized to the autocorrelation distance $a^+$ of the fluctuation, i.e., the average size of the vortices.

$$d = a^+ \tag{1}$$

Then, regardless of the nature of the boundary surface on the electrode, rigid or free, we must require

$$w = 0 \text{ for } z = 0 \tag{2}$$

We shall distinguish two kinds of boundary surfaces—the rigid surface on which no slip occurs and the free surface on which no tangential stress acts.

(a)  For the rigid surfaces:

Consider first the rigid surface. The condition that no slip occurs on the surface implies that $w$, as well as the horizontal components of the velocity, $u$ and $v$ vanish, i.e., $u = v = 0$. Since such a condition must be satisfied for all coordinates $x$ and $y$ on the surface, it follows from the continuity equation, Equation (B8) more explicitly, $\partial u / \partial x + \partial v / \partial y + \partial w / \partial z = 0$ that

$$\frac{\partial w}{\partial z} = 0 \text{ for } z = 0 \tag{3}$$

The condition of the normal component of the vorticity $\omega_z$ can also be deduced. More explicitly, $\omega_z$ is expressed by $\partial v / \partial x - \partial u / \partial y$, so that we have

$$\omega_z = 0 \quad \text{for } z = 0 \tag{4}$$

Substituting Equations (D1a) and (D1b) into Equations (2), (3) and (4), we obtain the following amplitude conditions.

$$W^0 = 0 \quad \text{for } z = 0 \tag{5a}$$

$$DW^0 = 0 \quad \text{for } z = 0 \tag{5b}$$

$$\Omega^0 = 0 \quad \text{for } z = 0 \tag{5c}$$

(b)  For the free surfaces:

The conditions on the free surface are that the stress tensors are zero, i.e.,

$$P_{xz} = P_{yz} = 0 \tag{6}$$

Since the isotropic term $-P\delta_{ij}$ has no transverse component, the condition Equation (6) is equivalent to the vanishing of the components $P_{xz}$ and $P_{yz}$ of the viscous stress tensor.

$$P_{xz} = \mu_s \left( \frac{\partial u}{\partial z} + \frac{\partial w}{\partial x} \right) \tag{7a}$$

and

$$P_{yz} = \mu_s \left( \frac{\partial v}{\partial z} + \frac{\partial w}{\partial y} \right) \tag{7b}$$

where $\mu_s$ implies the viscosity of the solution. As $w$ vanishes for all $x$- and $y$-coordinates on the boundary surface, it follows from Equations (6), (7a) and (7b) that

$$\frac{\partial u}{\partial z} = \frac{\partial v}{\partial z} = 0 \quad \text{for } z = 0 \tag{7c}$$

Substitution of Equation (7c) into the equation of continuity differentiated with respect to $z$, $\partial(\partial u / \partial x) / \partial z + \partial(\partial v / \partial y) / \partial z + \partial(\partial w / \partial z) / \partial z = 0$ leads to

$$\frac{\partial^2 w}{\partial z^2} = 0 \quad \text{for } z = 0 \tag{8}$$

Then, substituting Equation (7c) into the equation of $\omega_z$ (Equation (C10)) differentiated by $z$, $\partial\omega_z/\partial z = \partial(\partial v/\partial z)/\partial x - \partial(\partial u/\partial z)/\partial y$, we have

$$\frac{\partial\omega_z}{\partial z} = 0 \quad \text{for} \quad z = 0 \tag{9}$$

From Equations (2), (8) and (9), we have

$$W^0 = 0 \quad \text{for} \quad z = 0 \tag{10a}$$

$$\mathrm{D}^2 W^0 = 0 \quad \text{for} \quad z = 0 \tag{10b}$$

$$\mathrm{D}\Omega^0 = 0 \quad \text{for} \quad z = 0 \tag{10c}$$

(c)    For the upper boundary between the lower and upper layers:

Since the upper and lower layers are hydrodynamically connected, $w$ and $\omega_z$ do not vanish at the upper boundary, and there is no slip there. The boundary conditions at the upper boundary are given by Equations (E20a) and (E20b). In the lower layer, Equations (D6a) and (D6b) are fulfilled. Therefore, from Equation (E20a), we have the upper boundary condition,

$$\left(\mathrm{D}^2 - a^2\right)\Omega^0 = -T^*\mathrm{D}W^0 \quad \text{for} \quad z = 1 \tag{11a}$$

and inserting Equation (D6b) into Equation (E20b), we obtain the simpler condition.

$$Q^*\mathrm{D}^2 W^0 = d^2 T^*\mathrm{D}\Omega^0 \quad \text{for} \quad z = 1 \tag{11b}$$

2.4.2. Mass Transfer Conditions

In addition to the hydrodynamic conditions, we can also write down the mass transfer conditions. From Fick's first law, we obtain the following relationship between the current density fluctuation $j_z$ and the concentration fluctuation $c_m$.

$$j_z = -z_m F D_m \left(\frac{\partial c_m}{\partial z}\right)_{z=0} \tag{12}$$

where $z_m$ and $D_m$ are the charge number of the metallic ion and the diffusion coefficient, respectively. At the upper boundary, it is assumed that the concentration fluctuation vanishes.

$$c_m \to 0 \quad \text{for} \quad z \to 1 \tag{13}$$

As a result, using the amplitudes $J^0$ and $\Theta^0$, we have

$$J^0 = -z_m F D_m \mathrm{D}\Theta^0 \quad \text{for} \quad z = 0 \tag{14a}$$

and

$$\Theta^0 \to 0 \quad \text{for} \quad z \to 1 \tag{14b}$$

The conditions of Equations (14a) and (14b) suggest that two arbitrary constants in the solution of the concentration fluctuation are required.

*2.5. Solutions of $W^0$ and $\Omega^0$ in the Lower Layer*

From Appendix I, the general equation of the amplitude of the $z$-component of the velocity $W^0$ is provided by

$$W^0(z,t) = (\alpha_0 + \alpha_1 z)e^{az} + (\alpha_2 + \alpha_3 z)e^{-az} \tag{I12}$$

where $\alpha_0$, $\alpha_1$, $\alpha_2$, and $\alpha_3$ are arbitrary constants. Then, the first and second derivatives are derived as follows:

$$\mathrm{D}W^0(z,t) = \{\alpha_0 a + \alpha_1(1 + az)\}e^{az} + \{-\alpha_2 a + \alpha_3(1 - az)\}e^{-az} \tag{I14a}$$

and

$$D^2 W^0(z, t) = \left\{ \alpha_0 a^2 + \alpha_1(2 + az)a \right\} e^{az} + \left\{ \alpha_2 a^2 + \alpha_3(-2 + az)a \right\} e^{-az} \tag{I14b}$$

The vorticity in the lower layer is affected by the precessional motions in the upper layer at the upper boundary. In view of the boundary conditions in Equations (11a) and (11b) at the upper boundary, two arbitrary constants are necessary. This means that the vorticity depends only on $e^{az}$, so that $\Omega^0$ is expressed by

$$\Omega^0(z, t) = (\beta_0 + \beta_1 z)e^{az} \tag{I13}$$

where $\beta_0$ and $\beta_1$ are arbitrary constants. The first and second derivatives are

$$D\Omega^0(z, t) = \{\beta_0 a + \beta_1(1 + az)\}e^{az} \tag{I15a}$$

and

$$D^2\Omega^0(z, t) = a\{\beta_0 a + \beta_1(2 + az)\}e^{az} \tag{I15b}$$

The individual rigid and free surface components of $W^0$ and $\Omega^0$ are determined in the following.

(a) For the rigid surface vortices:

Substituting Equations (I12), (I13) and (I14a) into Equations (5a)–(5c), we obtain

$$\alpha_2 = -\alpha_0 \tag{15a}$$

$$\alpha_3 = -(2\alpha_0 a + \alpha_1) \tag{15b}$$

$$\beta_0 = 0 \tag{15c}$$

Inserting Equations (15a)–(15c) in Equations (I12) and (I13), we obtain the expressions for the rigid surface vortices,

$$W_r^0(z, t) = 2(\alpha_0 + \alpha_1 z)\sin h az - 2\alpha_0 aze^{-az} \tag{16a}$$

and

$$\Omega_r^0(z, t) = \beta_1 ze^{az} \tag{16b}$$

where the subscript 'r' implies the rigid surface component. $\beta_1$ denotes the vorticity coefficient of the rigid surface vortices. Using the upper boundary conditions in Equations (11a) and (11b), we will determine the velocity coefficients $\alpha_0$ and $\alpha_1$ of the rigid surface vortices as the functions of $\beta_1$.

(b) For the free surface vortices:

Substituting Equations (I12), (I14b) and (I15a) into Equations (10a)–(10c), we obtain

$$\alpha_2 = -\alpha_0 \tag{17a}$$

$$\alpha_3 = \alpha_1 \tag{17b}$$

$$\beta_1 = -\beta_0 a \tag{17c}$$

Inserting Equations (17a)–(17c) in Equations (I12) and (I13), we obtain the expressions for the free surface vortices,

$$W_f^0(z, t) = 2\alpha_0 \sin h az + 2\alpha_1 z \cosh az \tag{18a}$$

and

$$\Omega_f^0(z, t) = \beta_0(1 - az)e^{az} \tag{18b}$$

where the subscript 'f' implies the free surface component. $\beta_0$ denotes the vorticity coefficient of the free surface vortices. Using the upper boundary conditions in Equations (11a) and (11b), we will determine the velocity coefficients $\alpha_0$ and $\alpha_1$ of the free surface vortices as the functions of $\beta_0$.

*2.6. Determination of the Velocity Coefficients $\alpha_0$ and $\alpha_1$*

(a)  For the rigid and free surface vortices at the upper boundary:

At the upper boundary, we have two relationships. First, by substituting Equations (I13), (I14a) and (I15b) into Equation (11a), we have

$$\{\alpha_0 a + \alpha_1(1+a)\}e^a + \{-\alpha_2 a + \alpha_3(1-a)\}e^{-a} = -2T^{*-1}\beta_1 a e^a \tag{19a}$$

Then, the substitution of Equations (I14b) and (I15a) into Equation (11b) leads to

$$\left\{\alpha_0 a^2 + \alpha_1\left(2a+a^2\right)\right\}e^a + \left\{\alpha_2 a^2 + \alpha_3\left(-2a+a^2\right)\right\}e^{-a} = 2Q^{*-1}d^2 T^*\{\beta_0 a + \beta_1(1+a)\}e^a \tag{19b}$$

(b)  For the rigid surface vortices in the lower layer:

Substituting Equations (15a)–(15c) into Equations (19a) and (19b), we have

$$\alpha_0 a\left(\sinh a + a e^{-a}\right) + \alpha_1(\sinh a + a\cosh a) = -T^{*-1}\beta_1 a e^a \tag{20a}$$

and

$$\alpha_0 a\left\{\cosh a + (1-a)e^{-a}\right\} + \alpha_1(2\cosh a + a\sinh a) = -(2Q^* a)^{-1}d^2 T^*\beta_1(1+a)e^a \tag{20b}$$

Equations (20a) and (20b) form simultaneous equations with respect to $\alpha_0$ and $\alpha_1$, so that the following solutions are derived.

$$\alpha_0 = \beta_1 \alpha_{0r}^*(a) \tag{21a}$$

where $\alpha_{0r}^*(a)$ is given by

$$\alpha_{0r}^*(a) = -\frac{e^a\left\{2Q^* a^2(2\cosh a + a\sinh a) + d^2 T^{*2}(1+a)(\sinh a + a\cosh a)\right\}}{2Q^* T^* a^2\left(\sinh^2 a + a^2\right)} \tag{21b}$$

and

$$\alpha_1 = \beta_1 \alpha_{1r}^*(a) \tag{22a}$$

where $\alpha_{1r}^*(a)$ is expressed by

$$\alpha_{1r}^*(a) \equiv \frac{e^a\left[2Q^* a^2\left\{\cosh a + (1-a)e^{-a}\right\} + d^2 T^{*2}(1+a)(\sinh a + a e^{-a})\right]}{2Q^* T^* a\left(\sinh^2 a + a^2\right)} \tag{22b}$$

Here, we use the following formula.

$$\cosh^2 a - \sinh^2 a = 1 \tag{23}$$

(c)  For the free surface vortices in the lower layer:

Substituting Equations (17a)–(17c) into Equations (19a) and (19b), we have

$$\alpha_0 a\cosh a + \alpha_1(\cosh a + a\sinh a) = T^{*-1}\beta_0 a^2 e^a \tag{24a}$$

and

$$\alpha_0 a\sinh a + \alpha_1(2\sinh a + a\cosh a) = -(2Q^*)^{-1}d^2 T^*\beta_0 a e^a \tag{24b}$$

Equations (24a) and (24b) form simultaneous equations with respect to $\alpha_0$ and $\alpha_1$, so that the following solutions are derived:

$$\alpha_0 = \beta_0 \alpha_{0f}^*(a) \tag{25a}$$

where $\alpha_{0f}^*(a)$ is defined by

$$\alpha_{0f}^*(a) \equiv \frac{e^a \left\{ 2Q^* a (2\sinh a + a \cosh a) + d^2 T^{*2} (\cosh a + a \sinh a) \right\}}{2Q^* T^* (\sinh a \cosh a + a)} \tag{25b}$$

and

$$\alpha_1 = \beta_0 \alpha_{1f}^*(a) \tag{26a}$$

where $\alpha_{1f}^*(a)$ is written by

$$\alpha_{1f}^*(a) = -\frac{a e^a \left( 2Q^* a \sinh a + d^2 T^{*2} \cosh a \right)}{2Q^* T^* (\sinh a \cosh a + a)} \tag{26b}$$

### 2.7. The Solution of $\Theta^0$ and $D\Theta^0$ at the Electrode Surface

As shown in Appendix J, using the solution of $W^0$ in Equation (I12), we solve Equation (D3e). The general expression of $\Theta^0$ is given by

$$\begin{aligned} \Theta^0(z,t) = A_1 e^{-az} &+ \frac{R^*}{8a^3} \left\{ -2\alpha_0 a(-2az + 1) + \alpha_1 (2a^2 z^2 - 2az + 1) \right\} e^{az} \\ &+ \left\{ -2\alpha_2 a(2az + 1) - \alpha_3 (2a^2 z^2 + 2az + 1) \right\} e^{-az} \end{aligned} \tag{J6}$$

We also have the first derivative with respect to $z$

$$\begin{aligned} D\Theta^0(z,t) = \quad &-aA_1 e^{-az} \\ &+ \frac{R^*}{8a^2} \left[ \left\{ 2\alpha_0 a(2az + 1) + \alpha_1 (2a^2 z^2 + 2az - 1) \right\} e^{az} \right. \\ &\left. + \left\{ 2\alpha_2 a(2az - 1) + \alpha_3 (2a^2 z^2 - 2az - 1) \right\} e^{-az} \right] \end{aligned} \tag{J7a}$$

where the mass transfer coefficient $R^*$ is defined by

$$R^* \equiv \frac{L_m d^2}{D_m} \tag{J2b}$$

Therefore, at the electrode surface, $z = 0$, we have

$$D\Theta^0(0,t) = -aA_1 + \frac{R^*}{8a^2} (2\alpha_0 a - \alpha_1 - 2\alpha_2 a - \alpha_3) \tag{J7b}$$

and

$$\Theta^0(0,t) = A_1 + \frac{R^*}{8a^3} (-2\alpha_0 a + \alpha_1 - 2\alpha_2 a - \alpha_3) \tag{J7c}$$

where $A_1$ denotes an arbitrary constant. Using the amplitude equations of the current density, vorticity, and mass-flux fluctuations in Equations (D3b), (D3c) and (14a), respectively, we finally obtain the following equations, removing the arbitrary constant $A_1$.

$$D\Theta^0(0,t) = \frac{2\beta_1}{z_m F D_m S^*} \tag{J14}$$

and

$$\Theta^0(0,t) = -\frac{2\beta_1}{z_m F D_m S^* a} - \frac{R^*}{4a^3} (2\alpha_2 a + \alpha_3) \tag{J16}$$

where the magneto-viscosity coefficient $S^*$ is defined by

$$S^* \equiv \frac{B_0 d}{\rho \nu} \tag{J9b}$$

In Equations (J14) and (J16), the coefficients $\alpha_2$, $\alpha_3$, and $\beta_1$ are undecided, so using the conditions of the rigid and free surface vortices, we determine them in the following.

a.     For the rigid surface vortices:

Substituting Equations (15a)–(15c) into Equations (J14) and (J16), we have

$$D\Theta_r^0(0,t) = \frac{2\beta_1}{z_m F D_m S^*} \tag{27a}$$

and

$$\Theta_r^0(0,t) = -\frac{2\beta_1}{z_m F D_m S^* a} + \frac{R^*}{4a^3}(4\alpha_0 a + \alpha_1) \tag{27b}$$

Then, substitution of Equations (21a), (21b), (22a) and (22b) into Equation (27b) leads to

$$\Theta_r^0(0,t) = -\frac{2\beta_1}{z_m F D_m S^* a} \\ -\frac{\beta_1 R^* e^a \left[ 2Q^* a^2 \{5 \cosh a + 2a \sinh a + (1+a)e^a\} + d^2 T^{*2}(1+a)(3 \sinh a + 2a \cosh a + ae^a) \right]}{8Q^* T^* a^4 (\sinh^2 a + a^2)} \tag{27c}$$

The residual undetermined parameter, i.e., the vorticity coefficient of the rigid surface vortex $\beta_1$ in Equations (27a)–(27c) will be determined by examining the actual formulation of 2D or 3D nucleation.

(b)     For the free surface vortices:

Substituting Equations (17a)–(17c) into Equations (J14) and (J16), we have

$$D\Theta_f^0(0,t) = -\frac{2\beta_0 a}{z_m F D_m S^*} \tag{28a}$$

and

$$\Theta_f^0(0,t) = -\frac{2\beta_0}{z_m F D_m S^*} - \frac{R^*}{4a^3}(2\alpha_0 a - \alpha_1) \tag{28b}$$

Then, substitution of Equations (25a), (25b), (26a) and (26b) into Equation (28b) leads to

$$\Theta_f^0(0,t) = \frac{2\beta_0}{z_m F D_m S^*} + \frac{\beta_0 R^* e^a \{2Q^* a(5 \sinh a + 2a \cosh a) + d^2 T^{*2}(3 \cosh a + 2a \sinh a)\}}{8Q^* T^* a^2 (\sinh a \cosh a + a)} \tag{28c}$$

The residual undetermined parameter, i.e., the vorticity coefficient of the free surface vortex $\beta_0$ in Equations (28a)–(28c) will be determined by the actual formulation of 2D or 3D nucleation.

## 3. 2D Nucleation

### 3.1. Asymmetrical Fluctuations in 2D Nucleation Process

The 2D nucleation proceeds in an electric double layer. As shown in Figure 5, at the inner Helmholtz plane (IHP), dehydrated metallic ions receive electrons, being adsorbed as adatoms on the electrode surface [22].

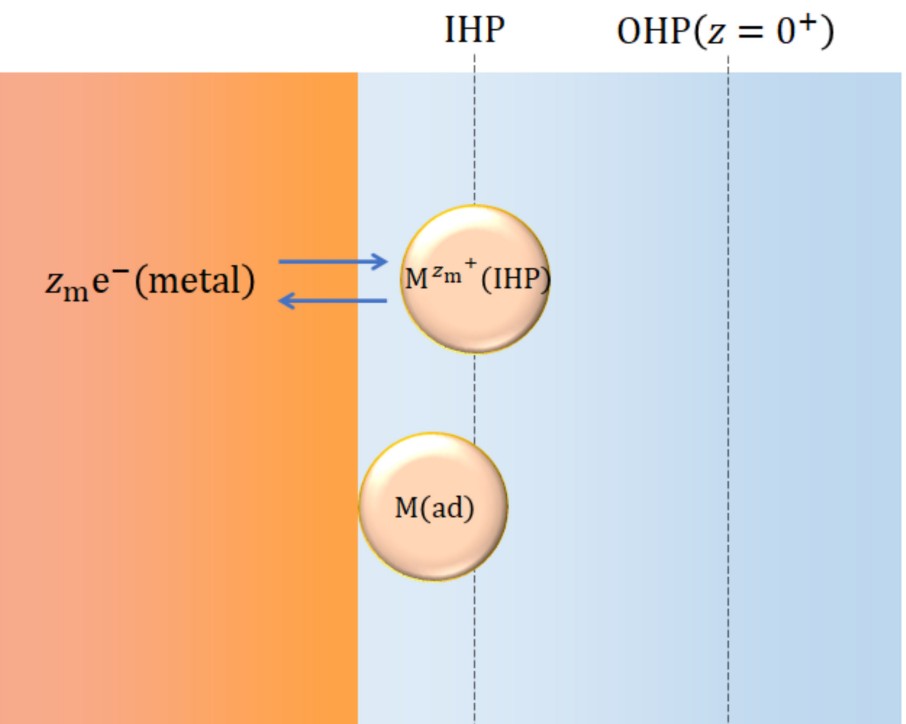

**Figure 5.** The 2D nucleation in an electric double layer.

Assuming equilibrium between the metallic ions at the IHP and the adatoms at the metal surface, the electron-transfer reaction is written as

$$M(ad) \rightleftharpoons M^{z_m^+}(IHP) + z_m e^-(metal) \tag{29}$$

where $M(ad)$ and $M^{z_m^+}(IHP)$ are the adatom and the metallic ion at the IHP, respectively, and $e^-(metal)$ is the free electron at the electrode.

According to Equation (29), their chemical and electrochemical potentials are related with

$$\mu_{ad}(x, y, t) = \overline{\mu_m}(x, y, \zeta^a, t) + z_m \overline{\mu_e}(x, y, t) \tag{30}$$

where $\mu_{ad}(x, y, t)$ implies the chemical potential of the adatom. $\zeta^a \equiv \zeta(x, y, t)^a$ denotes the surface deformation by 2D nucleation formed by asymmetrical fluctuations, $\overline{\mu_m}(x, y, \zeta^a, t)$ is the electrochemical potential of the metallic ion, and $\overline{\mu_e}(x, y, t)$ is the electrochemical potential of a free electron.

At the equilibrium potential, physical quantities fluctuate toward the positive and negative sides of their equilibrium states (Figure 6a). However, when the potential is deviated from the equilibrium to the cathodic direction, the cathodic reaction proceeds, and various asymmetrical fluctuations, including 2D nucleation, develop around the electrode. As shown in Figure 6b, they one-sidedly fluctuate from their equilibrium states, i.e., whether plus or minus, their signs are kept constant. This means that in a nonequilibrium state, either side of the amplitude of an equilibrium fluctuation is cut off (phase cutting).

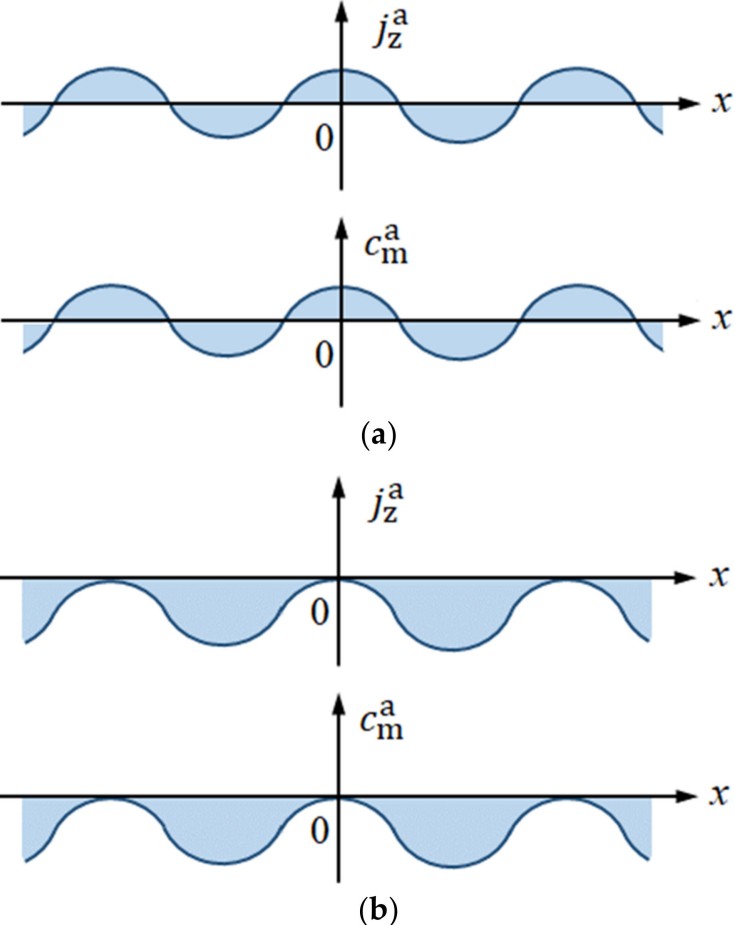

**Figure 6.** Equilibrium and nonequilibrium fluctuations. (**a**) Equilibrium concentration fluctuation. (**b**) Nonequilibrium concentration fluctuation in a cathodic deposition.

Due to large mobility in light speed, the fluctuation of the electron can be neglected in steady-state, so from Equation (30), the relationship between the fluctuations is expressed by

$$\delta\overline{\mu_m}(x, y, \zeta^a, t)^a = \delta\mu_{ad}(x, y, t)^a \tag{31}$$

where superscript 'a' implies asymmetrical fluctuation. Due to the small curvature of a 2D nucleus, the chemical potential fluctuation of the adatom, which arises from the change of the surface form of the deposit can be neglected $\delta\mu_{ad}(x, y, t)^a = 0$, so that we have the following condition of the electrochemical potential fluctuation of the metallic ion.

$$\delta\overline{\mu_m}(x, y, \zeta^a, t)^a = 0 \tag{32}$$

Accompanied by electrolytic current flowing, asymmetrical potential and concentration fluctuations occur. Based on Fick's first law, at the outer Helmholtz plane (OHP), $z = 0^+$, the current density fluctuation is written by

$$j_z(x, y, 0, t)^a = -z_m F D_m \left\{ \frac{\partial c_m(x, y, z, t)^a}{\partial z} \right\}_{z=0^+} \tag{33a}$$

In the presence of a large amount of supporting electrolytes, from Equation (B9), the current density fluctuation is also described by the potential fluctuation as follows:

$$j_z(x, y, 0, t)^a = -\sigma^* \left\{ \frac{\partial \phi_2(x, y, z, t)^a}{\partial z} \right\}_{z=0^+} \tag{33b}$$

where $\phi_2(x, y, z, t)^a$ implies the asymmetrical fluctuation of the overpotential $\Phi_2$ in the diffuse layer. From Equations (33a) and (33b), we obtain

$$\sigma^* \left\{ \frac{\partial \phi_2(x, y, z, t)^a}{\partial z} \right\}_{z=0^+} = z_m F D_m \left\{ \frac{\partial c_m(x, y, z, t)^a}{\partial z} \right\}_{z=0^+} \tag{33c}$$

Since both fluctuations have the same function form of $\exp(-az)$, Equation (33c) supplies the equation,

$$\sigma^* \phi_2(x, y, 0^+, t)^a = z_m F D_m c_m(x, y, 0^+, t)^a \tag{33d}$$

At the same time, the electrochemical potential fluctuation $\delta \bar{\mu}_m(x, y, \zeta^a, t)^a$ is also represented by these fluctuations,

$$\delta \bar{\mu}_m(x, y, \zeta^a, t)^a = z_m F \left\{ \phi_1(x, y, t)^a + \phi_2(x, y, \zeta^a, t)^a \right\} + \frac{RT}{C_m^*(z=0)} c_m(x, y, \zeta^a, t)^a \tag{34}$$

where $\phi_1(x, y, t)^a$ and $\phi_2(x, y, \zeta^a, t)^a$ are the overpotential fluctuations at the inner Helmholtz plane (IHP) and at the surface of a 2D nucleus in the diffuse layer, respectively. $C_m^*(z=0)$ is the surface concentration outside the double layer, $R$ is the universal gas constant ($8.31 \text{ J K}^{-1} \text{ mol}^{-1}$), $T$ is the absolute temperature (K), and $F$ is the Faraday constant ($96,500 \text{ C mol}^{-1}$).

By expanding with respect to the $z$-coordinate at the flat OHP without the 2D nuclei, $z = 0^+$, the potential fluctuation at the surface of a 2D nucleus $z = \zeta^a$ is expressed by

$$\phi_2(x, y, \zeta^a, t)^a = \phi_2(x, y, 0^+, t)^a + L_{\phi_2} \zeta(x, y, t)^a \tag{35a}$$

$L_{\phi_2}$ is the gradient of the electrostatic overpotential in the diffuse layer defined by [22]

$$L_{\phi_2} \equiv -\frac{\Phi_{2OHP}^*}{\lambda} \tag{35b}$$

where $\Phi_{2OHP}^*$ is the electrostatic overpotential at a flat OHP without 2D nuclei $z = 0^+$ measured from the outer boundary of the diffuse layer ($z = \infty^+$), and $\lambda$ is the Debye length equalized to the average diffuse layer thickness.

$$\lambda \equiv \left( \frac{\varepsilon RT}{F^2 \sum_{j \neq m} z_j^2 C_j^*(z=\infty)} \right)^{\frac{1}{2}} \tag{35c}$$

where $\varepsilon$ is the dielectric constant of water ($6.95 \times 10^{-10} \text{ J}^{-1} \text{ C}^2 \text{ m}^{-1}$, 25 °C), $z_j$ is the charge number including sign, and $C_j^*(z=\infty)$ is the bulk concentration of ionic species j except for the bulk metallic-ion concentration $C_m^*(z=\infty)$ (mol m$^{-3}$).

The concentration fluctuation in the diffuse layer is correspondingly expressed by

$$c_m(x, y, \zeta^a, t)^a = c_m(x, y, 0^+, t)^a + L_{m2} \zeta(x, y, t)^a \tag{36a}$$

where $L_{m2}$ is the average concentration gradient of the metallic ion in the diffuse layer [66], which is defined by

$$L_{m2} \equiv -\frac{z_m F C_m^*(z=0)}{\lambda RT} \Phi_{2OHP}^* \tag{36b}$$

Equations (35b) and (36b) have the relationship,

$$z_m F L_{\phi_2} = -\frac{RT}{C_m^*(z=0)} L_{m2} \tag{37}$$

Here, the overpotential fluctuation of the Helmholtz layer $\phi_1(x,y,t)^a$ is induced by the fluctuation $\phi_2(x,y,\zeta^a,t)^a$, which is depicted by the differential potential coefficient $(\partial\langle\Phi_1\rangle/\langle\Phi_2\rangle)_\mu$ in Equation (A1) in Appendix A.

$$\phi_1(x,y,t)^a = \left(\frac{\partial\langle\Phi_1\rangle}{\partial\langle\Phi_2\rangle}\right)_\mu \phi_2(x,y,\zeta^a,t)^a \tag{38}$$

Substituting for $\phi_1(x,y,t)^a$ from Equation (38) in Equation (34), then inserting Equations (36a) and (37) into the resulting equation, we obtain

$$\begin{aligned}
\delta\overline{\mu}_m(x,y,\zeta^a,t)^a &= z_m F\left\{\left(\frac{\partial\langle\Phi_1\rangle}{\partial\langle\Phi_2\rangle}\right)_\mu + 1\right\}\phi_2(x,y,0^+,t)^a + \frac{RT}{C_m^*(z=0)}c_m(x,y,0^+,t)^a \\
&\quad + z_m F\left(\frac{\partial\langle\Phi_1\rangle}{\partial\langle\Phi_2\rangle}\right)_\mu L_{\phi_2}\zeta(x,y,t)^a
\end{aligned} \tag{39a}$$

Under the limiting diffusion condition, the surface concentration $C_m^*(z=0)$ is sufficiently small in limiting diffusion, so that substituting for $\phi_2(x,y,0^+,t)^a$ from Equation (33d) in Equation (39a), we can derive the following condition:

$$(z_m F)^2 D_m\left\{\left(\frac{\partial\langle\Phi_1\rangle}{\partial\langle\Phi_2\rangle}\right)_\mu + 1\right\} \ll \frac{\sigma^* RT}{C_m^*(z=0)} \tag{39b}$$

Using Equations (35b) and (39b), we obtain the electrochemical potential fluctuation at the top of a nucleus, which is represented by

$$\delta\overline{\mu}_m(x,y,\zeta^a,t)^a = \frac{RT}{C_m^*(z=0)}c_m(x,y,0^+,t)^a - \frac{z_m F}{\lambda}\left(\frac{\partial\langle\Phi_1\rangle}{\partial\langle\Phi_2\rangle}\right)_\mu \Phi_{2OHP}^*\zeta(x,y,t)^a \tag{39c}$$

Applying the electrochemical condition Equation (32) to Equation (39c), we have the relationship between $\zeta(x,y,t)^a$ and $c_m(x,y,0^+,t)^a$.

$$c_m(x,y,0^+,t)^a = \frac{z_m F}{\lambda RT}\left(\frac{\partial\langle\Phi_1\rangle}{\partial\langle\Phi_2\rangle}\right)_\mu \Phi_{2OHP}^* C_m^*(z=0)\zeta(x,y,t)^a \tag{40}$$

Due to small wavenumbers of the fluctuations arising from large 2D nuclei, a higher order of smallness, such as surface energy, can be disregarded [22]. Considering that the surface deformation results from the mass transfer of metallic ions, we obtain

$$\frac{\partial}{\partial t}\zeta(x,y,t)^a = \Omega_m D_m\left\{\frac{\partial}{\partial z}c_m(x,y,z,t)^a\right\}_{z=0} \tag{41}$$

where $\Omega_m$ represents the molar volume of the deposited metal (m$^3$ mol$^{-1}$).

Here, in the scale of unit length $d^a$ of the asymmetrical fluctuations, the coordinate of the electrode surface $z=0$ is equalized to the coordinate of the OHP $z=0^+$, so that $c_m(x,y,0^+,t)^a$ at the OHP is regarded as $c_m(x,y,0,t)^a$ at the electrode surface. Therefore, the substitution of Equation (40) into Equation (41) leads to

$$\frac{\partial}{\partial t}c_m(x,y,0,t)^a = A_\theta\left\{\frac{\partial}{\partial z}c_m(x,y,z,t)^a\right\}_{z=0} \tag{42}$$

Equation (42) is transformed in terms of Fourier transform regarding $x$- and $y$-coordinates, i.e.,

$$\frac{\partial}{\partial t}\Theta^0(0,t)^a = A_\theta D\Theta^0(0,t)^a \tag{43a}$$

where $D\Theta^0(0,t)^a$ implies $\left\{\partial\Theta^0(z,t)^a/\partial z\right\}_{z=0}$, and $A_\theta$ is the adsorption coefficient.

$$A_\theta \equiv \left(\frac{z_m F}{RT}\right)\left(\frac{\partial\langle\Phi_1\rangle}{\partial\langle\Phi_2\rangle}\right)_\mu \frac{D_m \Omega_m C_m^*(z=0)}{\lambda}\Phi_{2\text{OHP}}^* \tag{43b}$$

Equation (43a) controls the 2D nucleation in an electric double layer, i.e., when the amplitude unstably develops with time, 2D nuclei can deterministically grow with micro-MHD flows, yielding chiral depositions.

*3.2. Characteristic Equations of the Vorticity Coefficients $\beta_0^a$ and $\beta_1^a$*

As will be shown below, the characteristic equations of the micro-MHD flows on the rigid and free surfaces (i.e., rigid and free surface vortices) are derived as the equations of the vorticity coefficients $\beta_0^a$ and $\beta_1^a$, which are solved under the condition,

$$B_0 \widetilde{\Omega} > 0 \tag{44}$$

Here, for 2D nucleation, $\widetilde{\Omega}$ corresponds to the angular velocity of a VMHDF. As shown in the preceding papers [5,6], according to Equation (44), the sign of $\widetilde{\Omega}$, i.e., the rotational direction of a VMHDF is determined by the sign of $B_0$, i.e., the direction of the magnetic field.

(a)  For the rigid surface vortices:

Substituting $D\Theta_r^0(0,t)$ from Equation (27a) and $\Theta_r^0(0,t)$ from Equation (27c) in Equation (43a), we obtain the following characteristic equation of the vorticity coefficient $\beta_1^a$ for the rigid surface vortices in 2D nucleation.

$$\frac{d\beta_1^a}{dt} = -A_\theta f_r^a(a)\beta_1^a \tag{45a}$$

where $f_r^a(a)$ implies the amplitude factor function of the rigid surface vortices in 2D nucleation.

$$f_r^a(a) = \frac{16Q^{*a}g_4(a)}{16Q^{*a}g_5(a) + S^{*a}T^{*a-1}R^{*a}g_6(a)} \tag{45b}$$

where, since Equation (44) is always fulfilled, $f_r^a(a)$ has no singular point and takes positive values.

As shown in Appendix G, the coefficients $R^*$, $Q^*$, $T^*$, and $S^*$ are redefined in accordance with 2D nucleation as follows.

$$R^{*a} \equiv \frac{L_m d^{a2}}{D_m} \tag{G5b}$$

$$Q^{*a} \equiv \frac{\sigma^* B_0^2 d^{a2}}{\rho\nu^a} \tag{G5c}$$

$$T^{*a} \equiv \frac{2\,\Omega\,\tilde{}\,d^a}{\nu^a} \tag{G5d}$$

$$S^{*a} \equiv \frac{B_0 d^a}{\rho\nu^a} \tag{G5e}$$

where $d^a$ and $\nu^a$ are the representative lengths in 2D nucleation and the kinematic viscosity of the bulk solution, respectively. Then, $g_4(a)$, $g_5(a)$, and $g_6(a)$ are defined by

$$g_4(a) \equiv a^4\left(\sin h^2 a + a^2\right) \tag{45c}$$

$$g_5(a) \equiv a^3\left(\sin h^2 a + a^2\right) \tag{45d}$$

$$g_6(a) \equiv z_m F D_m e^a \left[ 2Q^{*a}a^2 \{5\cos h\, a + 2a \sin h\, a + (1+a)e^a\} + d^{a2}T^{*a2}(1+a)(3\sin h\, a + 2a \cos h\, a + ae^a) \right] \quad (45e)$$

where, for the deposition, $Q^{*a} > 0$ and $R^{*a} > 0$ are always fulfilled, and from Equation (44), $S^{*a}T^{*a-1} > 0$ is also obtained. Equation (45a) is solved as

$$\beta_1^a(t) = \beta_1^a(0)\exp(p_r^a t) \quad (46a)$$

where $p_r^a$ implies the amplitude factor of the rigid surface vortices.

$$p_r^a \equiv -A_\theta f_r^a(a) \quad (46b)$$

Therefore, the vortex motions develop or diminish with time in accordance with the sign of the amplitude factor $p_r^a$.

(b)  For the free surface vortices:

Substituting $D\Theta_f^0(0, t)$ from Equation (28a) and $\Theta_f^0(0, t)$ from Equation (28c) in Equation (43a), we obtain the following characteristic equation of the vorticity coefficient $\beta_0^a$ for the free surface vortices:

$$\frac{d\beta_0^a}{dt} = -A_\theta f_f^a(a)\beta_0^a \quad (47a)$$

where $f_f^a(a)$ implies the amplitude factor function of the free surface vortices in 2D nucleation.

$$f_f^a(a) = \frac{16Q^{*a}g_1(a)}{16Q^{*a}g_2(a) + S^{*a}T^{*a-1}R^{*a}g_3(a)} \quad (47b)$$

and

$$g_1(a) \equiv a^3(\sin h\, a \cos h\, a + a) \quad (47c)$$

$$g_2(a) \equiv a^2(\sin h\, a \cos h a + a) \quad (47d)$$

$$g_3(a) \equiv z_m F D_m e^a \left\{ 2Q^{*a}a(5\sin h\, a + 2a\cos h\, a) + d^{a2}T^{*a2}(3\cos h\, a + 2a\sin h\, a) \right\} \quad (47e)$$

where, since Equation (44) is always fulfilled, $f_f^a(a)$ has no singular point and takes positive values. The characteristic equation, Equation (47a), can always be solved, i.e.,

$$\beta_0^a(t) = \beta_0^a(0)\exp(p_f^a t) \quad (48a)$$

where $p_f^a$ means the amplitude factor of the free surface vortices.

$$p_f^a \equiv -A_\theta f_f^a(a) \quad (48b)$$

From Equations (46a) and (48a), we can determine whether the activated vortices are stable or unstable. If the amplitude factors $p_r^a$ and $p_f^a$ are negative for all wavenumbers, the fluctuations, as well as the vortices, are stable, stochastically repeating activation and extinction. On the contrary, when they are positive for some of the wavenumbers, the corresponding fluctuation components once activated become unstable, deterministically developing with time.

In Figure 7, the representative function forms of $f_r^a(a)$ and $f_f^a(a)$ against the non-dimensional wavenumber $a$ are exhibited. As discussed above, they are always positive for $a$, and $f_f^a(a)$ is larger than $f_r^a(a)$. As will be discussed later, when the adsorption coefficient $A_\theta$ is negative, i.e., in non-specific adsorption of ions, the free surface vortices receive the precessions since the free surface vortices grow faster than the rigid surface vortices. On the other hand, in the case of specific adsorption, due to positive $A_\theta$, the rigid surface vortices dwindle more slowly than the free surface ones, so that the rigid surface vortices will receive the precessions.

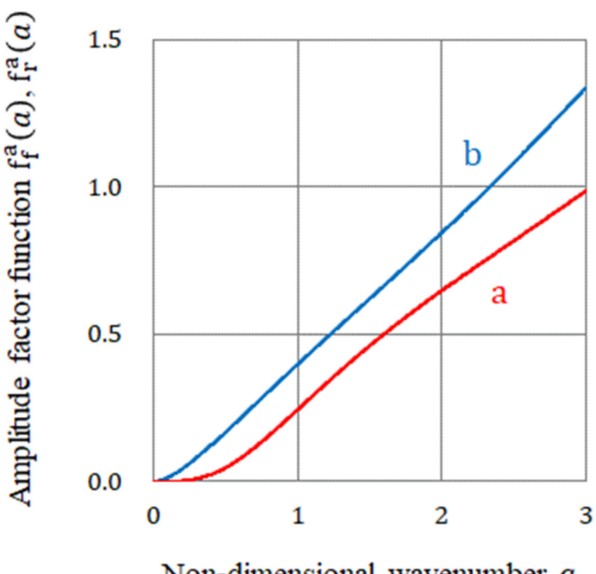

**Figure 7.** Amplitude factor functions vs. non-dimensional wavenumber *a*. a, $f_r^a(a)$ of the fluctuations on the rigid surfaces. b, $f_f^a(a)$ of the fluctuations on the free surfaces.

However, under some other conditions, if the difference of both functions became smaller, slight changes in the initial situation would easily lead to the inversion of the vortices receiving the precessions, i.e., breaking of chiral symmetry.

### 3.3. Nucleation by the Rigid and Free Surface Vortices

To calculate the nucleation with the rotations of the rigid and free surface vortices, we must solve the self-organization problem concerning two independent fluctuation components at once. This means that we must treat a two-components vector so that it is convenient to embed the amplitudes of both components in a single complex number.

We first suppose a horizontal wavenumber plane $(a_x, a_y)$ divided by 2D grids, where in the vertical $z$-direction, a solution phase $(z > 0)$ and an electrode phase $(z \leq 0)$ are defined. Then, considering a stochastic process in multi-nucleation of electrodeposition, we introduce a unit random complex number $R_d^a$ in the following: In the vertical $z$-direction within the autocorrelation distance of the fluctuation $a^+ (= d^a)$, i.e., in the lower layer $(0 \leq z \leq 1)$, the same random numbers are used, so that we define the following 2D random number.

$$R_d^a = \cos \theta_{\text{rand}}^a + i \cdot \sin \theta_{\text{rand}}^a \tag{49}$$

where $\theta_{\text{rand}}^a$ is a uniform random number between 0 and $2\pi$, which is assigned to all grid points defined on the $a_x - a_y$ plane. As a result, the actual values of $\beta_1^a(0)$ and $\beta_0^a(0)$ in Equations (46a) and (48a) are expressed by

$$\beta_1^a(0) = |\beta_1^a(0)| R_d^a \tag{50a}$$

$$\beta_0^a(0) = |\beta_0^a(0)| R_d^a \tag{50b}$$

Though the same random number is used, since the two components on the rigid and free surfaces are normal to each other, they are defined independently.

According to the discussion in Appendix F, the initial concentration fluctuation is assumed to have the following Gaussian-type power spectrum.

$$P_{\text{int}}(a_x, a_y) = \frac{1}{\pi} \exp\left(-a^2\right) \tag{F5}$$

where $a^2 \equiv a_x^2 + a_y^2$ is defined. Substituting Equations (G7a) and (50a) into Equation (46a), we have the exact expression of $\beta_1^a$.

$$\beta_1^a(t) = \gamma_1^a f_r^a(a) \exp\left(-\frac{a^2}{2}\right) \exp(p_r^a t) R_d^a \tag{51a}$$

where the power spectrum component $\exp(-a^2/2)$ plays a low-pass filter concerning the wavenumber for the fluctuations. The constant part of the vorticity coefficient of the rigid surface vortex in 2D nucleation $\gamma_1^a$ is given by

$$\gamma_1^a \equiv \frac{1}{2}\alpha_r^a \left(\frac{XY}{\pi}\right)^{\frac{1}{2}} z_m F D_m \theta_\infty^* S^{*a} \tag{G7b}$$

where $X$ and $Y$ are $x$- and $y$-lengths of the electrode, respectively, and $\alpha_r^a$ is the initial ratio of the rigid surface component to the total concentration fluctuation. As discussed initially, the rigid and free surface components, as well as the vortices, distribute equally over the electrode, so that $\alpha_r^a = \sqrt{2}/2$ equal to that of the free surface component is assumed. For cathodic deposition, the concentration difference $\theta_\infty^* > 0$ is fulfilled, so that $\gamma_1^a > 0$ is derived. Substitution of Equations (G10a) and (50b) into Equation (48a) leads to the explicit equation of the vorticity coefficient of the free surface vortices.

$$\beta_0^a(t) = \gamma_0^a f_f^a(a) a^{-1} \exp\left(-\frac{a^2}{2}\right) \exp(p_f^a t) R_d^a \tag{51b}$$

The constant part of the vorticity coefficient of the free surface vortex in 2D nucleation $\gamma_0^a$ is given by

$$\gamma_0^a \equiv \frac{1}{2}\alpha_f^a \left(\frac{XY}{\pi}\right)^{\frac{1}{2}} z_m F D_m \theta_\infty^* S^{*a} \tag{G10b}$$

where $\alpha_f^a = \sqrt{2}/2$ is assumed as the initial ratio of the free surface component to the total concentration fluctuation and for deposition, $\gamma_0^a > 0$ is obtained. Equations (51a) and (51b) show that the vorticity coefficients are also restricted by the initial spectrum component $\exp(-a^2/2)$.

After assigning the random numbers at all grid points in the solution phase and considering the stochastic effect of Equations (50a) and (50b), the amplitude of the gradients of the concentration fluctuations in Equations (G8b) and (G11b) is rewritten as

$$D\Theta_r^0(0,t)^a = \frac{2\gamma_1^a f_r^a(a)}{z_m F D_m S^{*a}} \exp\left(-\frac{a^2}{2}\right) \exp(p_r^a t) R_d^a \tag{52a}$$

and

$$D\Theta_f^0(0,t)^a = \frac{2\gamma_0^a f_f^a(a)}{z_m F D_m S^{*a}} \exp\left(-\frac{a^2}{2}\right) \exp(p_f^a t) R_d^a \tag{52b}$$

$D\Theta_r^0(0,t)^a$ and $D\Theta_f^0(0,t)^a$ must independently distribute over the electrode surface without any contradiction, which mathematically implies that they are orthogonal normal packed.

To self-consistently calculate their self-organization processes, a complex Fourier transform is used, e.g., the rigid and free surface components are embedded as the real and imaginary parts in the forms of odd and even functions, such as sine and cosine functions concerning the wavenumber, respectively. Generally, the odd and even functions are normal to each other, whose symmetries are preserved in the transform.

In accordance with the above discussion, to embed the odd and even functions into a complex space, the following operator $\overline{C}$ is introduced.

$$\overline{C} \equiv R_e(\text{even}) + i \cdot I_m(\text{odd}) \tag{53a}$$

or

$$\overline{C} \equiv R_e(\text{odd}) + i \cdot I_m(\text{even}) \tag{53b}$$

where '(even)' and '(odd)' imply the even and odd functions embedded in the real and imaginary parts, and vice versa. Then, the allotment of random numbers to the 2D grids allows us to introduce random phases to the odd and even functions.

As a result, the complex amplitude of the concentration gradient function with random phases is expressed in the following.

$$\overline{C}D\Theta^0(0,t)^a = D\Theta_r^0(0,t)^a(\text{even}) + i \cdot D\Theta_f^0(0,t)^a(\text{odd}) \tag{54}$$

The $\overline{C}D\Theta^0(0,t)^a$ is transformed to the complex concentration gradient by the complex Fourier inversion, i.e.,

$$\overline{C}\left\{\frac{\partial c_m(x,y,0,t)^a}{\partial z}\right\}_{z=0} = \frac{1}{2\pi}\int_{-\infty}^{\infty}\int_{-\infty}^{\infty}\overline{C}D\Theta^0(0,t)^a \exp\left[-i(a_x x + a_y y)\right]da_x da_y \tag{55a}$$

where it should be noted that the coordinates $x$ and $y$ are defined as non-dimensional. The Fourier transformation makes only periodic components transformed. After obtaining the result, to reproduce the actual form of the asymmetrical fluctuation, some average values regarding $x$- and $y$-coordinates must often be added to it. The complex amplitudes of the rigid and free surface components are transformed to the concentration gradient in a complex space in keeping their symmetries, i.e.,

$$\overline{C}\left\{\frac{\partial c_m(x,y,z,t)^a}{\partial z}\right\}_{z=0} = \left\{\frac{\partial c_{m,r}(x,y,z,t)^a}{\partial z}\right\}_{z=0}(\text{even}) + i \cdot \left\{\frac{\partial c_{m,f}(x,y,z,t)^a}{\partial z}\right\}_{z=0}(\text{odd}) \tag{55b}$$

where $c_{m,r}(x,y,z,t)^a$ and $c_{m,f}(x,y,z,t)^a$ are the rigid and free surface concentration fluctuations, respectively. The rigid and free surface components are transformed as even and odd functions with random phases of the $x$- and $y$-coordinates, respectively. As shown in Figure 6b, nonequilibrium fluctuations have either sign of positive or negative. For cathodic deposition, the concentration and concentration gradient fluctuations take negative and positive values, respectively. To derive the negative current density, with the root mean square (rms) values of the fluctuations, we must cut off the extra negative portions of the concentration gradient fluctuations. From Rayleigh's theorem, the mean square value of the concentration gradient fluctuation is equalized to the mean square value of its amplitude.

For the rigid or free surface component, it follows that

$$\frac{1}{XY}\iint_{-\infty}^{\infty}\left\{\frac{\partial c_{m,i}(x,y,z,t)^a}{\partial z}\right\}_{z=0}^2 dxdy = \frac{1}{XY}\iint_{-\infty}^{\infty}\left\{D\Theta_i^0(0,t)^a\right\}^2 da_x da_y \quad \text{for i = r or f} \tag{56a}$$

where $X$ and $Y$ imply the $x$- and $y$-lengths of the electrode. From Equation (56a), the root mean square (rms) value of the concentration gradient can be calculated by the rms value of the amplitude as follows.

$$\text{rms}\left\{\frac{\partial c_{m,i}(x,y,z,t)^a}{\partial z}\right\}_{z=0} \equiv \left[\frac{1}{XY}\iint_{-\infty}^{\infty}\left\{D\Theta_i^0(0,t)^a\right\}^2 da_x da_y\right]^{\frac{1}{2}} \quad (> 0) \text{ for i = r or f} \tag{56b}$$

Therefore, by adding the root mean square value, the "phase cutting" is completed, i.e., the concentration gradient fluctuations are redefined.

$$\left\{\frac{\partial c_{m,i}(x,y,z,t)^a}{\partial z}\right\}_{z=0} \equiv \left\{\frac{\partial c_{m,i}(x,y,z,t)^a}{\partial z}\right\}_{z=0} + \text{rms}\left\{\frac{\partial c_{m,i}(x,y,z,t)^a}{\partial z}\right\}_{z=0} \quad \text{for i = r or f} \tag{56c}$$

The concentration gradient distributed over the electrode surface is completely reproduced by the linear combination of the rigid and free surface components of the even and odd functions with respect to $x$- and $y$-coordinates.

$$\left\{\frac{\partial c_{\mathrm{m}}(x,y,0,t)^{\mathrm{a}}}{\partial z}\right\}_{z=0} = \left\{\frac{\partial c_{\mathrm{m,r}}(x,y,0,t)^{\mathrm{a}}}{\partial z}\right\}_{z=0} + \left\{\frac{\partial c_{\mathrm{m,f}}(x,y,0,t)^{\mathrm{a}}}{\partial z}\right\}_{z=0} \quad (>0) \quad (57)$$

Finally, the current densities on the rigid and free surfaces are derived from Fick's first law in the following.

$$j_{z,\,\mathrm{r}}(x,y,0,t)^{\mathrm{a}} = -z_{\mathrm{m}}FD_{\mathrm{m}}\left\{\frac{\partial c_{\mathrm{m,r}}(x,y,z,t)^{\mathrm{a}}}{\partial z}\right\}_{z=0} \quad (<0) \tag{58a}$$

and

$$j_{z,\,\mathrm{f}}(x,y,0,t)^{\mathrm{a}} = -z_{\mathrm{m}}FD_{\mathrm{m}}\left\{\frac{\partial c_{\mathrm{m,f}}(x,y,z,t)^{\mathrm{a}}}{\partial z}\right\}_{z=0} \quad (<0) \tag{58b}$$

The total current density $j_z(x,y,0,t)^{\mathrm{a}}$ is thus expressed by

$$j_z(x,y,0,t)^{\mathrm{a}} = j_{z,\,\mathrm{r}}(x,y,0,t)^{\mathrm{a}} + j_{z,\,\mathrm{f}}(x,y,0,t)^{\mathrm{a}} \tag{59a}$$

As a result, the surface morphology of the 2D nuclei is effectively calculated by the surface height fluctuation.

$$\zeta(x,y,t)^{\mathrm{a}} = -\frac{\Omega_{\mathrm{m}}}{z_{\mathrm{m}}F}\int_0^t j_z(x,y,0,t)^{\mathrm{a}}\mathrm{d}t \tag{59b}$$

where $\Omega_{\mathrm{m}}$ implies the molar volume of the deposit metal. Based on Equations (59a) and (59b), the surface morphology of the 2D nuclei can be divided into the rigid and free surface components, i.e.,

$$\zeta(x,y,t)^{\mathrm{a}} = \zeta_{\mathrm{r}}(x,y,t)^{\mathrm{a}} + \zeta_{\mathrm{f}}(x,y,t)^{\mathrm{a}} \tag{59c}$$

### 3.4. The Rotational Directions of the Micro-MHD Flows on the Rigid and Free Surfaces

As discussed initially, we can determine the characteristics of the rigid and free surface vortices by the signs of the $z$-components of the velocity and vorticity fluctuations. Namely, the rigid and free surface vortices correspond to the vortices with downward and upward flows, which are expressed by the negative and positive values of the $z$-components of the velocity fluctuation, respectively. In addition, the CW and ACW rotations are provided by negative and positive values of the $z$-component of the vorticity.

As shown in Appendices H and K, the $x$- and $y$-components of the velocity are calculated by the $z$-components of the velocity and vorticity.

(a)  For the rigid surfaces:

Using Equations (21a) and (22a), we redefine Equation (16a) as

$$W_{\mathrm{r}}^0(z,t)^a = 2\beta_1^{\mathrm{a}}[\{\alpha_{0\mathrm{r}}^{*\mathrm{a}}(a) + \alpha_{1\mathrm{r}}^{*\mathrm{a}}(a)z\}\sin\mathrm{h}\,az - \alpha_{0\mathrm{r}}^{*\mathrm{a}}(a)\,az\,\exp(-az)] \tag{60}$$

Then, inserting Equation (51a) into Equation (60), we obtain

$$W_{\mathrm{r}}^0(z,t)^a = 2\gamma_1^{\mathrm{a}}\mathrm{f}_{\mathrm{r}}^{\mathrm{a}}(a)\exp\left(-\frac{a^2}{2}\right)\exp(p_{\mathrm{r}}^{\mathrm{a}}t)[\{\alpha_{0\mathrm{r}}^{*\mathrm{a}}(a) + \alpha_{1\mathrm{r}}^{*\mathrm{a}}(a)z\}\sin\mathrm{h}\,az - \alpha_{0\mathrm{r}}^{*\mathrm{a}}(a)\,az\,\exp(-az)]R_{\mathrm{d}}^{\mathrm{a}} \tag{61a}$$

Replacing $\beta_1$ in Equation (16b) with $\beta_1^{\mathrm{a}}$ in Equation (51a), we have

$$\Omega_{\mathrm{r}}^0(z,t)^a = \gamma_1^{\mathrm{a}}\mathrm{f}_{\mathrm{r}}^{\mathrm{a}}(a)\exp\left(-\frac{a^2}{2}\right)\exp(p_{\mathrm{r}}^{\mathrm{a}}t)\,z\,\exp(az)R_{\mathrm{d}}^{\mathrm{a}} \tag{61b}$$

where Equations (21b) and (22b) supply the following expressions.

$$\alpha_{0r}^{*a}(a) = -\frac{e^a\{2Q^{*a}a^2(2\cos h\, a + a\sin h\, a) + d^{a2}T^{*a2}(1+a)(\sin h\, a + a\cos h\, a)\}}{2Q^{*a}T^{*a}a^2\left(\sin h^2 a + a^2\right)} \tag{62a}$$

and

$$\alpha_{1r}^{*a}(a) = \frac{e^a\left[2Q^{*a}a^2\{\cos h\, a + (1-a)e^{-a}\} + d^{a2}T^{*a2}(1+a)(\sin h\, a + ae^{-a})\right]}{2Q^{*a}T^{*a}a\left(\sin h^2 a + a^2\right)} \tag{62b}$$

(b)　For the free surfaces:

Using Equations (25a) and (26a), we redefine Equation (18a) as

$$W_f^0(z,t)^a = 2\beta_0^a\{\alpha_{0f}^{*a}(a)\sin h\, az + \alpha_{1f}^{*a}(a)z\cos h\, az\} \tag{63}$$

Inserting Equation (51b) into Equation (63), we obtain

$$W_f^0(z,t)^a = 2\gamma_0^a f_f^a(a)a^{-1}\exp\left(-\frac{a^2}{2}\right)\exp(p_f^a t)\{\alpha_{0f}^{*a}(a)\sin h\, az + \alpha_{1f}^{*a}(a)z\cos h\, az\}R_d^a \tag{64a}$$

Inserting Equation (51b) into Equation (18b), we have

$$\Omega_f^0(z,t)^a = \gamma_0^a f_f^a(a)a^{-1}\exp\left(-\frac{a^2}{2}\right)\exp(p_f^a t)(1-az)\exp(az)R_d^a \tag{64b}$$

where Equations (25b) and (26b) supply the expressions of $\alpha_{0f}^{*a}(a)$ and $\alpha_{1f}^{*a}(a)$.

$$\alpha_{0f}^{*a}(a) = \frac{e^a\{2Q^{*a}a(2\sin h\, a + a\cos h\, a) + d^{a2}T^{*a2}(\cos h\, a + a\sin h\, a)\}}{2Q^{*a}T^{*a}(\sin h\, a\cos h\, a + a)} \tag{65a}$$

and

$$\alpha_{1f}^{*a}(a) = -\frac{ae^a\left(2Q^{*a}a\sin h\, a + d^{a2}T^{*a2}\cos h\, a\right)}{2Q^{*a}T^{*a}(\sin h\, a\cos h\, a + a)} \tag{65b}$$

Then, the rigid and free surface components are embedded into real and imaginary parts of a complex amplitude in the forms of even and odd functions regarding the wavenumber.

$$\overline{C}W^0(0,t)^a = W_r^0(0,t)^a(\text{even}) + i\cdot W_f^0(0,t)^a(\text{odd}) \tag{66a}$$

and

$$\overline{C}\Omega^0(0,t)^a = \Omega_r^0(0,t)^a(\text{even}) + i\cdot\Omega_f^0(0,t)^a(\text{odd}) \tag{66b}$$

In accordance with the above discussion of the 2D nucleation, the actual $z$-components of the velocity and vorticity fluctuations are calculated by the following complex Fourier inversions, where the rigid and free surface components are embedded as real and imaginary parts in the forms of even and odd functions regarding $x$- and $y$-coordinates, respectively.

$$\overline{C}w(x,y,z,t)^a = \frac{1}{2\pi}\int_{-\infty}^{\infty}\int_{-\infty}^{\infty}\overline{C}W^0(z,t)^a\exp\left[-i(a_x x + a_y y)\right]da_x da_y \tag{67a}$$

and

$$\overline{C}\omega_z(x,y,z,t)^a = \frac{1}{2\pi}\int_{-\infty}^{\infty}\int_{-\infty}^{\infty}\overline{C}\Omega^0(z,t)^a\exp\left[-i(a_x x + a_y y)\right]da_x da_y \tag{67b}$$

As discussed in Equations (55a) and (55b), the Fourier transform makes the transformation of the periodic components, so that the actual forms are reproduced by adding the average components to them. As mentioned above, the vortices are classified by four characters; the rigid and free surface components correspond to the downward flow $w_r(x,y,z,t)^a < 0$

and the upward flow $w_f(x, y, z, t)^a > 0$, respectively. On the other hand, the rotational directions are determined by the precessions from the upper layer. CW and ACW rotations in bird's-eye view correspond to the negative vorticity $\omega_{z,i}(x, y, z, t)^a < 0$ and the positive vorticity $\omega_{z,i}(x, y, z, t)^a > 0$ for i = r or f, respectively. Therefore, in accordance with Equation (56c), we redefine the velocities and vorticities.

For the rigid surface vortices, we have

$$w_r(x, y, z, t)^a \equiv w_r(x, y, z, t)^a - \text{rms } w_r(x, y, z, t)^a \quad (< 0) \tag{68a}$$

and

$$\omega_{z,r}(x, y, z, t)^a \equiv \omega_{z,r}(x, y, z, t)^a \pm \text{rms } \omega_{z,r}(x, y, z, t)^a \quad (> 0 \text{ or } < 0) \tag{68b}$$

For the free surface vortices, we have

$$w_f(x, y, z, t)^a \equiv w_f(x, y, z, t)^a + \text{rms } w_r(x, y, z, t)^a \quad (> 0) \tag{69a}$$

and

$$\omega_{z,f}(x, y, z, t)^a \equiv \omega_{z,f}(x, y, z, t)^a \mp \text{rms } \omega_{z,f}(x, y, z, t)^a \quad (> 0 \text{ or } < 0) \tag{69b}$$

where rms $w_i(x, y, z, t)^a$ and rms $\omega_{z,i}(x, y, z, t)^a$ (i = r or f) imply the rms values of $w_i(x, y, z, t)^a$ and $\omega_{z,i}(x, y, z, t)^a$, which are expressed by

$$\text{rms } w_i(x, y, z, t)^a \equiv \left[ \frac{1}{XY} \iint\limits_{-\infty}^{\infty} \left\{ W_i^0(z, t)^a \right\}^2 da_x da_y \right]^{\frac{1}{2}} \quad (> 0) \text{ for i = r or f} \tag{70a}$$

and

$$\text{rms } \omega_{z,i}(x, y, z, t)^a \equiv \left[ \frac{1}{XY} \iint\limits_{-\infty}^{\infty} \left\{ \Omega_i^0(z, t)^a \right\}^2 da_x da_y \right]^{\frac{1}{2}} \quad (> 0) \text{ for i = r or f} \tag{70b}$$

Actual distributions of the velocity and vorticity are expressed by the linear combinations of rigid and free components, respectively.

$$w(x, y, z, t)^a = w_r(x, y, z, t)^a + w_f(x, y, z, t)^a \tag{71a}$$

and

$$\omega_z(x, y, z, t)^a = \omega_{z,r}(x, y, z, t)^a + \omega_{z,f}(x, y, z, t)^a \tag{71b}$$

As mentioned above, the pair of vortices are composed of the vortices with opposite $z$-components of velocities and vorticities, and downward and upward $z$-components of the velocity correspond to rigid and free surface vortices. Since a pair of vortices with upward and downward flows have opposite rotations, as shown in Figure 8, the vortices are simply classified into two sets, where the distributions of the phase-cut quantities in $x$-direction are schematically exhibited.

As shown in Figure 9, the formation processes of chiral 2D and 3D nuclei plus chiral screw dislocation constitute nesting boxes, i.e., chiral 2D nuclei develop on an electrode, chiral 3D nuclei grow on a chiral 2D nucleus, and chiral screw dislocations are created on a chiral 3D nucleus. As discussed initially, for chirality to emerge, two types of vortices are necessary. One of them (i.e., the rigid surface vortex) directly contributes to the chiral nucleation, where the rigid surfaces under a vortex of the micro-MHD flow are formed by locally exposed rigid surfaces, such as a bee's nest. Each local rigid surface corresponds to the bottom of a smaller rigid surface vortex, yielding a chiral nucleus of the next generation. On the other hand, on the free surfaces covered with ionic vacancies, due to the moving solution of a vortex at the electrode surface, such microstructures of 3D nuclei are not created.

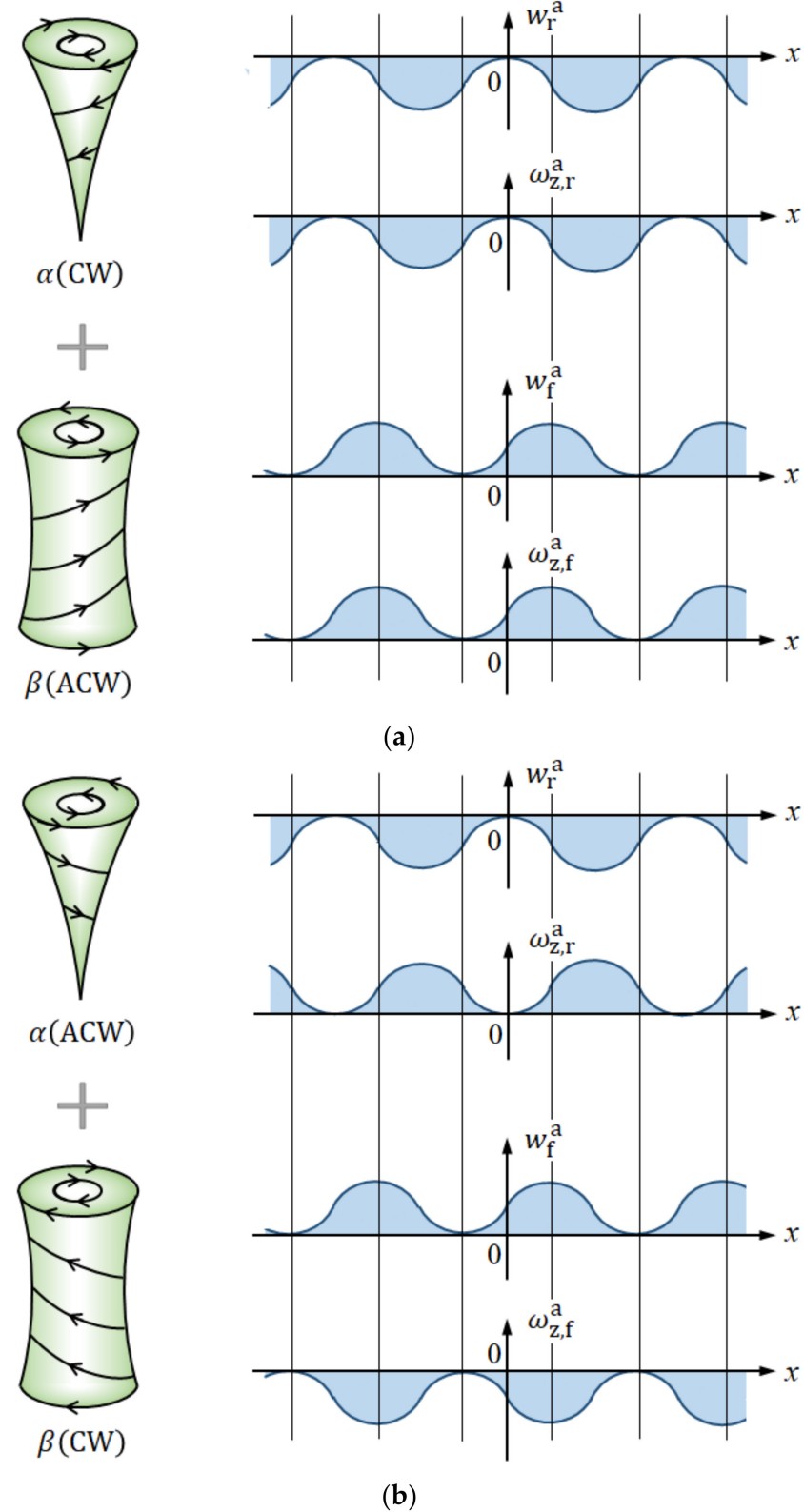

**Figure 8.** Two sets of the micro-MHD vortex pair formed on the rigid and free surfaces. (**a**) CW-ACW vortex pair. (**b**) ACW-CW vortex pair. $\alpha$, rigid surface vortex; $\beta$, free surface vortex; $w_r^a$, $z$-velocity component of the rigid surface vortex; $\omega_{z,r}^a$, $z$-vorticity component of the rigid surface vortex; $w_f^a$, $z$-velocity component of the free surface vortex; $\omega_{z,f}^a$, $z$-vorticity component of the free surface vortex.

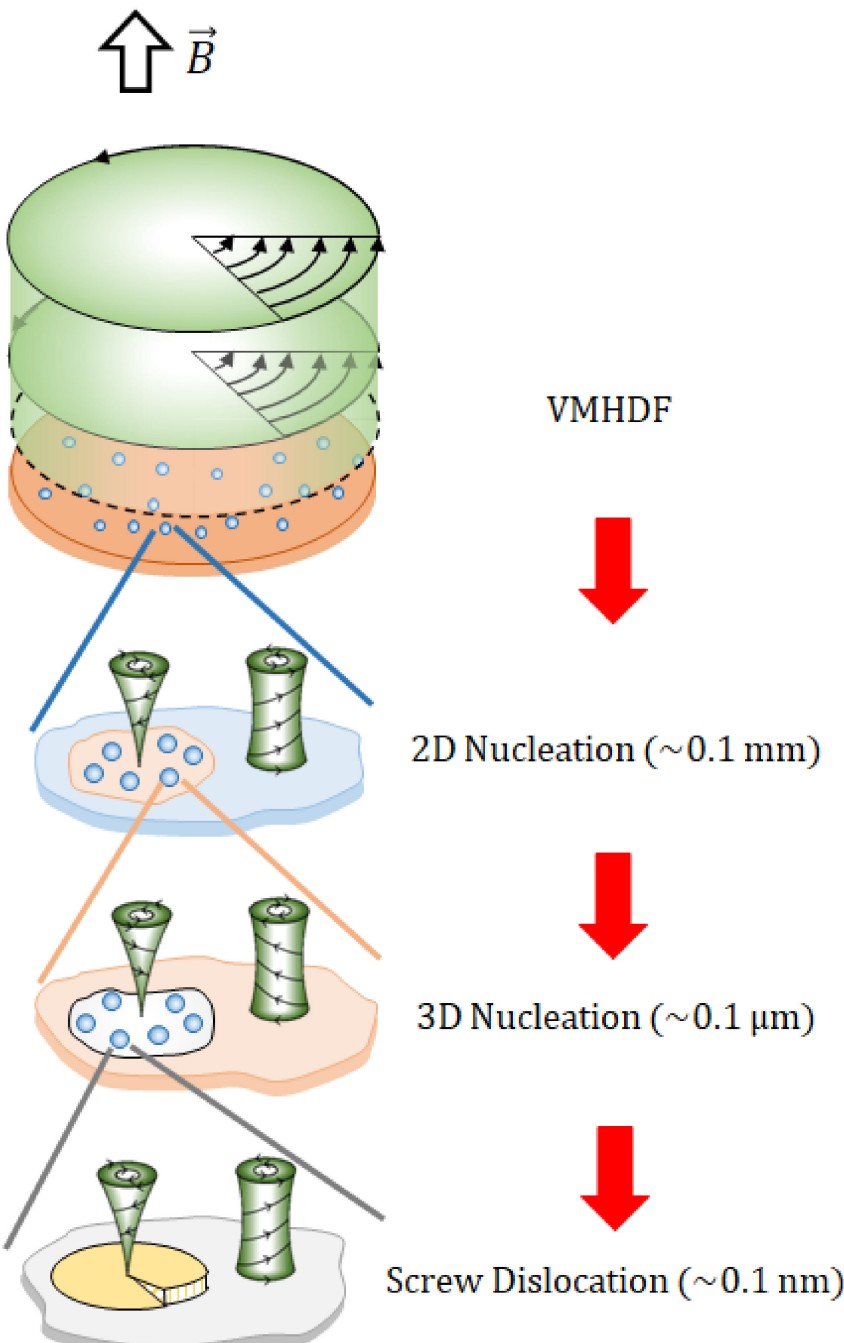

**Figure 9.** Nesting-boxes structure of chiral 2D and 3D nucleation plus chiral screw dislocation.

## 4. Results and Discussion

### 4.1. Micro-Mystery Circles Formed by the Non-Specific Adsorption of Ions

Figure 10 exhibits the copper electrode surfaces deposited under a VMHDF. Through the deposition, characteristic concave round patterns with diameters of the order of 100 μm, called micro-mystery circles, were observed (Figure 10a). However, after encircling the VMHDE with a sheath to stop the rotation of the VMHDF, such patterns disappear on the deposit surface (Figure 10b). Since these experimental results do not directly indicate the chirality of the 2D nuclei, in the following, based on the theoretical foundation discussed above, we calculate the morphological pattern of the micro-mystery circle and examine whether such a concave pattern is theoretically reproduced or not.

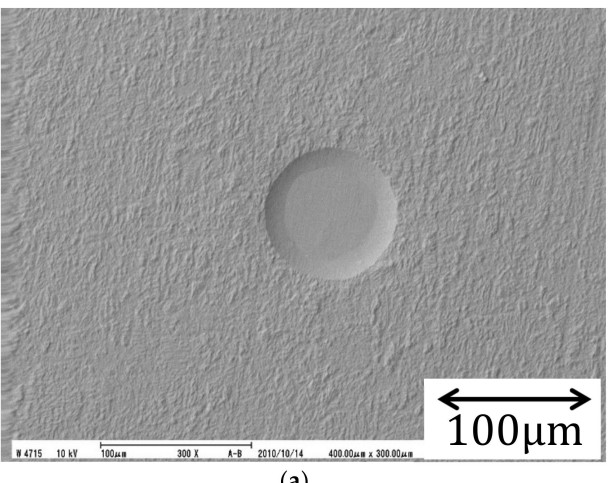 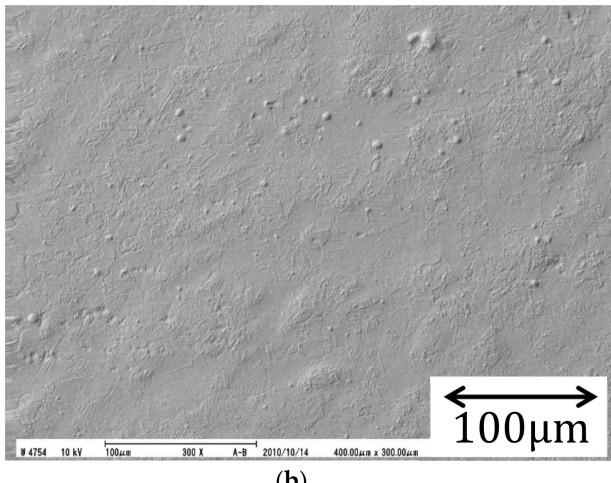

(**a**)          (**b**)

**Figure 10.** SEM images of the surface morphologies of copper electrodeposition under vertical magnetic fields. (**a**) A micro-mystery circle formed by a VMHDF. (**b**) Disappearance of the micro-mystery circle by the blocking of VMHDF with a sheath. Though the applied magnetic flux density is different, i.e., (**a**) 3 T and (**b**) 1 T, the other conditions are the same. Deposition time is 300 s. A $500 \, \text{mol m}^{-3}$ $CuSO_4$ + $500 \, \text{mol m}^{-3}$ $H_2SO_4$ solution was used at room temperature.

In the preceding paper [21], it has been clarified that the MHD flow patterns formed on the copper deposit surface are reproduced after multiple nucleation. In accordance with the procedure shown in Equations (49)–(59c), we calculated the copper deposit surfaces after repeating the one hundred-times nucleation under a VMHDF: Assigning random numbers to the grid points of a defined electrode surface at the beginning of each nucleation, 2D nuclei start to develop with time, expanding two-dimensionally, so that at their borders, some disorders take place. In actual 2D nucleation, such disorders would be self-consistently reformed. However, in this theory, because the treatment of such boundary disorders is not considered, as shown in Figure 11, the disorders of the boundaries remain as they are. In the calculated images, the concave circular patterns of the same order of magnitude as the SEM image appear on the deposit surface. In the present case of nonspecific adsorption of ions, the nucleus on the rigid surface grows more slowly than the nucleus on the free surface, so we can expect that the concave part is mainly composed of the rigid surfaces of 2D nuclei involving chiral 3D nuclei. We can also suppose that chiral screw dislocations would be created on the 3D nuclei. Namely, in quite high probability, it is expected that the chiral activity exists on the concave part.

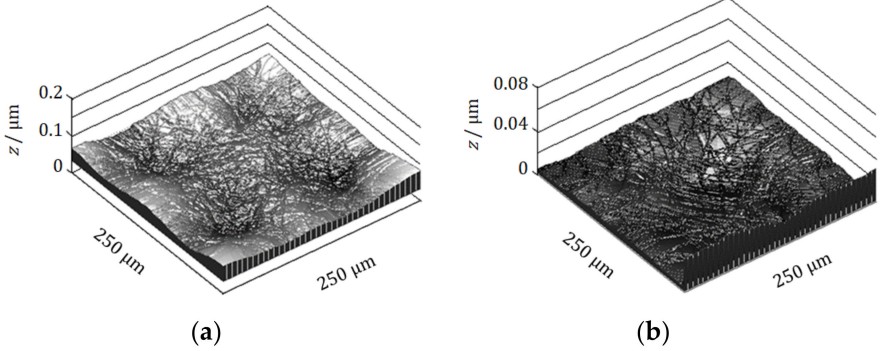

(**a**)          (**b**)

**Figure 11.** Theoretical calculation of micro-mystery circles after multiple 2D nucleation. (**a**) The case of four circles. (**b**) The case of a single circle. Calculation data are as follows: $B_0 = 5$ T, $\tilde{\Omega} = 62.8 \, \text{s}^{-1}$, $z_m = 2$, $D_m = 6 \times 10^{-10} \, \text{m}^2 \, \text{s}^{-1}$, $\langle \delta_c \rangle = 3.74 \times 10^{-4}$ m, $C_m(z = \infty) = 50$ mol $\text{m}^{-3}$. Supporting electrolyte, $500 \, \text{mol m}^{-3}$; applied overpotential, $-0.4$ V; nucleation period, 1.0 s; nucleation number, 100.

As discussed initially, due to the conservation of angular momentums of the vortices activated from a stationary state, vortices with ACW and CW rotations are equally evolved, so that half of the nuclei randomly created in each generation would be chiral ones formed on the rigid surfaces. In view of the nesting-boxes structure of the chiral nucleation shown in Figure 9, the probability that the chiral screw dislocations emerge from all active points is obtained by the product of the probability of each generation.

$$\varepsilon_{\text{screw}} = \frac{1}{2} \times \frac{1}{2} \times \frac{1}{2} = \frac{1}{8} (= 0.125) \tag{72}$$

Equation (72) is derived from the three-generation model of chiral nucleation under the initial condition that rigid surface and free surface vortices are equally distributed over the electrode. This is a strong restriction for all the vortices in the three generations.

The created screw dislocations act as single active points for enantiomeric reagents. Figure 12 schematically exhibits D-active and L-active surfaces of a nucleus, i.e., the surfaces are uniformly covered with single and achiral active points. A single active point is active for either D- and L-reagents and inactive for the other one, whereas an achiral active point is active for both D- and L-reagents. Owing to the uniform distribution, we can calculate the reaction current of the electrode by the ratios of the single active points and achiral active points, i.e., $\varepsilon_{\text{screw}}$ and $1 - \varepsilon_{\text{screw}}$, respectively.

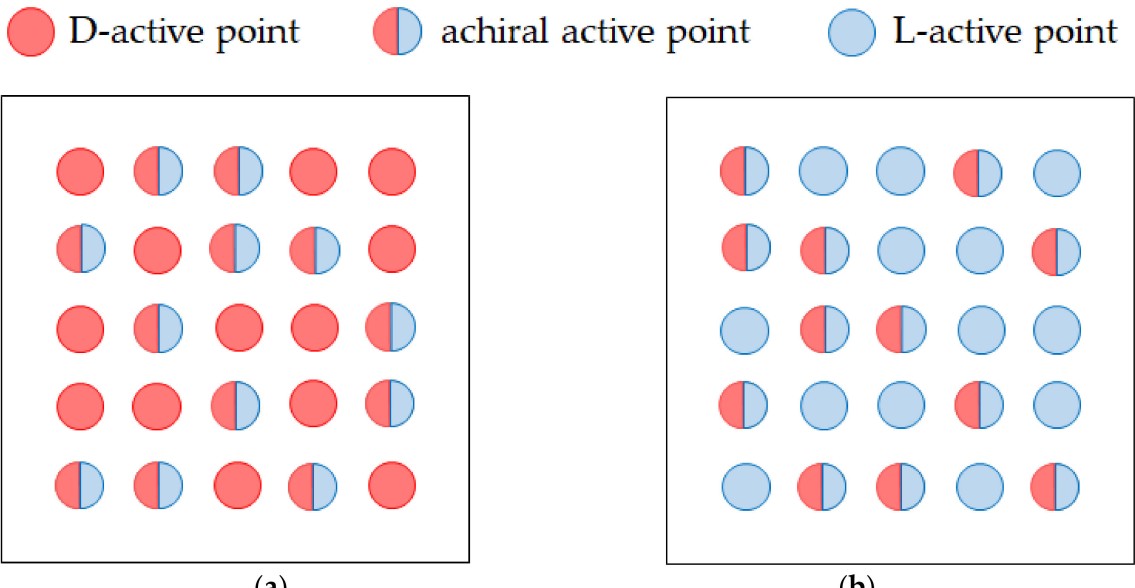

**Figure 12.** D-active and L-active surfaces of a nucleus. (**a**) D-active surface. (**b**) L-active surface. Red circle, D-active point; blue circle, L-active point; semicircles of red and blue, achiral active point.

Because an achiral active point is composed of both D- and L-active portions, the probability for an enantiomeric reagent to react at the active point is equal to 1/2. Therefore, we can assume that the activity of an achiral active point is half of a single active point. In view of twice larger activity of a single active point, for the electrode active for either D- or L-reagents, the total current is written by

$$I_{\text{active}} = 2\varepsilon_{\text{screw}} I_0 + (1 - \varepsilon_{\text{screw}}) I_0 = (1 + \varepsilon_{\text{screw}}) I_0 \tag{73a}$$

where $I_0$ implies the total current of the electrode covered with only achiral active points.

On the other hand, the single active point of the electrode is inactive for the other reagent, so that for the reagent, the current component of the single active point becomes zero, and the total current is equal to the current of the achiral active points.

$$I_{\text{inactive}} = (1 - \varepsilon_{\text{screw}})I_0 \tag{73b}$$

Using Equations (73a) and (73b), we can calculate the absolute value of the enantiomeric excess (*ee*) ratio defined by Mogi, i.e.,

$$|r(ee)| \equiv \frac{I_{\text{active}} - I_{\text{inactive}}}{I_{\text{active}} + I_{\text{inactive}}} = \varepsilon_{\text{screw}} \tag{73c}$$

where the *ee* ratio itself is defined as positive for L-activity and negative for D-activity. The absolute value of the *ee* ratio is equal to the probability that the chiral screw dislocations emerge from all the active points. Since $\varepsilon_{screw} = 0.125$ corresponds to an ideal limiting case, we can conclude that the absolute value of the *ee* ratio cannot exceed 0.125.

$$|r(ee)| \leq \varepsilon_{\text{screw}}(= 0.125) \tag{74}$$

Equation (74) is supported by the experimental data of Mogi's reports [9–18]. Namely, the three-generation model is experimentally validated. An *ee* ratio of 0.125 is declared as the ideal limiting value obtained by the present method, which results from the fact that the evolution probability of the rigid surface vortex is equal to that of the free surface vortex.

### 4.2. Inversion of Chirality by the Specific Adsorption of Chloride Ions

As mentioned above, the magnetic field, current, and Lorentz force consist of a right-handed system, so that according to the law of a right-handed system, Equation (44) is always fulfilled. As a result, under a parallel magnetic field $(B_0 < 0)$ or an antiparallel magnetic field $(B_0 > 0)$, in a bird's-eye view, the upper layer, i.e., the VMHDF rotates in a clockwise $\left(\widetilde{\Omega} < 0\right)$ or an anticlockwise $\left(\widetilde{\Omega} > 0\right)$ direction, respectively. Then, the vortices in the lower layer receive the precessions from the upper layer via the vortices in the upper layer. As shown in Figure 2C, due to the continuity of the vortex motion, two adjoining vortices form a pair of vortices with reverse rotations as well as downward and upward flows, so that a rigid surface vortex appears with a free surface vortex. If either of them starts a precessional motion, the other must subordinately rotate in the opposite direction. Then, the next problem is which the vortex receives the precession, rigid or free surface. There are two cases, i.e., one is an unstable case where the vortices develop with time and the other is a stable case where the vortices, once activated, dwindle with time.

As discussed in Equations (46b) and (48b), in accordance with the signs of the amplitude factors $p_r$ and $p_f$, the micro-MHD vortices will develop or decay with time. As shown in Figure 7, the amplitude factor functions $f_r^0(a)$ and $f_f^0(a)$ in Equations (45b) and (47b), always take positive values for all wavenumbers, so that from Equations (46b) and (48b), the signs of $p_r$ (i.e., $p_r^a$) and $p_f$ (i.e., $p_f^a$) determined by the adsorption coefficient $A_\theta$ shown in Equation (43b), whose sign depends on the sign of the product of the differential potential coefficient and the overpotential at the OHP($(\partial\langle\Phi_1\rangle/\langle\Phi_1\rangle)_\mu\Phi_{2\text{OHP}}^*$). As discussed in Appendix A, as for ionic adsorption, the following conditions concerning unstable and stable growths of the fluctuations are derived.

$$\left(\frac{\partial\langle\Phi_1\rangle}{\partial\langle\Phi_2\rangle}\right)_\mu \Phi_{2\text{OHP}}^* < 0 \quad (\text{unstable}) \text{ for non} - \text{specific adsorption} \tag{A9a}$$

$$\left(\frac{\partial\langle\Phi_1\rangle}{\partial\langle\Phi_2\rangle}\right)_\mu \Phi_{2\text{OHP}}^* > 0 \quad (\text{stable}) \text{ for specific adsorption} \tag{A9b}$$

Here, the differential potential coefficient is expressed by [66]

$$\left(\frac{\partial \langle \Phi_1 \rangle}{\partial \langle \Phi_2 \rangle}\right)_\mu = \frac{\varepsilon}{\lambda C_H} \left\{ \left(\frac{\partial Q_1^*}{\partial Q_2^*}\right)_\mu + 1 \right\} \tag{75}$$

where $\varepsilon$ implies the dielectric constant of water ($6.95 \times 10^{-10}$ $J^{-1}$ $C^2$ $m^{-1}$, 25 °C), $C_H$ is the electric capacity of the Helmholtz layer ($\approx 10$ µF $cm^{-2} = 0.1$ F $m^{-2}$ [67]), and $\lambda$ is the Debye length shown in Equation (35c). $\left(\partial Q_1^*/\partial Q_2^*\right)_\mu$ is the differential charge coefficient, where $Q_1^*$ and $Q_2^*$ imply the electric charges stored in the Helmholtz and diffuse layers of an electric double layer [68,69]. From our preliminary experiments, we obtained

$$\left(\frac{\partial Q_1^*}{\partial Q_2^*}\right)_\mu = 0.250 \quad \text{for a Cu electrode in a } CuSO_4 + H_2SO_4 \text{ solution} \tag{76a}$$

$$\left(\frac{\partial Q_1^*}{\partial Q_2^*}\right)_\mu = -2.10 \quad \text{for a Cu electrode in a } CuCl_2 + HCl \text{ solution} \tag{76b}$$

$$\left(\frac{\partial Q_1^*}{\partial Q_2^*}\right)_\mu = -2.02 \quad \text{for a Cu electrode in a } CuCl_2 + KCl \text{ solution} \tag{76c}$$

In accordance with Appendix A, the present case shown in Equation (76a) certainly corresponds to the non-specific adsorption, whereas Equations (76b) and (76c) indicate the strong specific adsorption of chloride ions. Since the chemical bonding force of a chloride ion is stronger than the repulsive electrostatic force, it can adsorb on the copper cathodic surface. For a 500 mol $m^{-3}$ $H_2SO_4$ supporting electrolyte solution, from Equation (35c), we obtain the Debye length $\lambda = 2.47 \times 10^{-10}$ m. Using $C_H = 0.1$ F $m^{-2}$ as well as Equation (75), we have the differential potential coefficients as follows:

$$\left(\frac{\partial \langle \Phi_1 \rangle}{\partial \langle \Phi_2 \rangle}\right)_\mu = 35.1 \quad \text{for a } CuSO_4 + H_2SO_4 \text{ system} \tag{77a}$$

$$\left(\frac{\partial \langle \Phi_1 \rangle}{\partial \langle \Phi_2 \rangle}\right)_\mu = -30.9 \quad \text{for a } CuCl_2 + HCl \text{ system} \tag{77b}$$

$$\left(\frac{\partial \langle \Phi_1 \rangle}{\partial \langle \Phi_2 \rangle}\right)_\mu = -28.7 \quad \text{for a } CuCl_2 + KCl \text{ system} \tag{77c}$$

In view of cathodic polarization of the diffuse layer $\Phi_{2OHP}^* < 0$, Equation (77a) corresponds to the unstable condition Equation (A9a), whereas Equations (77b) and (77c) derive the stable condition Equation (A9b).

The differential potential coefficients in the case of the specific adsorption of chloride ions in Equations (77b) and (77c) are smaller than $-1$. These results mean that the chloride ions induce strong specific adsorption. From the discussion in Appendix A, for the non-specific and strong specific adsorptions in the cathodic deposition, the overpotentials of the diffuse layers take negative values. Assuming $\Phi_{2OHP}^* \approx -1 \times 10^{-2}$ V, we can calculate the amplitude factors $p_r^a$ and $p_f^a$ in Equations (46b) and (48b).

As a result, in the case of the non-specific adsorptions of simple ions at the electrode surface, as shown in Figure 13a, the amplitude factors $p_r^a$ and $p_f^a$ take positive values for all wave numbers $a$, so that the asymmetrical fluctuations, including the vortices of micro-MHD flows, become unstable, i.e., activated vortices deterministically develop with time. Due to the larger positive amplitude factor, the free surface vortices grow faster than the rigid surface ones. The precessions by the VMHDF therefore transfer to the free surface vortices, so that the rigid surface vortices creating chiral 2D nuclei rotate in the opposite direction of the VMHDF rotation, yielding 2D nuclei with opposite chirality. In unstable 3D nucleation, we can also expect that the free surface vortices on a rigid surface of a 2D nucleus are given the priority in precession. This time, due to the opposite

rotational direction, the rigid surfaces of 3D nuclei and the nano-vortices will obtain the same chirality as the VMHDF. If the same process were repeated in the third generation, the screw dislocations formed on a rigid surface of 3D nucleus would obtain the opposite chirality. This means that the electrode gains the opposite chiral activity as the rotations of the VMHDF.

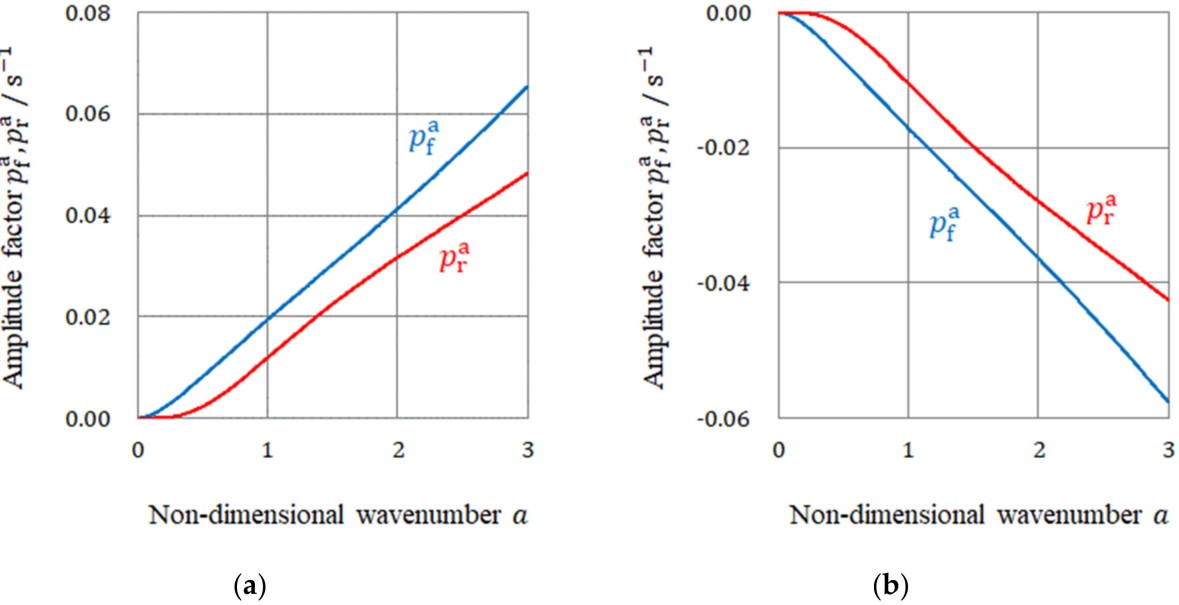

**(a)** **(b)**

**Figure 13.** Amplitude factors $p_r^a$ and $p_f^a$ for non-specific and specific adsorptions. (**a**) The case of non-specific adsorption. (**b**) The case of specific adsorption.

For the specific adsorption of chloride ions, as shown in Figure 13b, $p_r^a$ and $p_f^a$ become negative for all wavenumbers, so that the fluctuations are kept stable, i.e., an initially activated vortex dwindles to nothing with time. Such activation and extinction are stochastically repeated regarding time and location. In this case, due to the smaller negative amplitude factor, the rigid surface vortices decay more slowly than the free surface ones receiving the precessions. Therefore, in this case, 2D nuclei with the same chirality as the VMHDF rotation emerge under the rigid surface vortices. However, since 3D nuclei and screw dislocations do not grow in an electric double layer, but in a diffusion layer, their unstable free surface vortices will be again given priority in precession, so that the chirality in the first generation would be transferred to the third generation as it stands. This means that by adding a chloride additive, the electrode gains the same chiral activity as the VMHDF rotation.

As for 2D chiral nucleation, in the absence of specific adsorption of ions, it is concluded that under an upward antiparallel ($B_0 > 0$) or a downward parallel ($B_0 < 0$) magnetic field, clockwise (CW) $\left( \widetilde{\Omega}_r^a < 0 \right)$ or anticlockwise (ACW) $\left( \widetilde{\Omega}_r^a > 0 \right)$ rotations occur in the rigid surface vortices, respectively. That is, such a relationship is expressed by

$$B_0 \widetilde{\Omega}_r^a < 0 \tag{78}$$

where $\widetilde{\Omega}_r^a$ implies the representative angular velocity of the rigid surface vortices. This symmetry of rotation is consistent with the odd symmetry of chiral activity of the copper deposit surfaces.

On the other hand, for the specific adsorption, such as chloride adsorption, the amplitude factors become negative. As mentioned above, in the stable case, the rigid surface vortices rotate in the same direction as a VMHDF. Namely, in the presence of specific adsorption of ions, we can say that the rotational directions of the vortices creating chiral nuclei are reversed, i.e., upward antiparallel ($B_0 > 0$) and downward parallel ($B_0 < 0$) mag-

netic fields induce anticlockwise (ACW) $\left(\widetilde{\Omega}_r^{\mathrm{a}} > 0\right)$ and clockwise (CW) $\left(\widetilde{\Omega}_r^{\mathrm{a}} < 0\right)$ rotations of the rigid surface vortices, respectively, i.e.,

$$B_0 \widetilde{\Omega}_r^{\mathrm{a}} > 0 \tag{79}$$

That is, the rotations of the rigid surface vortices of the micro-MHD flow are reversed. In this model, the chirality of the rigid surface vortices changes but the symmetry breakdown does not occur.

However, the obtained amplification factors $p_r^{\mathrm{a}}$ and $p_f^{\mathrm{a}}$ are not so large that both kinds of vortices would easily fluctuate between both rotational directions, giving rise to the symmetry breaking.

## 5. Materials and Methods

The experiment was carried out in copper electrodeposition in a 300 mol m$^{-3}$ CuSO$_4$ + 500 mol m$^{-3}$ H$_2$SO$_4$ solution. The experimental apparatus was represented elsewhere [6]. Water was prepared by a pure water production system (MERCK KGAA, Darmstadt, Germany). The CuSO$_4$ and H$_2$SO$_4$ were analytical grades (FUJIFILM Wako Pure Chemical Corporation, Osaka, Japan). The VMHDE was made of a copper disk of 8 mm diameter (oxygen-free copper, 99.99% purity, The Nilaco Corporation, Tokyo, Japan) equipped with a 5 mm-wide fringe of PTFE resin (Flonchemical Co. Ltd., Osaka, Japan). To prevent natural convection, it was set in a downward direction. The counter electrode (oxygen-free copper, 99.99% purity, The Nilaco Corporation, Tokyo, Japan) was a copper plate, 25 mm in diameter, which was placed 30 mm from the VMHDE. A copper rod (1 mm diameter) was used as a reference electrode (oxygen-free copper, 99.99% purity, The Nilaco Corporation, Tokyo, Japan). To stop the VMHDF, a sheath with an 18 mm inner diameter and an 18 mm height was attached to the electrode. By using the limiting diffusion current at an overpotential of −400 mV under a given vertical magnetic field, the experiment was performed. The whole electrode system was settled at the place of a uniform magnetic field selected in the bore space of a 10T-cryocooled superconducting magnet (HF-10-100VH, Sumitomo Heavy Industries Ltd., Tokyo, Japan). The deposited electrode surfaces were observed by a surface roughness analysis 3D scanning electron microscope (ERA-8800, ELIONIX Inc., Tokyo, Japan).

## 6. Conclusions

The 2D chiral nuclei are formed under the rigid surface vortices, whose chirality arises from the precessions by the VMHDF. The chiral screw dislocations grow on a chiral 3D nucleus, which in turn develops on a chiral 2D nucleus. Based on these results, the initial five questions are answered as follows.

1. Chiral screw dislocations under a VMHDF arise from the three generations of chiral nuclei, which constitute nesting boxes. Namely, chiral 2D nuclei are formed by the chiral micro-MHD vortices with rigid surfaces. Then, chiral 3D nuclei are created by the chiral nano-MHD vortices with rigid surfaces on a chiral 2D nucleus. Finally, chiral screw dislocations grow by chiral ultra-micro MHD vortices with rigid surfaces on a chiral 3D nucleus. Such a structure was validated by the fact that the observed enantiomeric excess (*ee*) ratios are always smaller than 0.125.
2. The chiral nucleation system is composed of a rotating upper layer and a stationary lower layer so that vortices in the lower layer can receive the precessions from the upper layer and raise chiral nuclei at fixed places.
3. For chirality to emerge, two types of vortices are necessary, having rigid surfaces with friction and free surfaces covered with ionic vacancies. Due to the rigid surface with friction, the rigid surface vortices not only work as pins to stop the migration of the vortices in the lower layer but also create chiral nuclei at fixed positions. Which vortex receives the precession depends on whether the growth mode is unstable or stable. Free surface vortices unstably grow faster than the rigid surface vortices, whereas,

under stable conditions, rigid surface vortices activated dwindle with time more slowly than free surface vortices. Therefore, when unstable, free surface vortices have the priority of precession, and in stable cases, the precessions are donated to rigid surface vortices.

4. Due to fluid and vortex continuities, a pair of adjoining vortices are composed of rigid and free surface vortices with opposite rotations. To raise nuclei fixed to a solid surface, chiral nucleation must occur only under the rigid surface vortices. Since in a $CuSO_4$ + $H_2SO_4$ solution, simple non-specific adsorption takes place, unstable copper nucleation proceeds. As a result, the rotation of a VMHDF transfers to the free surface vortices as the precessions, so that 2D nuclei with reverse chirality are formed under rigid surface vortices in the rotation opposite to that of the VMHDF. Though this result does not directly explain the chiral activity of the electrode, we can understand the mechanism of the emergence of the opposite chirality to the VMHDF. In accordance with the three-generation model, if such a nucleation process were repeated three times, the opposite chirality would be realized.

5. When a chloride additive is added to a $CuSO_4$ + $H_2SO_4$ solution, specific adsorption of the chloride ions takes place, leading to stable nucleation. In this case, the rotation of a VMHDF is bestowed on the rigid surface vortices as precessions. Therefore, 2D nuclei growing under the rigid surface vortices have the same chirality as that of the VMHDF. Namely, due to the stability of the specific adsorption of chloride ions, we can expect a change in the chiral activity of the electrode. However, if the differences between both amplitude factors and their values themselves were sufficiently small, the breakdown would also take place.

**Author Contributions:** Conceptualization, R.A.; methodology, A.S., M.M. (Makoto Miura), Y.O. and R.A.; software, R.M., M.M. (Miki Miura), and R.A.; validation, A.S., M.M. (Makoto Miura), Y.O. and R.A.; formal analysis, R.M., A.S., M.M. (Makoto Miura), Y.O., S.T. and R.A.; investigation, R.M., A.S., M.M. (Makoto Miura), Y.O. and S.T.; resources, Y.O., I.M. and R.A.; data curation, R.M. and R.A.; writing—original draft preparation, R.A.; writing—review and editing, R.M., A.S., M.M. (Makoto Miura) and Y.O.; visualization, R.M., A.S., M.M. (Makoto Miura) and Y.O.; supervision, Y.Y. and R.A.; project administration, Y.O. and R.A. All authors have read and agreed to the published version of the manuscript.

**Funding:** This research was partially supported by the JSPS KAKENHI Grant-in-Aid for Scientific Research (C) no. 19K05230.

**Institutional Review Board Statement:** Not applicable.

**Informed Consent Statement:** Not applicable.

**Data Availability Statement:** Not applicable.

**Acknowledgments:** This work was performed in part at the Queensland node of the Australian National Fabrication Facility, a company established under the National Collaborative Research Infrastructure Strategy, to provide nano and microfabrication facilities for Australia's researchers. The authors thank the staff members of High Field Laboratory for Superconducting Materials of IMR, Tohoku University, for the use of the cryocooled superconducting magnet, and the Tsukuba Magnet Laboratory, National Institute for Materials Science (NIMS), for the use of superconducting magnets.

**Conflicts of Interest:** The authors declare no conflict of interest.

### Nomenclature

| | |
|---|---|
| $x_1$, $x_2$, $x_3$ | Cartesian coordinates corresponding to $x$, $y$, $z$ (m) |
| $x$, $y$, $z$ | non-dimensional coordinates normalized by $d$ |
| $\vec{r}$ | position vector (m) |
| $d$ | representative length (m) |
| $d^a$ | representative length of asymmetrical fluctuations in 2D nucleation (m) |
| $k_x$, $k_y$ | wavenumber components in the $x$- and $y$-directions (m$^{-1}$) |

| | |
|---|---|
| $k$ | wavenumber defined by $\left(k_x^2 + k_y^2\right)^{1/2}$ $(m^{-1})$ |
| $a_x$, $a_y$ | wavenumber components of $a$ in the $x$- and $y$-directions |
| $a$ | non-dimensional wavenumber $(= kd$ or $kd^a)$ |
| $a^+$ | autocorrelation distance of the fluctuation, i.e., the average size of the vortices (m) |
| $\vec{U}$ | velocity which an observer feels $(m\,s^{-1})$ |
| $U_i^*$ | i-component of the main flow velocity of the rotation $(m\,s^{-1})$ |
| $\vec{u}$ | velocity $(m\,s^{-1})$ |
| $u_i$ | i-component of $\vec{u}$ (i = 1, 2, 3) $(m\,s^{-1})$ |
| $u$ | $x$-component of the velocity, $u_1$ $(m\,s^{-1})$ |
| $v$ | $y$-component of the velocity, $u_2$ $(m\,s^{-1})$ |
| $w$ | $z$-component of the velocity, $u_3$ $(m\,s^{-1})$ |
| $\omega_i$ | i-component of the vorticity $(s^{-1})$ |
| $\omega_z$ | $z$-component of the vorticity $(s^{-1})$ |
| $\phi_s$ | $x$-component of stream function $(m\,s^{-1})$ |
| $\psi_s$ | $y$-component of stream function $(m\,s^{-1})$ |
| $P_{int}\left(a_x, a_y\right)$ | Gaussian-type power spectrum defined by Equation (F5) |
| $P_{xz}$ | viscous stress tensor defined in Equation (7a) $(N\,m^{-2})$ |
| $P_{yz}$ | viscous stress tensor defined in Equation (7b) $(N\,m^{-2})$ |
| $\rho$ | density of solution $(kg\,m^{-3})$ |
| $\mu_s$ | viscosity of solution $(N\,s\,m^{-2})$ |
| $\nu$ | kinematic viscosity $(m^2\,s^{-1})$ |
| $\nu^a$ | kinematic viscosity of bulk solution in 2D nucleation |
| $\Omega_m$ | molar volume of deposit metal $(m^3\,mol^{-1})$ |
| $P$ | pressure $(N\,m^{-2})$ |
| $\mu_0$ | magnetic permeability $(4\pi \times 10^7\,N\,A^{-2})$ |
| $\eta$ | resistivity defined by Equation (B14) |
| $\varepsilon$ | dielectric constant of water $(6.95 \times 10^{-10}\,J^{-1}\,C^2\,m^{-1}, 25\,°C)$ |
| $R$ | universal gas constant $(8.31\,J\,K^{-1}\,mol^{-1})$ |
| $T$ | absolute temperature (K) |
| $F$ | Faraday constant $(96{,}500\,C\,mol^{-1})$ |
| $\vec{B}$ | magnetic flux density (T) |
| $B_i$ | i-component of $\vec{B}$ (T) |
| $\vec{B}^*$ | external magnetic flux density in the absence of reactions (T) |
| $B_j^*$ | j-component of $\vec{B}^*$ (T) |
| $B_0$ | $z$-component of $\vec{B}^*$ with sign (T) |
| $\vec{b}$ | fluctuation of $\vec{B}$ by reactions (T) |
| $b_i$ | i-component of $\vec{b}$ (T) |
| $b_z$ | $z$-component of $\vec{b}$ (T) |
| $\vec{E}$ | electric field $(V\,m^{-1})$ |
| $\vec{J}$ | current density $(A\,m^{-2})$ |
| $j_i$ | i-component of the current density fluctuation $(A\,m^{-2})$ |
| $j_z$ | $z$-component of the current density fluctuation $(A\,m^{-2})$ |
| $j_z(x, y, 0, t)^a$ | asymmetrical fluctuation of $j_z$ at the electrode $(A\,m^{-2})$ |
| $\sigma^*$ | electrical conductivity $(S\,m^{-1})$ |
| $z_i$ | charge number of ionic species i including sign |
| $z_m$ | charge number of the metallic ion |
| $\lambda_i^*$ | mobility of ionic species i $(m^2\,V^{-1}\,s^{-1})$ |
| $\lambda_i$ | i-component of unit normal vector |

| | |
|---|---|
| $C_i$ | concentration of ionic species i (mol m$^{-3}$) |
| $D_i$ | diffusion coefficient of ionic species i (m$^2$ s$^{-1}$) |
| $D_m$ | diffusion coefficient of the metallic ion (m$^2$ s$^{-1}$) |
| $\vec{F}_L$ | Lorentz force per unit volume (N m$^{-3}$) |
| $F_{L,i}$ | i-component of $\vec{F}_L$ (N m$^{-3}$) |
| $\vec{F}_R$ | acceleration which an observer feels in a frame of reference rotation with the same angular velocity as the upper layer (N m$^{-3}$) |
| $f_{Ri}$ | i-component of the fluctuation of $\vec{F}_R$ (N m$^{-3}$) |
| $f_{Li}$ | i-component of the fluctuation of the Lorentz force (N m$^{-3}$) |
| $C_m$ | concentration of the metallic ion (mol m$^{-3}$) |
| $C_m^*$ | concentration of the metallic ion in the absence of fluctuation (mol m$^{-3}$) |
| $c_m$ | concentration fluctuation of the metallic ion (mol m$^{-3}$) |
| $c_m(x,y,z,t)^a$ | asymmetrical fluctuation of the concentration of the metallic ion (mol m$^{-3}$) |
| $c_m(x,y,0^+,t)^a$ | $c_m(x,y,z,t)^a$ at OHP (mol m$^{-3}$) |
| $C_m^*(z=0)$ | surface concentration of the metallic ion outside the double layer (mol m$^{-3}$) |
| $C_m^*(z=\infty)$ | bulk concentration of the metallic ion (mol m$^{-3}$) |
| $C_j^*(z=\infty)$ | bulk concentration of ionic species j (mol m$^{-3}$) |
| $L_m$ | average concentration gradient in the diffusion layer defined by Equation (C8) (mol m$^{-4}$) |
| $\theta_\infty^*$ | concentration difference between the bulk and the surface (mol m$^{-3}$) |
| $\langle\delta_c\rangle$ | average thickness of a diffusion layer (m) |
| $U^0$ | amplitude of $u$ (m s$^{-1}$) |
| $V^0$ | amplitude of $v$ (m s$^{-1}$) |
| $W^0$ | amplitude of $w$ (m s$^{-1}$) |
| $\Omega^0$ | amplitude of $\omega_z$ (s$^{-1}$) |
| $W^{0*}$ | real amplitude without i (m s$^{-1}$) |
| $\Omega^{0*}$ | real amplitude without i (s$^{-1}$) |
| $\Phi_s^0$ | amplitudes of the stream functions $\phi_s$ (m s$^{-1}$) |
| $\Psi_s^0$ | amplitudes of the stream functions $\psi_s$ (m s$^{-1}$) |
| $K^0$ | amplitude of $b_z$ (T) |
| $J^0$ | amplitude of $j_z$ (A m$^{-2}$) |
| $\Theta^0$ | amplitude of $c_m$ (mol m$^{-3}$) |
| $W_r^0(z,t)$ | amplitude of $w$ of the rigid surface vortices (m s$^{-1}$) |
| $W_f^0(z,t)$ | amplitude of $w$ of the free surface vortices (m s$^{-1}$) |
| $W_r^0(z,t)^a$ | $W_r^0(z,t)$ in 2D nucleation (m s$^{-1}$) |
| $W_f^0(z,t)^a$ | $W_f^0(z,t)$ in 2D nucleation (m s$^{-1}$) |
| $\Omega_r^0(z,t)$ | amplitude of $\omega_z$ of the rigid surface vortices (s$^{-1}$) |
| $\Omega_f^0(z,t)$ | amplitude of $\omega_z$ of the free surface vortices (s$^{-1}$) |
| $\Omega_r^0(z,t)^a$ | $\Omega_r^0(z,t)$ in 2D nucleation (s$^{-1}$) |
| $\Omega_f^0(z,t)^a$ | $\Omega_f^0(z,t)$ in 2D nucleation (s$^{-1}$) |
| $\Theta_r^0(0,t)$ | amplitude of $c_m$ at the rigid surface (mol m$^{-3}$) |
| $\Theta_f^0(0,t)$ | amplitude of $c_m$ at the free surface (mol m$^{-3}$) |
| $Q$ | magneto-induction coefficient defined by Equation (D4c) |
| $Q^*$ | non-dimensional magneto-induction coefficient defined by Equation (D5a) |
| $\vec{\Omega}$ | angular velocity vector (s$^{-1}$) |
| $\tilde{\Omega}$ | angular velocity of the upper layer corresponding to VMHDF (s$^{-1}$) |



| | |
|---|---|
| $\widetilde{\Omega}_r^a$ | representative angular velocity of the rigid surface vortices $(s^{-1})$ |
| $T^*$ | rotation coefficient defined by Equation (E20c) $(m^{-1})$ |
| $R^*$ | mass transfer coefficient defined by Equation (J2b) $(mol\ m^{-4}\ s)$ |
| $S^*$ | magneto-viscosity coefficient defined by Equation (J9b) $(m^2\ A^{-1}\ s^{-1})$ |
| $R^{*a}$ | $R^*$ in 2D nucleation defined by Equation (G5b) $(mol\ m^{-4}\ s)$ |
| $Q^{*a}$ | $Q^*$ in 2D nucleation defined by Equation (G5c) |
| $T^{*a}$ | $T^*$ in 2D nucleation defined by Equation (G5d) $(m^{-1})$ |
| $S^{*a}$ | $S^*$ in 2D nucleation defined by Equation (G5e) $(m^2\ A^{-1}\ s^{-1})$ |
| $p_r^a$ | amplitude factor of the rigid surface vortices in 2D nucleation defined by Equation (46b) $(s^{-1})$ |
| $p_f^a$ | amplitude factor of the free-surface vortices in 2D nucleation defined by Equation (48b) $(s^{-1})$ |
| $f_r^a(a)$ | amplitude factor function of the rigid surface vortices in 2D nucleation defined by Equation (45b) |
| $f_f^a(a)$ | amplitude factor function of the free surface vortices in 2D nucleation defined by Equation (47b) |
| $\mu_{ad}(x,y,t)$ | chemical potential of the ad-atom $(J\ mol^{-1})$ |
| $\zeta(x,y,t)^a$ | surface morphology of 2D nuclei by the asymmetrical fluctuations (m) |
| $\zeta^a$ | shortened expression of $\zeta(x,y,t)^a$ (m) |
| $\overline{\mu_m}(x,y,\zeta^a,t)$ | electrochemical potential of the metallic ion $(J\ mol^{-1})$ |
| $\overline{\mu_e}(x,y,t)$ | electrochemical potential of the free electron $(J\ mol^{-1})$ |
| $\delta\overline{\mu_m}(x,y,\zeta^a,t)^a$ | asymmetrical fluctuation of $\overline{\mu_m}(x,y,\zeta^a,t)$ $(J\ mol^{-1})$ |
| $\delta\mu_{ad}(x,y,t)$ | asymmetrical fluctuation of $\mu_{ad}(x,y,t)$ $(J\ mol^{-1})$ |
| IHP | inner Helmholtz plane |
| OHP | outer Helmholtz plane |
| $0^+$ | $z$-coordinate of the outer Helmholtz plane (OHP) |
| $\Phi_1$ | overpotential at IHP (V) |
| $\phi_1(x,y,t)^a$ | asymmetrical fluctuation of $\Phi_1$ (V) |
| $\Phi_{2OHP}^*$ | overpotential at the flat OHP without 2D nuclei $(z=0^+)$ measured from the outer boundary of the diffuse layer $(z=\infty^+)$ (V) |
| $\Phi_2$ | overpotential of the diffuse layer (V) |
| $\phi_2(x,y,z,t)^a$ | asymmetrical fluctuation of $\Phi_2$ (V) |
| $\phi_2(x,y,0^+,t)^a$ | asymmetrical fluctuation of $\Phi_2$ at OHP (V) |
| $\phi_2(x,y,\zeta^a,t)^a$ | asymmetrical fluctuation at the surface of 2D nuclei in the diffuse layer (V) |
| $L_{\phi_2}$ | gradient of the electrostatic overpotential in the diffuse layer defined by Equation (35b) $(V\ m^{-1})$ |
| $\lambda$ | Debye length equalized to the average diffuse layer thickness defined by Equation (35c) (m) |
| $L_{m2}$ | average concentration gradient of the metallic ion in the diffuse layer defined by Equation (36b) $(mol\ m^{-4})$ |
| $(\partial\langle\Phi_1\rangle/\partial\langle\Phi_2\rangle)_\mu$ | differential potential coefficient |
| $A_\theta$ | adsorption coefficient defined by Equation (43b) $(s^{-1})$ |
| $\theta_{rand}^a$ | uniform random number between 0 and $2\pi$ |
| $R_d^a$ | 2D random number defined by Equation (49) |
| $\alpha_r^a\left(=\sqrt{2}/2\right)$ | initial ratio of the rigid surface component to the total concentration fluctuation |
| $\alpha_f^a\left(=\sqrt{2}/2\right)$ | initial ratio of the free surface component to the total concentration fluctuation |

| | |
|---|---|
| $\gamma_0^{\mathrm{a}}$ | constant of the vorticity coefficient of the free surface vortex in 2D nucleation defined by Equation (G10b) (s$^{-1}$) |
| $\gamma_1^{\mathrm{a}}$ | constant of the vorticity coefficient of the rigid surface vortex in 2D nucleation defined by Equation (G7b) (s$^{-1}$) |
| $\varepsilon_{\mathrm{screw}}$ | probability that the chiral screw dislocations emerge from all the active points |
| $I_0$ | total current of an electrode covered with only achiral active points (A) |
| $I_{\mathrm{active}}$ | total current of the electrode active for either of D- and L-reagents (A) |
| $I_{\mathrm{inactive}}$ | total current of the electrode for the other reagent (A) |
| $r(ee)$ | enantiomeric excess (*ee*) ratio |
| $Q_1^*$ | electric charge stored in the Helmholtz layer of an electric double layer (A) |
| $Q_2^*$ | electric charge stored in the diffuse layer of an electric double layer (A) |
| $\left(\partial Q_1^*/\partial Q_2^*\right)_\mu$ | differential charge coefficient |
| $C_{\mathrm{H}}$ | electric capacity of the Helmholtz layer (F m$^{-2}$) |
| $\nabla^2$ | $\equiv \partial^2/\partial x_1^2 + \partial^2/\partial x_2^2 + \partial^2/\partial x_3^2$ |
| $\varepsilon_{\mathrm{ijk}}$ | transposition of tensor |
| D | operator defined by d/d$z$ or non-dimensional operator defined by Equation (D5b) |
| $\overline{\mathrm{C}}$ | operator to embed the odd and even functions into a complex space defined by Equation (53a) or Equation (53b) |
| rms | operator defining the root mean square value |
| $\mathrm{g}_1(a)$ | function of *a* defined by Equation (47c) |
| $\mathrm{g}_2(a)$ | function of *a* defined by Equation (47d) |
| $\mathrm{g}_3(a)$ | function of *a* defined by Equation (47e) |
| $\mathrm{g}_4(a)$ | function of *a* defined by Equation (45c) |
| $\mathrm{g}_5(a)$ | function of *a* defined by Equation (45d) |
| $\mathrm{g}_6(a)$ | function of *a* defined by Equation (45e) |
| $\alpha_0, \alpha_1$ | arbitrary constants of the *z*-velocity component of vortices (m s$^{-1}$) |
| $\alpha_2, \alpha_3$ | arbitrary constants of the *z*-velocity component of vortices (m s$^{-1}$) |
| $\alpha_{0\mathrm{r}}^*(a)$ | velocity coefficient of the rigid surface vortices defined by Equation (21b) (m) |
| $\alpha_{1\mathrm{r}}^*(a)$ | velocity coefficient of the rigid surface vortices defined by Equation (22b) (m) |
| $\alpha_{0\mathrm{f}}^*(a)$ | velocity coefficient of the free surface vortices defined by Equation (25b) (m) |
| $\alpha_{1\mathrm{f}}^*(a)$ | velocity coefficient of the free surface vortices defined by Equation (26b) (m) |
| $\beta_0$ | vorticity coefficient of the free surface vortices (s$^{-1}$) |
| $\beta_1$ | vorticity coefficient of the rigid surface vortices (s$^{-1}$) |
| $\beta_0^{\mathrm{a}}$ | vorticity coefficient of the free surface vortices in 2D nucleation (s$^{-1}$) |
| $\beta_1^{\mathrm{a}}$ | vorticity coefficient of the rigid surface vortices in 2D nucleation (s$^{-1}$) |
| Superscript 'a' | implies asymmetrical fluctuation |
| Subscripts 'r' and 'f' | mean rigid surface and free surface components, respectively |
| Subscripts '1' and '2' | imply the Helmholtz and diffuse layers, respectively |

## Appendix A  Stability by the Non-Specific and Specific Adsorption in 2D Nucleation

In electrodeposition, as shown in Figure A1, 2D nucleation arises from an electric double layer. At the inner Helmholtz plane (IHP), dehydrated metallic ions are deposited on the electrode. The potential in the Helmholtz layer resultantly changes, which simultaneously induces the potential change in the diffuse layer. Such a process is represented by the asymmetrical fluctuations of the potentials in the following:

$$\phi_1(x,y,t)^{\mathrm{a}} = \left(\frac{\partial \langle \Phi_1 \rangle}{\partial \langle \Phi_2 \rangle}\right)_{\mu} \phi_2(x,y,z,t)^{\mathrm{a}} \tag{A1}$$

where the subscripts '1' and '2' imply the Helmholtz and diffuse layers, respectively. $\phi_1(x,y,t)^{\mathrm{a}}$ is the overpotential fluctuation at the IHP, and $\phi_2(x,y,z,t)^{\mathrm{a}}$ is the overpotential fluctuation at the outer Helmholtz plane (OHP), where the $z$-coordinate takes the position of the top $(z = \zeta(x,y,t)^{\mathrm{a}})$ or bottom $(z = 0^+)$ of a 2D nucleus at the OHP, and $\zeta(x,y,t)^{\mathrm{a}}$ is a surface height fluctuation at the OHP. $(\partial \langle \Phi_1 \rangle / \partial \langle \Phi_2 \rangle)_{\mu}$ is the differential potential coefficient of the double layer at $\langle \Phi_2 \rangle = \Phi_{2OHP}^*$. $\Phi_{2OHP}^*$ is the equilibrium electrostatic overpotential at the flat OHP $z = 0^+$ measured from the outer boundary of the diffuse layer $(z = \infty^+)$ (See Figure A1a–c), and $\langle \Phi_1 \rangle$ and $\Phi_2$ denote the average electrostatic potential differences at the Helmholtz and diffuse layers, respectively, and the subscript $\mu$ implies the chemical potentials (activities) of all the components are kept constant.

The sign of the potential coefficient $(\partial \langle \Phi_1 \rangle / \partial \langle \Phi_2 \rangle)_{\mu}$ depends on the type of ionic adsorption at IHP. Ionic adsorption is generally classified into two types [22]; one is non-specific adsorption, where polarized solvent molecules and ions are arranged according to the electrostatic force, and the potential monotonously changes with distance, so that $(\partial \langle \Phi_1 \rangle / \partial \langle \Phi_2 \rangle)_{\mu} > 0$ and $\Phi_{2OHP}^* < 0$ are derived (Figure A1a). The other is the specific adsorption, where anions, such as chloride ions, are combined with the electrode surface by strong chemical bindings, which largely shift the potential at OHP to the negative side under the conditions $(\partial \langle \Phi_1 \rangle / \partial \langle \Phi_2 \rangle)_{\mu} < -1$ and $\Phi_{2OHP}^* < 0$ (Figure A1b). Specific adsorption would also be possible in the case of cation, such as hydrogen ion, where the chemical bonding is not so strong that the potential distribution may draw a weak maximum at OHP, so that $-1 < (\partial \langle \Phi_1 \rangle / \partial \langle \Phi_2 \rangle)_{\mu} < 0$ and $\Phi_{2OHP}^* > 0$ are fulfilled (Figure A1c). Therefore, non-specific and specific adsorptions are characterized by the signs of the differential potential coefficient; namely, $(\partial \langle \Phi_1 \rangle / \langle \partial \Phi_2 \rangle)_{\mu} > 0$ and $(\partial \langle \Phi_1 \rangle / \partial \langle \Phi_2 \rangle)_{\mu} < 0$, respectively. In Figure A1d, the relationship between $\langle \Phi_1 \rangle$ and $\langle \Phi_2 \rangle$ is schematically exhibited, where $\langle \Phi_1 \rangle$ is a function of $\langle \Phi_2 \rangle$, and the differential potential coefficient $(\partial \langle \Phi_1 \rangle / \partial \langle \Phi_2 \rangle)_{\mu}$ is the slope of the tangent at $\langle \Phi_2 \rangle = \Phi_{2OHP}^*$.

Under a constant thickness of the Helmholtz layer, we can derive the potential change between the top and bottom of a 2D nucleus in the diffuse layer. By expanding with respect to the $z$-coordinate at the flat OHP, $z = 0^+$, the following equation is obtained:

$$\Delta \phi_2(x,y,\zeta^{\mathrm{a}},t)^{\mathrm{a}} = L_{\phi_2} \zeta(x,y,t)^{\mathrm{a}} \tag{A2a}$$

where $\Delta \phi_2(x,y,\zeta^{\mathrm{a}},t)^{\mathrm{a}}$ is defined by the potential change at the OHP between the top and bottom of the 2D nucleus as follows.

$$\Delta \phi_2(x,y,\zeta^{\mathrm{a}},t)^{\mathrm{a}} \equiv \phi_2(x,y,\zeta^{\mathrm{a}},t)^{\mathrm{a}} - \phi_2(x,y,0^+,t)^{\mathrm{a}} \tag{A2b}$$

$L_{\phi_2}$ is the gradient of the electrostatic equilibrium overpotential of the diffuse layer defined by [22]

$$L_{\phi_2} \equiv -\frac{\Phi_{2OHP}^*}{\lambda} \tag{A3}$$

where $\lambda$ is the Debye length equalized to the average diffuse layer thickness.

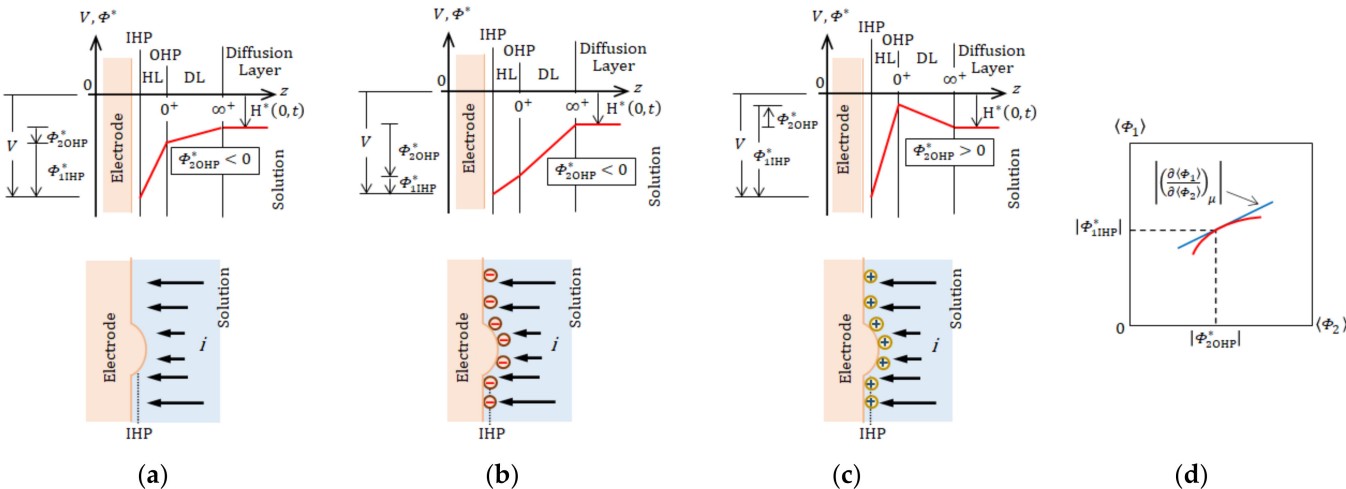

**Figure A1.** The 2D nucleation in an electric double layer [22]. (**a**) Non-specific adsorption. (**b**) Anionic specific adsorption. (**c**) Cationic specific adsorption. (**d**) Schematic view of the relationship between $\langle \Phi_1 \rangle$ and $\langle \Phi_2 \rangle$. $z = 0^+$, the coordinate of OHP; $z = \infty^+$, the outer boundary coordinate of the diffuse layer; $\ominus$, anion; $\oplus$, cation; HL; Helmholtz layer, DL; diffuse layer, $H^*(0,t)$; the equilibrium concentration overpotentials. Reproduced with permission from Morimoto, R.; Miura, M.; Sugiyama, A.; Miura, M.; Oshikiri, Y.; Kim, Y.; Mogi, I.; Takagi, S.; Yamauchi, Y.; Aogaki, R., *The Journal of Physical Chemistry B*; published by the American Chemical Society, 2020.

$$\lambda \equiv \left( \frac{\varepsilon R T}{F^2 \sum_{j \neq m} z_j^2 C_j (z = \infty)} \right)^{\frac{1}{2}} \tag{A4}$$

where $\varepsilon$ is the dielectric constant (F m$^{-1}$), $R$ is the universal gas constant (8.31 J K$^{-1}$ mol$^{-1}$), and $T$ is an absolute temperature (K). $z_j$ is the charge number, including the sign, and $C_j(z = \infty)$ is the bulk concentration of ionic species j except for the bulk metallic concentration $C_m(z = \infty)$ (mol m$^{-3}$) [66]. Substituting Equation (A3) into Equation (A2a), we have

$$\Delta \phi_2(x, y, \zeta^a, t)^a = -\frac{\Phi^*_{2\text{OHP}}}{\lambda} \zeta(x, y, t)^a \tag{A5a}$$

From Equation (A1), the potential change at the IHP is given by

$$\Delta \phi_1(x, y, t)^a = \left( \frac{\partial \langle \Phi_1 \rangle}{\partial \langle \Phi_2 \rangle} \right)_\mu \Delta \phi_2(x, y, \zeta^a, t)^a \tag{A5b}$$

By adding Equation (A5a) to Equation (A5b), the total potential change $\Delta \phi_0(x, y, \zeta^a, t)^a$ of the double layer between the top and bottom of the 2D nucleus is expressed by

$$\Delta \phi_0(x, y, \zeta^a, t)^a = -\frac{1}{\lambda} \left\{ \left( \frac{\partial \langle \Phi_1 \rangle}{\partial \langle \Phi_2 \rangle} \right)_\mu + 1 \right\} \Phi^*_{2\text{OHP}} \zeta(x, y, t)^a \tag{A6}$$

By means of Equation (A6), we can determine whether 2D nucleation is stable or not. When 2D nucleation has a tendency that the reaction resistance increases, suppressing the reaction, we can say it is stable. So, 2D nuclei are kept in flat shapes without growth, distributed randomly on the electrode. On the contrary, in the case where the resistance decreases, the nucleation turns unstable, and 2D nuclei deterministically grow at fixed sites. At the early stage of cathodic deposition neglecting concentration overpotential, the

positive change in the total double layer potential decreases the reaction resistance, so that the unstable condition is expressed by

$$\Delta\phi_0(x, y, \zeta^a, t)^a > 0 \tag{A7}$$

Therefore, $\Delta\phi_0(x, y, \zeta^a, t)^a = 0$ provides the critical condition for the neutral stability. Because the nucleation is expressed by a positive surface deformation, $\zeta(x, y, t)^a > 0$, from Equations (A6) and (A7), $\left\{ (\partial\langle\Phi_1\rangle/\partial\langle\Phi_2\rangle)_\mu + 1 \right\}\Phi^*_{2OHP} < 0$ corresponds to the unstable condition in Equation (A7). The condition $(\partial\langle\Phi_1\rangle/\partial\langle\Phi_2\rangle)_\mu = -1$ is impossible to completely realize, so that $\Phi^*_{2OHP} = 0$, i.e., a flat potential distribution in the diffuse layer, gives the critical condition. As examined in Figure A1a, at the early stage of the electrodeposition without specific adsorption, $(\partial\langle\Phi_1\rangle/\partial\langle\Phi_2\rangle)_\mu + 1 > 0$ and $\Phi^*_{2OHP} < 0$ are automatically fulfilled, so that 2D nucleation without specific adsorption is always unstable. However, according to Figure A1b, by adding anions of intense specific adsorption, such as chloride ion, we can expect that the potential distribution in the electric double layer changes to $(\partial\langle\Phi_1\rangle/\partial\langle\Phi_2\rangle)_\mu + 1 < 0$ and $\Phi^*_{2OHP} < 0$, so that 2D nucleation turns stable, leveling the deposit surface. Though the differential potential coefficient is negative, cationic specific adsorption of the hydrogen ion is not so strong, i.e., $-1 < (\partial\langle\Phi_1\rangle/\partial\langle\Phi_2\rangle)_\mu < 0$, as discussed in Figure A1c, $(\partial\langle\Phi_1\rangle/\partial\langle\Phi_2\rangle)_\mu + 1 > 0$ and $\Phi^*_{2OHP} > 0$ are derived. Namely, according to Equation (A7), hydrogen ion adsorption also makes the early 2D nucleation stable.

In the summary of above discussion, at the early stage of deposition, the unstable condition of the 2D nucleation without specific adsorption of the ion is

$$\left\{ \left(\frac{\partial\langle\Phi_1\rangle}{\partial\langle\Phi_2\rangle}\right)_\mu + 1 \right\}\Phi^*_{2OHP} < 0 \text{(unstable)} \tag{A8a}$$

The stable condition with specific adsorption of the ion is

$$\left\{ \left(\frac{\partial\langle\Phi_1\rangle}{\partial\langle\Phi_2\rangle}\right)_\mu + 1 \right\}\Phi^*_{2OHP} > 0 \text{(stable)} \tag{A8b}$$

As for ionic adsorption, we can provide the following conditions.

$$\left(\frac{\partial\langle\Phi_1\rangle}{\partial\langle\Phi_2\rangle}\right)_\mu \Phi^*_{2OHP} < 0 \quad \text{for non–specific adsorption} \tag{A9a}$$

$$\left(\frac{\partial\langle\Phi_1\rangle}{\partial\langle\Phi_2\rangle}\right)_\mu \Phi^*_{2OHP} > 0 \quad \text{for specific adsorption} \tag{A9b}$$

Namely, in the early stage of deposition, 2D nucleation is unstable for non-specific adsorption, whereas for specific adsorption, it is kept stable.

### Appendix B Basic MHD Equations in the Stationary Lower Layer

In the stationary lower layer, due to the conservation of angular momentum and the pinning effect of the microscopic vortices on the rigid surfaces, all the vortices belonging to the same area keep their positions constant without migration. Then, first, consider explicitly the inertial frame with a static magnetic field. Because the sizes of fluctuations are much smaller than the belonging area, the Cartesian coordinate system $(x, y, z)$ is taken for the area.

With the displacement current ignored [65], Maxwell's equations are

$$\nabla \cdot \vec{B} = 0 \tag{B1}$$

$$\nabla \times \vec{B} = \mu_0 \vec{J} \tag{B2}$$

$$\nabla \times \vec{E} = -\frac{\partial \vec{B}}{\partial t} \tag{B3}$$

where $\vec{E}$ and $\vec{B}$ are the electric field strength (V m$^{-1}$) and the magnetic flux density (T), $\vec{J}$ is the current density (A m$^{-2}$), and $\mu_0$ is the magnetic permeability ($4\pi \times 10^{-7}$ N A$^{-2}$). The overall current density $\vec{J}$ flows under a magnetic flux density $\vec{B}$, so that the Lorentz force per unit volume is generated in the following,

$$\vec{F}_{L} = \vec{J} \times \vec{B} \tag{B4}$$

Substituting for $\vec{J}$ from Equation (B2) in Equation (B4), we obtain

$$\vec{F}_{L} = \frac{1}{\mu_0}\left(\nabla \times \vec{B}\right) \times \vec{B} \tag{B5}$$

By using Equation (B1), an alternative form $\vec{F}_{L}$ in the tensor notation is

$$F_{Li} = -\frac{\partial}{\partial x_i}\left(\frac{\left|\vec{B}\right|^2}{2\mu_0}\right) + \frac{\partial}{\partial x_k}\left(\frac{1}{\mu_0}B_i B_k\right) \tag{B6}$$

Then, consider an incompressible fluid at a uniform temperature, so that the basic equations are given in the following. The momentum equation is

$$\frac{\partial u_i}{\partial t} + u_j\frac{\partial u_i}{\partial x_j} - \frac{B_j}{\rho\mu_0}\frac{\partial B_i}{\partial x_j} = \nu\nabla^2 u_i - \frac{\partial}{\partial x_i}\left(\frac{P}{\rho} + \frac{\left|\vec{B}\right|^2}{2\mu_0\rho}\right) \tag{B7}$$

where $u_i$ is the velocity component (m s$^{-1}$) (i = 1, 2, 3), and the coordinate (m) $(x, y, z)$ is expressed by $(x_1, x_2, x_3)$. $\nu$ and $\rho$ are the kinematic viscosity (m$^2$ s$^{-1}$) and the density (kg m$^{-3}$), respectively. In view of an incompressible fluid, the continuity is held.

$$\frac{\partial u_i}{\partial x_i} = 0 \tag{B8}$$

If a fluid element has a velocity $\vec{u}$, the electric field it will experience is not $\vec{E}$, as measured by a stationary observer, but $\vec{E} + \vec{u} \times \vec{B}$. In an electrolytic solution, the electricity is carried by the diffusion as well as conductivity of ionic species, so that the current density will be given by

$$\vec{J} = \sigma^*\left(\vec{E} + \vec{u} \times \vec{B}\right) - F\sum_i z_i D_i \nabla C_i \tag{B9}$$

where $\sigma^*$ is the electrical conductivity (S m$^{-1}$) defined by

$$\sigma^* = F^2 \sum_i z_i^2 \lambda_i^* C_i \tag{B10}$$

where $z_i$ is the charge number, including sign, $\lambda_i^*$ is the mobility (m$^2$ V$^{-1}$ s$^{-1}$), $F$ is Faraday constant (96,500 C mol$^{-1}$), $C_i$ is the concentration of the ionic species i (mol m$^{-3}$), and $D_i$

is the diffusion constant ($m^2 s^{-1}$). Substitution for $\vec{J}$ from Equation (B2) in Equation (B9) leads to

$$\vec{E} = \frac{1}{\sigma^* \mu_0} \nabla \times \vec{B} - \vec{u} \times \vec{B} + \frac{1}{\sigma^*} F \sum_i z_i D_i \nabla C_i \tag{B11}$$

From a formula in the vector analysis, the curl of a potential gradient is equal to zero, so that the third term on the right-hand side of Equation (B11) disappears.

$$\nabla \times \vec{E} = \nabla \times \left( \eta \nabla \times \vec{B} \right) - \nabla \times \left( \vec{u} \times \vec{B} \right) \tag{B12}$$

After substituting Equation (B12) into Equation (B3), we finally derive

$$\frac{\partial \vec{B}}{\partial t} - \nabla \times \left( \vec{u} \times \vec{B} \right) = -\nabla \times \left( \eta \nabla \times \vec{B} \right) \tag{B13}$$

where $\eta$ is the resistivity defined by

$$\eta \equiv \frac{1}{\sigma^* \mu_0} \tag{B14}$$

From a formula in the vector analysis,

$$\nabla \times \left( \nabla \times \vec{B} \right) = \nabla \left( \nabla \cdot \vec{B} \right) - \nabla^2 \vec{B} \tag{B15a}$$

is held, so that from Equation (B1),

$$\nabla \times \left( \nabla \times \vec{B} \right) = -\nabla^2 \vec{B} \tag{B15b}$$

Substituting Equation (B15b) into Equation (B13), the equation of the magnetic flux density is rewritten in the tensor notation.

$$\frac{\partial B_i}{\partial t} + \frac{\partial}{\partial x_j} \left( u_j B_i - u_i B_j \right) = \eta \nabla^2 \vec{B} \tag{B16}$$

where $\nabla^2$ implies $\partial^2 / \partial x_1^2 + \partial^2 / \partial x_2^2 + \partial^2 / \partial x_3^2$.

Finally, the mass transfer equation of metallic ion is given by

$$\frac{\partial C_m}{\partial t} + \left( \vec{u} \cdot \nabla \right) C_m = D_m \nabla^2 C_m \tag{B17}$$

where subscript 'm' implies the metallic ion.

**Appendix C  Non-Equilibrium Fluctuations Activated in the Stationary Lower Layer**

As the reaction proceeds, the magnetic flux density first fluctuates, expressed by

$$\vec{B} = \vec{B}^* + \vec{b} \tag{C1}$$

where $\vec{B}^*$ is the external magnetic flux density (T) in the absence of the reaction, and $\vec{b}$ is the fluctuation (T) by the reaction. The fluctuation of the Lorentz force is written as

$$f_{Li} = \frac{\partial}{\partial x_i} \left( \frac{\vec{b} \cdot \vec{B}^*}{\mu_0} \right) + B_j^* \frac{\partial}{\partial x_j} \left( \frac{b_i}{\mu_0} \right) \tag{C2}$$

According to Equation (C2), Equation (B7) is rewritten as

$$\frac{\partial u_i}{\partial t} + u_j \frac{\partial u_i}{\partial x_j} - \frac{B_j^*}{\rho \mu_0} \frac{\partial b_i}{\partial x_j} = \nu \nabla^2 u_i - \frac{\partial}{\partial x_i} \delta \xi \tag{C3a}$$

where $B_j^*$ and $b_i$ are the j-component of $\vec{B}^*$ and the i-component of $\vec{b}$, respectively. In view of the initial stationary state, $u_i$ is the velocity fluctuation component activated by the Lorentz force fluctuation. The second order smallness $u_i(\partial u_i / \partial x_j)$ is disregarded, and

$$\delta \xi \equiv \frac{\delta P}{\rho} + \frac{\vec{B}^* \cdot \vec{b}}{\rho \mu_0} \tag{C3b}$$

where $\delta P$ denotes the pressure fluctuation.

As a result, the continuity equation, Equation (B8), remains the same. The corresponding fluctuation forms of Equations (B1) and (B16) are supplied by

$$\frac{\partial b_i}{\partial x_i} = 0 \tag{C4}$$

and

$$\frac{\partial b_i}{\partial t} = B_j^* \frac{\partial u_i}{\partial x_j} + \eta \nabla^2 b_i \tag{C5}$$

The concentration of the metallic ion is expressed by

$$C_m = C_m^* + c_m \tag{C6}$$

where $C_m^*$ and $c_m$ are the concentration in the absence of fluctuation (mol m$^{-3}$) and the concentration fluctuation (mol m$^{-3}$), respectively. The mass transfer equation, Equation (B17), is also rewritten as

$$\frac{\partial c_m}{\partial t} + w L_m = D_m \nabla^2 c_m \tag{C7}$$

where $w$ denotes the z-component of the velocity $u_3$, and $L_m$ is the average concentration gradient.

$$L_m \equiv \frac{\theta_\infty^*}{\langle \delta_c \rangle} \tag{C8}$$

where $\theta_\infty^*$ denotes the concentration difference between the bulk and surface, and $\langle \delta_c \rangle$ is the average diffusion layer thickness (m).

In the tensor notation, the i-component of the current density fluctuation is expressed from Equation (B2) as

$$j_i = \frac{1}{\mu_0} \varepsilon_{ijk} \frac{\partial}{\partial x_j} b_k \tag{C9}$$

where $\varepsilon_{ijk}$ implies the transposition of the tensor. The vorticity $\omega_i$ is given by

$$\omega_i = \varepsilon_{ijk} \frac{\partial}{\partial x_j} u_k \tag{C10}$$

After applying $\varepsilon_{ijk} \partial / \partial x_j$ to the k-component of Equation (C3a), we have

$$\frac{\partial \omega_i}{\partial t} = \nu \nabla^2 \omega_i + \frac{B_j^*}{\rho} \frac{\partial j_i}{\partial x_j} \tag{C11}$$

where

$$\varepsilon_{ijk} \frac{\partial}{\partial x_j} \frac{\partial}{\partial x_k} \delta \xi = \left( \frac{\partial}{\partial x_j} \frac{\partial}{\partial x_k} - \frac{\partial}{\partial x_k} \frac{\partial}{\partial x_j} \right) \delta \xi = 0 \tag{C12}$$

Taking the curl of Equation (C11), we have

$$\frac{\partial}{\partial t}\nabla^2 u_i = \nu\nabla^4 u_i + \frac{B_j^*}{\rho\mu_0}\frac{\partial}{\partial x_j}\nabla^2 b_i \tag{C13}$$

where the following relationships are used:

$$\varepsilon_{ijk}\frac{\partial}{\partial x_j}\omega_k = -\nabla^2 u_i \tag{C14a}$$

$$\varepsilon_{ijk}\frac{\partial}{\partial x_j}j_k = -\frac{1}{\mu_0}\nabla^2 b_i \tag{C14b}$$

and from Equation (C10), we have

$$\varepsilon_{ijk}\frac{\partial}{\partial x_j}u_k = \omega_i \tag{C15}$$

To derive Equations (C14a) and (C14b), the formula

$$\varepsilon_{ijk}\varepsilon_{klm} = \delta_{il}\delta_{jm} - \delta_{im}\delta_{jl} \tag{C16}$$

is used. Then, the curl of Equation (C5), together with Equations (C9) and (C10), leads to the following equation.

$$\frac{\partial j_i}{\partial t} = \frac{B_j^*}{\mu_0}\frac{\partial\omega_i}{\partial x_j} + \eta\nabla^2 j_i \tag{C17}$$

To extract the *z*-components from Equations (C5), (C11), (C13), and (C17), the unit normal vector $\lambda_i$ is multiplied to them.

$$\frac{\partial b_z}{\partial t} = \eta\nabla^2 b_z + B_j^*\frac{\partial w}{\partial x_j} \tag{C18a}$$

$$\frac{\partial j_z}{\partial t} = \eta\nabla^2 j_z + \frac{B_j^*}{\mu_0}\frac{\partial\omega_z}{\partial x_j} \tag{C18b}$$

$$\frac{\partial\omega_z}{\partial t} = \nu\nabla^2\omega_z + \frac{B_j^*}{\rho}\frac{\partial j_z}{\partial x_j} \tag{C18c}$$

$$\frac{\partial}{\partial t}\nabla^2 w = \nu\nabla^4 w + \frac{B_j^*}{\rho\mu_0}\frac{\partial}{\partial x_j}\nabla^2 b_z \tag{C18a}$$

where $b_z$, $j_z$, $\omega_z$, and $w$ denote the *z*-components of $\vec{b}$, $\vec{j}$, $\vec{\omega}$, and $\vec{u}$, respectively. Equations (C18a), (C18b), (C18c), and (C18d) describe the electromagnetic induction by the velocity of the microscopic vortices, electromagnetic induction by the rotation of the vortices, the rotation induced by the Lorentz force, and the velocity of the vortices induced by the electromagnetic induction. Then, Equation (C7) expresses the mass transfer enhanced by them.

We shall restrict our discussion of this problem to the case where magnetic flux density is imposed vertically to the electrode.

$$\vec{B}^* = (0, 0, B_0) \tag{C19}$$

Therefore, we obtain

$$\frac{\partial b_z}{\partial t} = \eta\nabla^2 b_z + B_0\frac{\partial w}{\partial z} \tag{C20a}$$

$$\frac{\partial j_z}{\partial t} = \eta \nabla^2 j_z + \frac{B_0}{\mu_0} \frac{\partial \omega_z}{\partial z} \tag{C20b}$$

$$\frac{\partial \omega_z}{\partial t} = \nu \nabla^2 \omega_z + \frac{B_0}{\rho} \frac{\partial j_z}{\partial z} \tag{C20c}$$

$$\frac{\partial}{\partial t} \nabla^2 w = \nu \nabla^4 w + \frac{B_0}{\rho \mu_0} \frac{\partial}{\partial z} \nabla^2 b_z \tag{C20d}$$

The mass transfer equation is still given by Equation (C7).

## Appendix D  Derivation of the Amplitude Equations of the Fluctuations in the Stationary Lower Layer

For the fluctuations, we assume the following 2D plane waves.

$$w = W^0(z,t) \exp\left[i\left(k_x x + k_y y\right)\right] \tag{D1a}$$

$$\omega_z = \Omega^0(z,t) \exp\left[i\left(k_x x + k_y y\right)\right] \tag{D1b}$$

$$b_z = K^0(z,t) \exp\left[i\left(k_x x + k_y y\right)\right] \tag{D1c}$$

$$j_z = J^0(z,t) \exp\left[i\left(k_x x + k_y y\right)\right] \tag{D1d}$$

$$c_m = \Theta^0(z,t) \exp\left[i\left(k_x x + k_y y\right)\right] \tag{D1e}$$

where $W^0(z,t)$, $\Omega^0(z,t)$, $K^0(z,t)$, $J^0(z,t)$, and $\Theta^0(z,t)$ are the amplitudes of the fluctuations, and $k_x$ and $k_y$ are the wavenumbers in the $x$- and $y$-directions, respectively.

Substituting Equations (D1a)–(D1e) into Equations (C20a)–(C20d) and Equation (C7), we have

$$\left(D^2 - k^2 - \frac{1}{\eta}\frac{\partial}{\partial t}\right) K^0 = -\left(\frac{B_0}{\eta}\right) DW^0 \tag{D2a}$$

$$\left(D^2 - k^2 - \frac{1}{\eta}\frac{\partial}{\partial t}\right) J^0 = -\left(\frac{B_0}{\mu_0 \eta}\right) D\Omega^0 \tag{D2b}$$

$$\left(D^2 - k^2 - \frac{1}{\nu}\frac{\partial}{\partial t}\right) \Omega^0 = -\left(\frac{B_0}{\rho \nu}\right) DJ^0 \tag{D2c}$$

$$\left(D^2 - k^2\right)\left(D^2 - k^2 - \frac{1}{\nu}\frac{\partial}{\partial t}\right) W^0 = -\left(\frac{B_0}{\mu_0 \rho \nu}\right) D\left(D^2 - k^2\right) K^0 \tag{D2d}$$

$$\left(D^2 - k^2 - \frac{1}{D_m}\frac{\partial}{\partial t}\right) \Theta^0 = \left(\frac{L_m}{D_m}\right) W^0 \tag{D2e}$$

where $D \equiv \partial/\partial z$ and $k \equiv \left(k_x^2 + k_y^2\right)^{1/2}$. Since the fluctuations are at a quasi-steady state, neglecting the time-differential terms, we have

$$\left(D^2 - k^2\right) K^0 = -\left(\frac{B_0}{\eta}\right) DW^0 \tag{D3a}$$

$$\left(D^2 - k^2\right) J^0 = -\left(\frac{B_0}{\mu_0 \eta}\right) D\Omega^0 \tag{D3d}$$

$$\left(D^2 - k^2\right) \Omega^0 = -\left(\frac{B_0}{\rho \nu}\right) DJ^0 \tag{D3c}$$

$$\left(D^2 - k^2\right)^2 W^0 = -\left(\frac{B_0}{\mu_0 \rho \nu}\right) D\left(D^2 - k^2\right) K^0 \tag{D3d}$$

$$\left(D^2 - k^2\right)\Theta^0 = \left(\frac{L_m}{D_m}\right) W^0 \tag{D3e}$$

Substituting Equation (D3b) into Equation (D3c), and using Equation (B14), we have

$$\left\{\left(D^2 - k^2\right)^2 - QD^2\right\}\Omega^0 = 0 \tag{D4a}$$

Then, substitution of Equation (D3a) into Equation (D3d) leads to

$$\left\{\left(D^2 - k^2\right)^2 - QD^2\right\}W^0 = 0 \tag{D4b}$$

where $Q$ implies the magneto-induction coefficient, expressed by

$$Q \equiv \frac{\sigma^* B_0^2}{\rho \nu} \tag{D4c}$$

Here, we introduce a representative length $d$. Then, let $a = kd$ be the wavenumber in the non-dimensional unit. We shall, however, let $x$, $y$, and $z$ stand for the non-dimensional coordinates normalized by $d$, so that the following parameter $Q$ and operator D are changed as follows.

$$Q^* \equiv \frac{\sigma^* B_0^2 d^2}{\rho \nu}\left(= Qd^2\right) \tag{D5a}$$

$$D \equiv \frac{d}{dz}\left(= Dd\right) \tag{D5b}$$

where the coordinate $z$ is in the new unit of length $d$. Equations (D4a) and (D4b) are rewritten as

$$\left\{\left(D^2 - a^2\right)^2 - Q^* D^2\right\}\Omega^0 = 0 \tag{D6a}$$

$$\left\{\left(D^2 - a^2\right)^2 - Q^* D^2\right\}W^0 = 0 \tag{D6b}$$

As shown in Equations (D6a) and (D6b), $\Omega^0$ and $W^0$ are independent of each other. This means that the $z$-component of the vorticity does not interact with the $z$-component of velocity as they are. Both equations are unrelated with external rotations.

**Appendix E  Microscopic Vortices Induced in the Rotating Upper Layer**

　　　The rotating upper layer acts as a reservoir of the vortices in the lower layer. In the lower layer, due to the pinning effect of downward vortices on the rigid surfaces, the downward and upward vortices are regularly fixed at the same positions. Through the boundary between the upper and lower layers, vortices with the same velocities and vorticities are newly induced in the upper layer, similar to miller images, covering the same area. Such a process, as shown in Figure 4a, forms a positive feedback cycle.

　　　Due to the low electric conductivity of electrolyte solutions, electromagnetic induction is neglected, so that we only think of the effects of the Coriolis force and centrifugal force. Here, let us consider a rotating incompressible fluid accompanied with microscopic vortices. As shown in Figure 4b, an observer at rest on a frame of reference rotating with the same angular velocity recognizes two kinds of acceleration [65], i.e.,

$$\vec{F}_R = 2\vec{\Omega} \times \vec{U} - \frac{1}{2}\nabla\left(\left|\vec{\Omega} \times \vec{r}\right|^2\right) \tag{E1}$$

where $\vec{\Omega}$ is the vector of the angular velocity of the rotating upper layer $(s^{-1})$, $\vec{U}$ is the vector of the velocity $(m\ s^{-1})$, and $\vec{r}$ is the vector of position (m). The term $2\vec{\Omega} \times \vec{U}$ represents the Coriolis acceleration and the term $-(1/2)\nabla\left(\left|\vec{\Omega} \times \vec{r}\right|^2\right)$ is the centrifugal force.

The momentum equation is expressed by

$$\frac{\partial U_i}{\partial t} + U_j\frac{\partial U_i}{\partial x_j} = 2\varepsilon_{ijk}U_j\Omega_k + \nu\nabla^2 U_i - \frac{\partial}{\partial x_i}\left(\frac{P}{\rho} - \frac{1}{2}\left|\vec{\Omega},\times,\vec{r}\right|^2\right) \quad \text{for } i = 1, 2, 3 \quad \text{(E2)}$$

In view of the incompressible fluid, the continuity equation is derived.

$$\frac{\partial U_i}{\partial x_i} = 0 \tag{E3}$$

The velocity is expressed by the rotational component $U_i^*$ and the vortex component $u_i$, i.e.,

$$U_i = U_i^* + u_i \tag{E4a}$$

Since the observer is rotating with the upper layer, the rotational component is zero.

$$U_i^* = 0 \tag{E4b}$$

According to the activation, the acceleration in Equation (E1) fluctuates in the following,

$$f_{Ri} = 2\varepsilon_{ijk}U_j\Omega_k - \frac{1}{2}\frac{\partial}{\partial x_i}\left(\left|\vec{\Omega} \times \vec{r}\right|^2\right) \tag{E5}$$

where the first and second terms on the right-hand side of Equation (E5) denote the contributions of the Coriolis and centrifugal forces, respectively, where the second term is equal to zero without fluctuation. The momentum equation of the micro-MHD flow is written by

$$\frac{\partial u_i}{\partial t} = 2\varepsilon_{ijk}u_j\Omega_k + \nu\nabla^2 u_i - \frac{\partial}{\partial x_i}\left(\frac{\delta P}{\rho}\right) \tag{E6}$$

where the second order of smallness $u_j\left(\partial u_i/\partial x_j\right)$ is disregarded. Equation (E3) is also rewritten by

$$\frac{\partial u_i}{\partial x_i} = 0 \tag{E7}$$

The potential gradient $\partial/\partial x_i(\delta P/\rho)$ in Equation (E6) can be eliminated by applying the operator $\varepsilon_{ijk}\partial/\partial x_j$, i.e., taking a curl of Equation (E6).

$$\varepsilon_{ijk}\frac{\partial}{\partial x_j}\cdot\frac{\partial}{\partial x_k}\left(\frac{\delta P}{\rho}\right) = \left(\frac{\partial}{\partial x_j}\cdot\frac{\partial}{\partial x_k} - \frac{\partial}{\partial x_k}\cdot\frac{\partial}{\partial x_j}\right)\left(\frac{\delta P}{\rho}\right) = 0 \tag{E8}$$

Using Equations (C16) and (E7), we obtain

$$\varepsilon_{ijk}\frac{\partial}{\partial x_j}\cdot\varepsilon_{klm}u_l\Omega_m = \Omega_j\frac{\partial u_i}{\partial x_j} \tag{E9}$$

Taking the curl of (E6) and using Equations (E8) and (E9), we obtain

$$\frac{\partial \omega_i}{\partial t} = \nu\nabla^2\omega_i + 2\Omega_j\frac{\partial u_i}{\partial x_j} \tag{E10}$$

where the vorticity $\omega_i$ is defined by Equation (C10) in Appendix C. In the same way, we have

$$\varepsilon_{ijk}\frac{\partial}{\partial x_j}\omega_k = -\nabla^2 u_i \tag{C14a}$$

and

$$\varepsilon_{ijk}\frac{\partial}{\partial x_j}u_k = \omega_i \tag{C15}$$

Then, taking the curl of Equation (E10), and using Equations (C14a) and (C15), we finally have

$$\frac{\partial}{\partial t}\nabla^2 u_i = \nu\nabla^4 u_i - 2\Omega_j\frac{\partial \omega_i}{\partial x_j} \tag{E11}$$

To extract the *z*-components from Equations (E10) and (E11), multiplying them by the unit normal vector $\lambda_i$, we have

$$\frac{\partial \omega_z}{\partial t} = \nu\nabla^2\omega_z + 2\Omega_j\frac{\partial w}{\partial x_j} \tag{E12}$$

and

$$\frac{\partial}{\partial t}\nabla^2 w = \nu\nabla^4 w - 2\Omega_j\frac{\partial \omega_z}{\partial x_j} \tag{E13}$$

where $\omega_z$ and $w$ denote the *z*-components of $\vec{\omega}$ and $\vec{u}$, respectively.

Considering that a vector of the rotation is an axial vector with *z*-axis, we can write the following notation,

$$\vec{\Omega} \equiv \left(0,\ 0, \widetilde{\Omega}\right) \tag{E14}$$

where $\widetilde{\Omega}$ denotes the angular velocity of the rotating upper layer. Therefore, we have

$$\frac{\partial \omega_z}{\partial t} = \nu\nabla^2\omega_z + 2\widetilde{\Omega}\frac{\partial w}{\partial z} \tag{E15}$$

and

$$\frac{\partial}{\partial t}\nabla^2 w = \nu\nabla^4 w - 2\widetilde{\Omega}\frac{\partial \omega_z}{\partial z} \tag{E16}$$

Substituting Equations (D1a) and (D1b) in Appendix D into Equations (E15) and (E16), we obtain

$$\left(D^2 - k^2 - \frac{1}{\nu}\frac{\partial}{\partial t}\right)\Omega^0 = -\left(\frac{2\widetilde{\Omega}}{\nu}\right)DW^0 \tag{E17}$$

and

$$\left(D^2 - k^2\right)\left(D^2 - k^2 - \frac{1}{\nu}\frac{\partial}{\partial t}\right)W^0 = \left(\frac{2\widetilde{\Omega}}{\nu}\right)D\Omega^0 \tag{E18}$$

where the operator D implies $\partial/\partial z$. Since the fluctuations are in a quasi-steady state, disregarding the time-differential terms, we have

$$\left(D^2 - k^2\right)\Omega^0 = -\left(\frac{2\widetilde{\Omega}}{\nu}\right)DW^0 \tag{E19a}$$

and

$$\left(D^2 - k^2\right)^2 W^0 = \left(\frac{2\widetilde{\Omega}}{\nu}\right)D\Omega^0 \tag{E19b}$$

Let $a = kd$ be the wavenumber in the non-dimensional. We shall, however, let $x, y, z$ stand for the coordinates in the new unit of length $d$. As a result, Equations (E19a) and (E19b) are changed to

$$\left(D^2 - a^2\right)\Omega^0 = -T^*DW^0 \tag{E20a}$$

and

$$\left(D^2 - a^2\right)^2 W^0 = d^2 T^* D\Omega^0 \tag{E20b}$$

where D is defined by the new coordinate $z$ as $d/dz$, and $T^*$ is the rotation coefficient expressed by

$$T^* \equiv \frac{2\widetilde{\Omega}d}{\nu} \tag{E20c}$$

At the upper boundary, the vortices in the lower layer will receive the precessional motions of the vortices revolving with the upper layer shown in Equations (E20a) and (E20b).

**Appendix F  Intrinsic Spectrum of the Asymmetrical Fluctuations in 2D Nucleation**

Asymmetrical fluctuations arise with electrochemical reactions, accompanied by the vortices, i.e., micro-MHD flows. On the solution side, the micro-MHD flows prevail over the fluctuations, so that the spectrum of the fluctuation is controlled by the micro-MHD flows.

For the micro-MHD flows, the autocorrelation distance of the fluctuation $a^+$ is taken as a unit of length $d^a$, which is defined by the average diffusion layer thickness $\langle \delta_c \rangle$ for 2D nucleation [5].

$$a^+ = d^a (\equiv \delta_c) \tag{F1}$$

An asymmetrical concentration fluctuation outside the double layer is given by the difference between the concentration $C_m(x, y, z, t)$ and the bulk concentration $C_m^*(z = \infty)$.

$$c_m(x, y, z, t)^a \equiv C_m(x, y, z, t) - C_m^*(z = \infty) \quad (< 0) \tag{F2}$$

For cathodic deposition, as shown in Figure 6b, it takes negative values. In the case of an unstable deposition, after applying a potential step, the fluctuation at the electrode surface would develop up to its ultimate value, i.e., $-\theta_\infty^*$, where $\theta_\infty^*$ implies the concentration difference between the bulk and surface.

$$\theta_\infty^* \equiv C_m^*(z = \infty) - C_m^*(z = 0) \quad (> 0) \tag{F3}$$

$C_m^*(z = 0)$ is the surface concentration outside the double layer (mol m$^{-3}$). With the normalization of $\theta_\infty^*$, the intrinsic spectrum of the concentration fluctuation controlled by the micro-MHD flow is represented by

$$P_{\text{int}}\left(a_x, a_y\right) \equiv \frac{1}{XY} \frac{\left|\Theta^0(0,0)^a\right|^2}{\theta_\infty^{*2}} \tag{F4}$$

where $X$ and $Y$ are the non-dimensional $x$- and $y$-lengths of an electrode, respectively, and $a_x$ and $a_y$ are non-dimensional wavenumbers in $x$- and $y$-directions, respectively. For the assumption of an isotropic Gaussian distribution with normalization, the spectrum has the following form.

$$P_{\text{int}}\left(a_x, a_y\right) = \frac{1}{\pi} \exp\left(-a^2\right) \tag{F5}$$

where $a^2 \equiv a_x^2 + a_y^2$ is defined. The important role of the spectrum is to determine the upper limits of the amplitude factor functions $f_r^a(a)$ and $f_f^a(a)$ as a spatial filter. From Equations (F4) and (F5), the initial amplitude of the concentration fluctuation is expressed by

$$\left| \Theta^0(0,0)^a \right|^2 = \frac{XY}{\pi} \theta_\infty^{*2} \exp\left( -a^2 \right) \tag{F6a}$$

Then, the average of the amplitude is calculated by

$$\left\langle \left| \Theta^0(0,0)^a \right|^2 = \theta_\infty^{*2} \right\rangle \tag{F6b}$$

## Appendix G  Amplitudes of the Asymmetrical Concentration and Concentration Gradient Fluctuations in 2D Nucleation

In view of the association with all the components of the fluctuations, from Rayleigh's theorem, the mean squares (ms) of the fluctuations concerning the electrode surface are expressed by the mean squares of the amplitudes regarding the wavenumbers.

$$\left\langle \left| c_m(x,y,0,t)^a \right|^2 \right\rangle = \frac{1}{XY} \int_{-\infty}^{\infty} \int_{-\infty}^{\infty} \left| \Theta^0(0,t)^a \right|^2 da_x da_y \left( \equiv \left\langle \left| \Theta^0(0,t)^a \right|^2 \right\rangle \right) \tag{G1}$$

where $X$ and $Y$ are the non-dimensional $x$- and $y$-lengths of the electrode. Since two types of fluctuations on the rigid and free surfaces take part in a reaction, the initial amplitude of the concentration is divided into the rigid and free surface components.

$$\left| \Theta^0(0,0)^a \right|^2 = \left| \Theta_r^0(0,0)^a \right|^2 + \left| \Theta_f^0(0,0)^a \right|^2 \tag{G2a}$$

where subscripts 'r' and 'f' imply the rigid surface and free surface, respectively. Each component is expressed by the total amplitude.

$$\Theta_j^0(0,0)^a = \alpha_j^a \Theta^0(0,0)^a \quad \text{for j = r or f} \tag{G2b}$$

where $\alpha_j^a$ denotes the ratio of each initial component to the total amplitude. Substituting Equation (G2b) into Equation (G2a), we have

$$\alpha_r^{a2} + \alpha_f^{a2} = 1 \tag{G2c}$$

As initially discussed, in the present case, all the fluctuations arise from a stationary state, so that the concentration fluctuations on the rigid and free surfaces as well as the rigid and free surface vortices make equal contributions to the nucleation, i.e., $\alpha_r^a = \alpha_f^a = \sqrt{2}/2$ is derived. The mean square of the concentration gradient fluctuation is defined by

$$\left\langle \left| \left\{ \frac{\partial}{\partial z} c_m(x,y,0,0)^a \right\}_{z=0} \right|^2 \right\rangle = \left( \frac{\theta_\infty^*}{\langle \delta_c \rangle} \right)^2 \left( \equiv \left\langle \left| D\Theta^0(0,0)^a \right|^2 \right\rangle \right) \tag{G3}$$

In the same way as that of Equation (G2a), we derive the following relationship.

$$\left| D\Theta^0(0,0)^a \right|^2 = \left| D\Theta_r^0(0,0)^a \right|^2 + \left| D\Theta_f^0(0,0)^a \right|^2 \tag{G4}$$

Here, $D \equiv d/dz$ is defined, and $z$ stands for the coordinate in the new unit of length $d^a$ shown in Equation (F1).

(a)    For a rigid surface:

With Equations (45c)–(45e), Equation (27c) is rewritten as

$$\Theta_r^0(0,t)^a = -\frac{\beta_1^a\left\{16Q^{*a}T^{*a}g_5(a) + S^{*a}R^{*a}g_6(a)\right\}}{8z_m F D_m Q^{*a} S^{*a} T^{*a} g_4(a)} \tag{G5a}$$

where $\beta_1^a$ represents the vorticity coefficient of the rigid surface vortices in 2D nucleation, which is a function of time. The other coefficients in 2D nucleation are defined as

$$R^{*a} \equiv \frac{L_m d^{a2}}{D_m} \tag{G5b}$$

$$Q^{*a} \equiv \frac{\sigma^* B_0^2 d^{a2}}{\rho \nu^a} \tag{G5c}$$

$$T^{*a} \equiv \frac{2\widetilde{\Omega}d^a}{\nu^a} \tag{G5d}$$

and

$$S^{*a} \equiv \frac{B_0 d^a}{\rho \nu^a} \tag{G5e}$$

$d^a$ and $\nu^a$ are the representative lengths in 2D nucleation and the kinematic viscosity of the bulk solution, respectively. In view of Equation (46a), substitution of Equation (45b) into Equation (G5a) leads to

$$\Theta_r^0(0,t)^a = -\frac{2f_r^a(a)^{-1}}{z_m F D_m S^{*a}}\beta_1^a(t) \tag{G6}$$

where $f_r^a(a)$ denotes the amplitude factor function of the rigid surface vortices in 2D nucleation.

$$f_r^0(a) = \frac{16Q^{*a}g_4(a)}{16Q^{*a}g_5(a) + S^{*a}T^{*a-1}R^{*a}g_6(a)} \tag{45b}$$

Then, substituting Equations (F6a) and (G6) into Equation (G2b), we have

$$\left|\beta_1^a(0)\right| = \left|\gamma_1^a\right|f_r^a(a)\exp\left(-\frac{a^2}{2}\right) \tag{G7a}$$

where the constant of the vorticity coefficient of the rigid surface vortex in the 2D nucleation is expressed by

$$\gamma_1^a \equiv \frac{1}{2}\alpha_r^a\left(\frac{XY}{\pi}\right)^{\frac{1}{2}} z_m F D_m \theta_\infty^* S^{*a} \tag{G7b}$$

Then, substituting Equations (G7a) and (46a) into Equation (G6), we have

$$\left|\Theta_r^0(0,t)^a\right| = \frac{2\left|\gamma_1^a\right|}{z_m F D_m |S^{*a}|}\exp\left(-\frac{a^2}{2}\right)\exp(p_r^a t) \tag{G8a}$$

where $p_r^a$ implies the amplitude factor of the rigid surface vortex. Finally, by substituting Equations (G7a) and (46a) into Equation (27a), we have

$$\left|D\Theta_r^0(0,t)^a\right| = \frac{2\left|\gamma_1^a\right|f_r^a(a)}{z_m F D_m |S^{*a}|}\exp\left(-\frac{a^2}{2}\right)\exp(p_r^a t) \tag{G8b}$$

(b)   For a free surface:

Substitution of Equations (47c)–(47e) into Equation (28c) leads to

$$\Theta_f^0(0,t)^a = \frac{a\beta_0^a\left\{16Q^{*a}T^{*a}g_2(a) + S^{*a}R^{*a}g_3(a)\right\}}{8z_m F D_m Q^{*a} S^{*a} T^{*a} g_1(a)} \tag{G9a}$$

where $\beta_0^a$ implies the vorticity coefficient of the free surface vortices in the 2D nucleation, which is also a function of time. Then, substituting Equation (47b) into Equation (G9a), we have

$$\Theta_f^0(0,t)^a = \frac{2af_f^a(a)^{-1}}{z_m F D_m S^{*a}} \beta_0^a(t) \tag{G9b}$$

where $f_f^a(a)$ represents the amplitude factor function of the free surface vortices in 2D nucleation.

$$f_f^a(a) = \frac{16Q^{*a}g_1(a)}{16Q^{*a}g_2(a) + S^{*a}T^{*a-1}R^{*a}g_3(a)} \tag{47b}$$

Furthermore, we substitute Equation (F6a) and Equation (G9b) into Equation (G2b), and we have

$$|\beta_0^a(0)| = |\gamma_0^a|f_f^a(a)a^{-1}\exp\left(-\frac{a^2}{2}\right) \tag{G10a}$$

where the constant of the vorticity coefficient of the free surface vortex in the 2D nucleation is given by

$$\gamma_0^a \equiv \frac{1}{2}\alpha_f^a\left(\frac{XY}{\pi}\right)^{\frac{1}{2}} z_m F D_m \theta_\infty^* S^{*a} \tag{G10b}$$

Then, substituting Equations (G10a) and (48a) into Equation (G9b), we have

$$\left|\Theta_f^0(0,t)^a\right| = \frac{2|\gamma_0^a|}{z_m F D_m |S^{*a}|}\exp\left(-\frac{a^2}{2}\right)\exp(p_f^a t) \tag{G11a}$$

where $p_f^a$ implies the amplitude factor of the free surface vortex. Finally, by substituting Equations (G10a) and (48a) into Equation (28a), we have

$$\left|D\Theta_f^0(0,t)^a\right| = \frac{2|\gamma_0^a|f_f^a(a)}{z_m F D_m |S^{*a}|}\exp\left(-\frac{a^2}{2}\right)\exp(p_f^a t) \tag{G11b}$$

**Appendix H  Amplitude Equations of $x$- and $y$-Components of the Velocity Fluctuation**

Supposing that the $x$- and $y$-components of the velocity fluctuations $u$ and $v$ are expressed by the stream functions $\phi_s$ and $\psi_s$, we have the following equations,

$$u = \frac{\partial \phi_s}{\partial x} - \frac{\partial \psi_s}{\partial y} \tag{H1}$$

and

$$v = \frac{\partial \phi_s}{\partial y} + \frac{\partial \psi_s}{\partial x} \tag{H2}$$

Inserting Equations (H1) and (H2) into the continuity equation for an incompressible fluid,

$$\frac{\partial u}{\partial x} + \frac{\partial v}{\partial y} + \frac{\partial w}{\partial z} = 0 \tag{H3}$$

we obtain the gradient of the $z$-component $w$ of the velocity fluctuation.

$$\frac{\partial w}{\partial z} = -\left(\frac{\partial^2}{\partial x^2} + \frac{\partial^2}{\partial y^2}\right)\phi_s \tag{H4}$$

Here, the $z$-component of the vorticity fluctuation is defined by

$$\omega_z \equiv \frac{\partial v}{\partial x} - \frac{\partial u}{\partial y} \tag{H5}$$

Substituting Equations (H1) and (H2) into (H5), we have

$$\omega_z = \left( \frac{\partial^2}{\partial x^2} + \frac{\partial^2}{\partial y^2} \right) \psi_s \tag{H6}$$

Then, Equations (H4) and (H6) are changed to the following amplitude equations by the 2D Fourier transform.

$$\frac{\partial W^0}{\partial z} = a^2 \Phi_s^0 \tag{H7}$$

$$\Omega^0 = -a^2 \Psi_s^0 \tag{H8}$$

where $\Phi_s^0$ and $\Psi_s^0$ are the amplitudes of the stream functions $\phi_s$ and $\psi_s$, respectively. The non-dimensional wave number $a$ is equal to $\left( a_x^2 + a_y^2 \right)^{1/2}$. After taking the Fourier transform of Equations (H1) and (H2) concerning $x$ and $y$, then substituting Equations (H7) and (H8) into the resulting equations, we obtain the amplitude equations of the $x$- and $y$-components of the velocity fluctuations $U^0$ and $V^0$, as follows.

$$U^0 = \frac{1}{a^2} \left( \frac{\partial^2 W^0}{\partial x \partial z} + \frac{\partial \Omega^0}{\partial y} \right) \tag{H9}$$

$$V^0 = \frac{1}{a^2} \left( \frac{\partial^2 W^0}{\partial y \partial z} - \frac{\partial \Omega^0}{\partial x} \right) \tag{H10}$$

Therefore, by determining the amplitudes of the $z$-components of the velocity and vorticity $W^0$ and $\Omega^0$, we can calculate the amplitudes of the $x$- and $y$-components of the velocity.

**Appendix I Solutions of the Amplitudes $W^0$ and $\Omega^0$ of the Fluctuations of Velocity and Vorticity in the Lower Layer**

Since the scale of length $d$ $\left( \approx 10^{-4} \text{ m} \right)$ and the electric conductivity $\sigma^*$ $\left( \approx 10 \text{ S m}^{-1} \right)$ are small, in an electrolytic system, the non-dimensional parameter $Q^*$ of electromagnetic induction can be disregarded (electrochemical approximation). However, to protect against a mistake according to the degeneration of the solution when neglecting $Q^*$, the equations to solve are treated with a limiting value of $Q^*$.

$$\left\{ \left( D^2 - a^2 \right)^2 - Q^* D^2 \right\} \Omega^0 = 0 \tag{D6a}$$

and

$$\left\{ \left( D^2 - a^2 \right)^2 - Q^* D^2 \right\} W^0 = 0 \tag{D6b}$$

As shown above, $\Omega^0$ and $W^0$ satisfy the same equation form. This means that both of them are expressed by the same function form.

We assume the function form of $W^0$ in the following,

$$W^0 = f(z, t) e^{\pm az} \tag{I1}$$

Namely, for $z \gg a^{-1}$, $W^0$ follows $e^{\pm az}$, whereas for $z \ll a^{-1}$, $W^0$ depends on $f(z, t)$, which is expressed by

$$f(z, t) = \alpha_0 + \alpha_1 z + \alpha_2 z + \cdots = \sum_{i=1} \alpha_{i-1} z^{i-1} \tag{I2}$$

where $\alpha_{i-1}$ is defined as a function of time. Here, for convenience, we derive the following two formulas.

$$D^n f(z, t) = \sum_{i=1} \frac{(n+i-1)!}{(i-1)!} \alpha_{n+i-1} z^{i-1} \tag{I3a}$$

and

$$D^n W^0 = e^{\pm az}(D \pm a)^n f(z,t) \tag{I3b}$$

where n $\geq$ 0 is required. Using Equation (I3b), we can easily rewrite the left-hand side of Equation (D6b).

$$\left\{\left(D^2 - a^2\right)^2 - Q^* D^2\right\} W^0 = e^{\pm az}\left\{D^4 \pm 4aD^3 + 4a^2 D^2 \mp 2Q^* aD - Q^* a^2\right\} f(z,t) \tag{I4}$$

where for almost all $a$'s, $a^2 \gg Q^*$ is considered. Using Equation (I3a), we furthermore rewrite Equation (I4) as

$$\left\{\left(D^2 - a^2\right)^2 - Q^* D^2\right\} W^0 \\ = e^{\pm az} \sum_{i=1} \left[\frac{(i+3)!}{(i-1)!}\alpha_{i+3} \pm 4\frac{(i+2)!}{(i-1)!}\alpha_{i+2}a + 4\frac{(i+1)!}{(i-1)!}\alpha_{i+1}a^2 \mp 2\frac{i!}{(i-1)!}Q^*\alpha_i a - Q^*\alpha_{i-1}a^2\right] z^{i-1} \tag{I5}$$

Equation (D6b) is therefore expressed by

$$\sum_{i=1} \left[\frac{(i+3)!}{(i-1)!}\alpha_{i+3} \pm 4\frac{(i+2)!}{(i-1)!}\alpha_{i+2}a + 4\frac{(i+1)!}{(i-1)!}\alpha_{i+1}a^2 \mp 2\frac{i!}{(i-1)!}Q^*\alpha_i a - Q^*\alpha_{i-1}a^2\right] z^{i-1} = 0 \tag{I6}$$

In Equation (I6), inserting i $= 1$ and $\alpha_3 = \alpha_4 = 0$, we first obtain the constant term,

$$8\alpha_2 a^2 \mp 2Q^*\alpha_1 a - Q^*\alpha_0 a^2 = 0 \tag{I7}$$

From Equation (I7), we have

$$\alpha_2 = \frac{Q^*}{8a}(\alpha_0 a \pm 2\alpha_1) \tag{I8}$$

With Equation (I8), the function f$(z,t)$ in Equation (I2) is expressed by

$$f(z,t) = \alpha_0 + \alpha_1 z + \frac{Q^*}{8a}(\alpha_0 a \pm 2\alpha_1)z^2 \tag{I9}$$

As mentioned above, in the present case, due to the low electric conductivity and the small scale of length, we adopt the following condition,

$$Q^* \ll a \tag{I10}$$

So, we can approximate f$(z,t)$ by

$$f(z,t) = \alpha_0 + \alpha_1 z \tag{I11}$$

From Equation (I1), the general equation of $W^0$ is thus provided by

$$W^0(z,t) = (\alpha_0 + \alpha_1 z)e^{az} + (\alpha_2 + \alpha_3 z)e^{-az} \tag{I12}$$

where $\alpha_0$, $\alpha_1$, $\alpha_2$, and $\alpha_3$ are arbitrary constants, which will be expressed by the functions of time. On the right-hand side of Equation (I12), the first term of $e^{az}$ and the second term of $e^{-az}$ correspond to the components surviving and disappearing at the outer boundaries of the vortices, respectively.

The vorticity is also activated at the upper boundary, and in view of the boundary conditions Equations (11a) and (11b), two arbitrary constants are necessary. This means that the vorticity depends only on $e^{az}$, so that $\Omega^0$ is expressed by

$$\Omega^0(z,t) = (\beta_0 + \beta_1 z)e^{az} \tag{I13}$$

where $\beta_0$ and $\beta_1$ are arbitrary constants.

Using the formula Equations (I3a) and (I3b), we obtain the following relationships:

$$DW^0(z,t) = \{\alpha_0 a + \alpha_1(1 + az)\}e^{az} + \{-\alpha_2 a + \alpha_3(1 - az)\}e^{-az} \tag{I14a}$$

and

$$D^2 W^0(z,t) = \left\{\alpha_0 a^2 + \alpha_1(2 + az)a\right\}e^{az} + \left\{\alpha_2 a^2 + \alpha_3(-2 + az)a\right\}e^{-az} \tag{I14b}$$

Then, from Equation (I13), we have

$$D\Omega^0(z,t) = \{\beta_0 a + \beta_1(1 + az)\}e^{az} \tag{I15a}$$

and

$$D^2 \Omega^0(z,t) = a\{\beta_0 a + \beta_1(2 + az)\}e^{az} \tag{I15b}$$

## Appendix J  Solution of the Amplitude $\Theta^0$ of the Concentration Fluctuation in the Lower Layer

The solution $\Theta^0$ is expressed by two kinds of solutions, i.e., general and special. From Equation (D3e), the general solution is obtained by the non-dimensional equation.

$$\left(D^2 - a^2\right)\Theta_g^0 = 0 \tag{J1a}$$

In terms of the boundary condition in Equation (14b), $\Theta^0 \to 0$ for $z \to 1$, we have

$$\Theta_g^0 = A_1 e^{-az} \tag{J1b}$$

where $A_1$ implies an arbitrary constant. The special solution will be obtained from the equation.

$$\left(D^2 - a^2\right)\Theta_s^0 = R^* W^0 \tag{J2a}$$

where $R^*$ is the mass transfer coefficient defined by

$$R^* \equiv \frac{L_m d^2}{D_m} \tag{J2b}$$

The solution is formally expressed by

$$\Theta_s^0 = \frac{R^*}{\left(D^2 - a^2\right)} W^0 \tag{J3}$$

Using the following formulas,

$$\frac{1}{\left(D^2 - a^2\right)}\left(ze^{\pm az}\right) = \pm\frac{1}{8a^3}\left(2a^2 z^2 \mp 2az + 1\right)e^{\pm az} \tag{J4a}$$

$$\frac{1}{\left(D^2 - a^2\right)}\left(e^{\pm az}\right) = -\frac{1}{4a^2}\left(\mp 2az + 1\right)e^{\pm az} \tag{J4b}$$

Substituting Equation (I12) into Equation (J3) and using Equations (J4a) and (J4b), we have

$$\Theta_s^0 = \frac{R^*}{8a^3}\left[\left\{-2\alpha_0 a(-2az + 1) + \alpha_1\left(2a^2 z^2 - 2az + 1\right)\right\}e^{az} + \left\{-2\alpha_2 a(2az + 1) - \alpha_3\left(2a^2 z^2 + 2az + 1\right)\right\}e^{-az}\right] \tag{J5}$$

The solution of $\Theta^0$ is expressed by

$$
\begin{aligned}
\Theta^0(z,t) &= \Theta_{\mathrm{g}}^0(z,t) + \Theta_{\mathrm{s}}^0(z,t) \\
&= A_1 \mathrm{e}^{-az} + \frac{R^*}{8a^3}\big\{ -2\alpha_0 a(-2az+1) + \alpha_1\big(2a^2z^2 - 2az + 1\big)\big\}\mathrm{e}^{az} \\
&\quad + \big\{ -2\alpha_2 a(2az+1) - \alpha_3\big(2a^2z^2 + 2az + 1\big)\big\}\mathrm{e}^{-az}
\end{aligned}
\tag{J6}
$$

Therefore, we have

$$
\begin{aligned}
\mathrm{D}\Theta^0(z,t) &= -aA_1 \mathrm{e}^{-az} \\
&\quad + \frac{R^*}{8a^2}\Big[ \big\{ 2\alpha_0 a(2az+1) + \alpha_1\big(2a^2z^2 + 2az - 1\big)\big\}\mathrm{e}^{az} \\
&\quad + \big\{ 2\alpha_2 a(2az-1) + \alpha_3\big(2a^2z^2 - 2az - 1\big)\big\}\mathrm{e}^{-az} \Big]
\end{aligned}
\tag{J7a}
$$

and

$$
\mathrm{D}\Theta^0(0,t) = -aA_1 + \frac{R^*}{8a^2}\big(2\alpha_0 a - \alpha_1 - 2\alpha_2 a - \alpha_3\big)
\tag{J4b}
$$

Then, we also have

$$
\Theta^0(0,t) = A_1 + \frac{R^*}{8a^3}\big(-2\alpha_0 a + \alpha_1 - 2\alpha_2 a - \alpha_3\big)
\tag{J7c}
$$

To determine the arbitrary constants, i.e., $A_1$, $\alpha_0$, $\alpha_1$, $\alpha_2$, and $\alpha_3$ of $\Theta^0(0,t)$ and $\mathrm{D}\Theta^0(0,t)$, the amplitude of the current density fluctuation $J^0$ is solved. Due to the low electric conductivity, the contribution of the electromagnetic induction to the current density is negligible, so Equation (D3b) is approximated by

$$
\big(\mathrm{D}^2 - a^2\big)J^0 = 0
\tag{J8}
$$

On the other hand, the vorticity is controlled by the fluctuation of the current density through Equation (D3c), which is rewritten with the non-dimensional wavenumber $a$ as

$$
\big(\mathrm{D}^2 - a^2\big)\Omega^0 = -S^*\mathrm{D}J^0
\tag{J9a}
$$

where $S^*$ represents the magneto-viscosity coefficient defined by

$$
S^* \equiv \frac{B_0 d}{\rho\nu}
\tag{J9b}
$$

From Equation (J8), the function form of $J^0$ is provided by

$$
J^0 = B_1 \mathrm{e}^{az}
\tag{J10}
$$

where $B_1$ implies an arbitrary constant, and $J^0$ satisfies Equation (14a). Here, Equations (I13) and (I15b) provide the following identity, i.e.,

$$
\big(\mathrm{D}^{*2} - a^2\big)\Omega^0 = 2\beta_1 a \mathrm{e}^{az}
\tag{J11}
$$

Substituting Equations (J10) and (J11) into Equation (J9a), we have

$$
B_1 = -\frac{2\beta_1}{S^*}
\tag{J12}
$$

Then, inserting Equation (J10) into Equation (14a), we obtain

$$
B_1 = -z_{\mathrm{m}} F D_{\mathrm{m}} \mathrm{D}\Theta^0(0,t)
\tag{J13}
$$

Substituting for $B_1$ from Equation (J12) in Equation (J13), we have

$$\text{D}\Theta^0(0,t) = \frac{2\beta_1}{z_\text{m}FD_\text{m}S^*} \tag{J14}$$

Substituting Equation (J14) into Equation (J7b), we have

$$A_1 = -\frac{2\beta_1}{z_\text{m}FD_\text{m}S^*a} + \frac{R^*}{8a^3}(2\alpha_0 a - \alpha_1 - 2\alpha_2 a - \alpha_3) \tag{J15}$$

Substitution for $A_1$ from Equation (J15) in Equation (J7c), we finally obtain

$$\Theta^0(0,t) = -\frac{2\beta_1}{z_\text{m}FD_\text{m}S^*a} - \frac{R^*}{4a^3}(2\alpha_2 a + \alpha_3) \tag{J16}$$

**Appendix K Derivation of the $x$- and $y$-Components of the Velocity in the Lower Layer in 2D Nucleation**

Using the relationships $\partial/\partial x = \text{i}a_\text{x}$, $\partial/\partial y = \text{i}a_\text{y}$ and $\text{d}/\text{d}z \equiv \text{D}$, from Equations (H9) and (H10) in Appendix H, we obtain the amplitudes of the $x$- and $y$-components of the velocity.

$$U_\text{j}^{0\text{a}} = \text{i}\cdot\frac{1}{a^2}\left(a_\text{x}\text{D}W_\text{j}^{0\text{a}} + a_\text{y}\Omega_\text{j}^{0\text{a}}\right) \text{ for } \text{j} = \text{r or f} \tag{K1a}$$

$$V_\text{j}^{0\text{a}} = \text{i}\cdot\frac{1}{a^2}\left(a_\text{y}\text{D}W_\text{j}^{0\text{a}} - a_\text{x}\Omega_\text{j}^{0\text{a}}\right) \text{ for } \text{j} = \text{r or f} \tag{K1b}$$

The unit imaginary number i introduced implies that the $x$- and $y$-components $u$ and $w$ are normal to the $z$-components $w$ and $\omega_\text{z}$.

(a) For the rigid surface:

The gradient of the amplitude of the $z$-component of the velocity is explicitly expressed as follows; from Equation (61a), we have

$$\text{D}W_\text{r}^0(z,t)^\text{a} = 2A_\text{r}(a)\exp(p_\text{r}^\text{a}t)[a\{\alpha_{0\text{r}}^{*\text{a}}(a) + \alpha_{1\text{r}}^{*\text{a}}(a)z\}\cos\text{h } az + \alpha_{1\text{r}}^{*\text{a}}(a)\sin\text{h } az - a(1-az)\alpha_{0\text{r}}^{*\text{a}}(a)\exp(-az)]R_\text{d}^\text{a} \tag{K2a}$$

From Equation (61b), we obtain the amplitude of the $z$-component of the vorticity.

$$\Omega_\text{r}^0(z,t)^\text{a} = A_\text{r}(a)\exp(p_\text{r}^\text{a}t)z\exp(az)R_\text{d}^\text{a} \tag{K2b}$$

where $A_\text{r}(a)$ is defined by

$$A_\text{r}(a) \equiv \gamma_1^\text{a}f_\text{r}^\text{a}(a)\exp\left(-\frac{a^2}{2}\right) \tag{K2c}$$

Substituting Equations (K2a) and (K2b) into Equations (K1a) and (K1b), we obtain the explicit forms of $U_\text{r}^0(z,t)^\text{a}$ and $V_\text{r}^0(z,t)^\text{a}$.

$$U_\text{r}^0(z,t)^\text{a} = \text{i}\cdot a^{-2}A_\text{r}(a)\exp(p_\text{r}^\text{a}t)\left[2a_\text{x}a\{\alpha_{0\text{r}}^{*\text{a}}(a) + \alpha_{1\text{r}}^{*\text{a}}(a)z\}\cos\text{h } az + 2a_\text{x}\alpha_{1\text{r}}^{*\text{a}}(a)\sin\text{h } az - 2a_\text{x}a(1-az)\alpha_{0\text{r}}^{*\text{a}}(a)\exp(-az) + a_\text{y}z\exp(az) * R_\text{d}^\text{a}\right] \tag{K3a}$$

and

$$V_\text{r}^0(z,t)^\text{a} = \text{i}\cdot a^{-2}A_\text{r}(a)\exp(p_\text{r}^\text{a}t)\left[2a_\text{y}a\{\alpha_{0\text{r}}^{*\text{a}}(a) + \alpha_{1\text{r}}^{*\text{a}}(a)z\}\cos\text{h } az + 2a_\text{y}\alpha_{1\text{r}}^{*\text{a}}(a)\sin\text{h } az - 2a_\text{y}a(1-az)\alpha_{0\text{r}}^{*\text{a}}(a)\exp(-az) - a_\text{x}z\exp(az)\right]R_\text{d}^\text{a} \tag{K3b}$$

(b) For the free surface:

The gradient of the amplitude of the $z$-component of the velocity is explicitly expressed as follows; from Equation (64a), we have

$$DW_f^0(z,t)^a = 2a^{-1}A_f(a)\exp(p_f^a t)[\{a\alpha_{0f}^{*a}(a) + \alpha_{1f}^{*a}(a)\}\cosh az + a\alpha_{1f}^{*a}(a)z\sinh az]R_d^a \tag{K4a}$$

From Equation (64b), we obtain the $z$-component of the amplitude of the vorticity.

$$\Omega_f^0(z,t)^a = a^{-1}A_f(a)\exp(p_f^a t)(1 - az)\exp(az)R_d^a \tag{K4b}$$

where we have

$$A_f(a) \equiv \gamma_0^a f_f^a(a)\exp\left(-\frac{a^2}{2}\right) \tag{K4c}$$

Substituting Equations (K4a) and (K4b) into Equations (K1a) and(K1b), we have

$$U_f^0(z,t)^a = i\cdot a^{-3}A_f(a)\exp(p_f^a t)\left[2a_x\{a\alpha_{0f}^{*a}(a) + \alpha_{1f}^{*a}(a)\}\cosh az + 2a_x a\alpha_{1f}^{*a}(a)\,z\sinh az + a_y(1 - az)\exp(az)\right]R_d^a \tag{K5a}$$

and

$$V_f^0(z,t)^a = i\cdot a^{-3}A_f(a)\exp(p_f^a t)\left[2a_y\{a\alpha_{0f}^{*a}(a) + \alpha_{1f}^{*a}(a)\}\cosh az + 2a_y a\alpha_{1f}^{*a}(a)z\sinh az - a_x(1 - az)\exp(az)\right]R_d^a \tag{K5b}$$

Due to the unit imaginary number i, $U_r^0(z,t)^a$ and $V_r^0(z,t)^a$ change their phases from even to odd, whereas $U_f^0(z,t)^a$ and $V_f^0(z,t)^a$ change their phases from odd to even. As a result, the free and rigid surface components are embedded into the real and imaginary parts of the complex amplitude, as follows:

$$\overline{C}U^0(z,t)^a = -U_f^0(z,t)^{a*}(\text{even}) + i\cdot U_r^0(z,t)^{a*}(\text{odd}) \tag{K6a}$$

and

$$\overline{C}V^0(z,t)^a = -V_f^0(z,t)^{a*}(\text{even}) + i\cdot V_r^0(z,t)^{a*}(\text{odd}) \tag{K6b}$$

where the sign '*' means the real component without i. From the relation $i^2 = -1$, minus signs are added to the free surface components.

The complex amplitudes of the $x$- and $y$-components of the velocity and vorticity fluctuations are transformed by the complex Fourier inversion to the complex $x$- and $y$-components of the velocity and vorticity fluctuations.

$$\overline{C}u(x,y,z,t)^a = \frac{1}{2\pi}\int_{-\infty}^{\infty}\int_{-\infty}^{\infty} C\,U^0(z,t)^a\exp\left[-i\left(a_x x + a_y y\right)\right]da_x da_y \tag{K7a}$$

and

$$\overline{C}v(x,y,z,t)^a = \frac{1}{2\pi}\int_{-\infty}^{\infty}\int_{-\infty}^{\infty}\overline{C}V^0(z,t)^a\exp\left[-i\left(a_x x + a_y y\right)\right]da_x da_y \tag{K7b}$$

where the $x$- and $y$-components of a complex velocity fluctuation, with respect to the $x$- and $y$-coordinates, are obtained as follows,

$$\overline{C}u(x,y,z,t)^a = -u_f(x,y,z,t)^a(\text{even}) + i\cdot u_r(x,y,z,t)^a(\text{odd}) \tag{K8a}$$

and

$$C\,v(x,y,z,t)^a = -v_f(x,y,z,t)^a(\text{even}) + i\cdot v_r(x,y,z,t)^a(\text{odd}) \tag{K8b}$$

The total components of the velocity are described by the odd and even functions.

$$u(x,y,z,t)^a = u_r(x,y,z,t)^a(\text{odd}) + u_f(x,y,z,t)^a(\text{even}) \tag{K9a}$$

and

$$v(x,y,z,t)^a = v_r(x,y,z,t)^a(\text{odd}) + v_f(x,y,z,t)^a(\text{even}) \tag{K9b}$$

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
