# Peer review of "Theory of Chiral Electrodeposition by Chiral Micro-Nano-Vortices under a Vertical Magnetic Field -1: 2D Nucleation by Micro-Vortices"

_magnetochemistry, doi:10.3390/magnetochemistry8070071_

Round 1

Reviewer 1 Report

This manuscript by Prof. Morimoto and prof. Aogaki et al. describes theoretical investigation of chiral electrodeposition under the influence of magnetic field. The author states that under a vertical magnetic field vertical magnetohydrodynamic flow occurs on the electrode surface, which facilitates remarkable chiral activity. The magnetohydrodynamic flow creates 3 kinds of chiral vortexes: micro-, nano- and ultra-micro vortexes. They lead to an enantiomeric excess ratio up to 0.125.

The theoretical development is interesting and is complemented by experiments. The results clearly explain the ee ratios on the electrode and the discrepancy between the experimental values and the calculated one. The results of the manuscript has warranted the acceptance to the Journal except for the following issue.

Concerning the Cl- adsorption, it is experimentally measured that the values of the differential charge coefficient (eq. 76b and 76c) are negative, leading to negative values of the differential potential coefficient (eq. 75). It is uncommon to find that strong Cl- adsorption is present on the cathode surface. Since the differential charge coefficient is experimentally obtained, how accurate is it?

This Reviewer finds that CuSO4 and H2SO4 in analytical grade are used, not ultra-pure grade. Since some ions other than Cu2+, SO42-, H+ present in the analytical grade chemicals may alter the observed results, it is recommended to doubly check the results.

Reviewer 2 Report

Manuscript: ID magnetochemistry-1774003   Title: Theory of Chiral Electrodeposition by Chiral Micro- and Nano-Vortexes under a Vertical Magnetic Field -1. 2D Nucleation by Micro-Vortexes

The ms is interesting and worth publication, although it is a rather complex paper. The writing is rather obscure. Authors should go through the intro and theoretical sections making them simpler and clearer.  Some minor points in the following.

MINOR POINTS (obscure writing)

Intro

In recent years, it has been found that ionic vacancies are produced in solution phases  as byproducts of electrode reactions [1, 2], which are charged particles created to keep the conservations of linear momentum and electricity during electron transfers in electrode  reactions. The created embryo vacancies are energetically unstable so that they are stabilized by solvation, surrounded by ionic clouds. Using the solvation energies emitted from the ionic clouds, as well known, ions in free space produce entropies in solutions, whereas  since embryo vacancies utilize the energies to enlarge their own cores, the solvation of  embryo vacancies does not produce entropies.”  The all text is totally obscure, if not meaningless, the authors must rewrite it, it is not clear nor the physical meaning nor the logic. What is the subject of “which” ? What is an  “emitted solvation energy” ? “the solvation of embryo vacancies does not produce entropies” why “entropies” ?

Theory

Line 221 “the solution around vortexes is assumed zero if possible.” What is meant by “the solution” ?

IT seems that equations D1c and D1d are identical

Reviewer 3 Report

The topic considered in the paper seems suitable to Magnetochemistry, and the experts in the subject area would likely find this work interesting and relevant. It is thus acceptable for publication. I have only a few minor suggested corrections.

1. The numbering of the formulas looks strange, and starts with (B7). The numbering of the formulas should be brought to a common form.

2. The article points out that Fig. 11 is the result of theoretical calculations. It is not quite clear why the calculated surfaces are not smooth and look more like SEM images. It should be explained how exactly the calculations were made (according to which expressions), presented in Fig. 11.

After clarification of these remarks, I recommend the paper for publication.
